# TR2-D2: Tree Search Guided Trajectory-Aware Fine-Tuning for Discrete Diffusion

## Abstract

Reinforcement learning with stochastic optimal control offers a promising framework for diffusion fine-tuning, where a pre-trained diffusion model is optimized to generate paths that lead to a reward-tilted distribution. While these approaches enable optimization without access to explicit samples from the optimal distribution, they require training on rollouts under the current fine-tuned model, making them susceptible to reinforcing sub-optimal trajectories that yield poor rewards. To overcome this challenge, we introduce **TR**ee Search Guided **TR**ajectory-Aware Fine-Tuning for **D**iscrete **D**iffusion (**TR2-D2**), a novel framework that optimizes reward-guided discrete diffusion trajectories with tree search to construct replay buffers for trajectory-aware fine-tuning. These buffers are generated using Monte Carlo Tree Search (MCTS) and subsequently used to fine-tune a pre-trained discrete diffusion model under a stochastic optimal control objective. We validate our framework on single- and multi-objective fine-tuning of biological sequence diffusion models, highlighting the overall effectiveness of TR2-D2 for reliable reward-guided fine-tuning in discrete sequence generation.

## 1 Introduction

Diffusion generative models (Sohl-Dickstein et al., 2015; Song et al., 2020a; Ho et al., 2020) have led to significant advancements across continuous video and image generation, and more recently in discrete state spaces (Austin et al., 2021) for natural language (Sahoo et al., 2024; Nie et al., 2025; Khanna et al., 2025; Song et al., 2025; Team et al., 2023) and biomolecular sequence generation (Avdeyev et al., 2023; Alamdari et al., 2023; Hayes et al., 2025). Inference-time guidance and fine-tuning of diffusion models have enabled the repurposing of pre-trained diffusion models for highly specialized tasks, such as accurate text-to-image generation (Ruiz et al., 2023; Voynov et al., 2023) and design of biomolecules with therapeutic properties (Gruver et al., 2023; Tang et al., 2025; Wang et al., 2025). These methods aims to sample from the data distribution $p_{\text{data}}$ tilted by a **reward function** $r(\boldsymbol{X})$, which amplifies the density of high-reward samples $p_{\text{target}}(\boldsymbol{X}) \propto p_{\text{data}}(\boldsymbol{X}) \exp(r(\boldsymbol{X})/\alpha)$ and minimizes sub-optimal samples. While inference-time guidance avoids model training, it incurs increased inference costs due to reward evaluations and does not prevent the model from generating suboptimal samples, particularly in regions of high data density. Alternatively, fine-tuning is theoretically guaranteed to fit the reward-tilted distribution, which permanently modifies the model's terminal distribution with inexpensive inference calls.

An effective strategy for fine-tuning involves **off-policy reinforcement learning (RL)** (Bengio et al., 2021; Peng et al., 2019) that uses trajectories generated by reference policy models to inform the next update to the current policy. However, its effectiveness in practice is limited by the quality of the trajectories generated from the policy. This motivates advancements in optimizing diffusion trajectories, which have been explored in continuous state spaces with differentiable gradients along the diffusion trajectory (Tian et al., 2025), but remain challenging in discrete state spaces where gradients are undefined. To this end, we introduce **TR**ee Search Guided **TR**ajectory-Aware Fine-Tuning for **D**iscrete **D**iffusion (**TR2-D2**), which leverages tree search to generate reward-guided trajectories for off-policy RL for discrete diffusion fine-tuning.

**Contributions** Our main contributions can be summarized as follows: **(1)** We develop a general framework for enhancing off-policy RL techniques with search-optimized discrete diffusion

trajectories (Sec 3). **(2)** We implement our framework to develop an efficient discrete diffusion fine-tuning strategy that leverages Monte-Carlo Tree Search (MCTS) to curate a replay buffer of optimal trajectories for off-policy control-based RL (Sec 4). **(3)** We introduce the **first** method for **multiobjective fine-tuning** of discrete diffusion models by generating Pareto-optimal replay buffers for fine-tuning (Sec 5). **(4)** We demonstrate that **TR2-D2** achieves state-of-the-art performance in discrete diffusion fine-tuning for **regulatory DNA design optimized for enhancer activity** (Sec 6.1) and **multi-objective therapeutic peptide design** (Sec 6.2).

**Related Works**    We provide a comprehensive discussion of related works in App A.

## 2    PRELIMINARIES

**Continuous-Time Markov Chains**    A continuous-time Markov chain (CTMC) defines a stochastic process $\boldsymbol{X}_{0:T} = (\boldsymbol{X}_t)_{t \in [0,T]}$ over a discrete state space $\mathcal{X} = \{1, \dots, D\}$. The evolution and law of a CTMC is characterized by a *generator* $(\boldsymbol{Q}_t)_{t \in [0,T]} \in \mathbb{R}^{\mathcal{X} \times \mathcal{X}}$, defined by

$$\boldsymbol{Q}_t(x, y) = \lim_{\Delta t \to 0} \frac{1}{\Delta t}(\Pr(\boldsymbol{X}_{t+\Delta t} = y | \boldsymbol{X}_t = x) - \mathbf{1}_{x=y})$$

whose value $\boldsymbol{Q}_t(x, y)$ describes the transition rate from a state $x \in \mathcal{X}$ to another state $y \in \mathcal{X}$. We refer to Appendix B.1 for a theoretical background of CTMCs and relevant stochastic calculus tools.

**Discrete Diffusion Models**    Discrete diffusion models are a class of generative models that aim to learn the generator of a CTMC, which starts from an easy-to-sample prior distribution $p_{\text{prior}}$ and arrives at the target distribution $p_{\text{data}}$ in finite time. Discrete diffusion models consist of a pair of noise-injection and generative denoising CTMCs, which are the time-reversal of each other.

An effective formulation for discrete diffusion is the **masked discrete diffusion model (MDM)** (Sahoo et al., 2024; Shi et al., 2024; Ou et al., 2024; Zheng et al., 2024), where the prior distribution is chosen to be the Dirac distribution concentrated on a sequence where all tokens are an absorbing mask token, denoted as $\boldsymbol{M}$. The forward process of MDM injects noise into the sequence by independently converting data tokens into the mask token following a noise scheduler. The backward generative process reverses this process by starting from a fully-masked sequence and iteratively decoding masks back to data tokens, following a parameterized probability distribution conditioned on the previously unmasked tokens in the sequence.

Let $\boldsymbol{X} \in \mathcal{X}^L$ be a partially masked sequence of $L$ tokens, $\boldsymbol{X}^{\text{UM}} = (\boldsymbol{X}^\ell : \boldsymbol{X}^\ell \neq \boldsymbol{M})$ denotes the collection of non-mask token in $\boldsymbol{X}$, and $\boldsymbol{X}^{\ell \leftarrow d}$ represents the sequence modified from $\boldsymbol{X}$ by replacing the $\ell$-th position with data token $d$, it's proven in Ou et al. (2024) that the optimal generator of the generative process for MDM has the following special decomposition,

$$\boldsymbol{Q}_t(\boldsymbol{x}, \boldsymbol{y}) = \gamma(t) \Pr_{\boldsymbol{X} \sim p_{\text{data}}} (\boldsymbol{X}^\ell = d | \boldsymbol{X}^{\text{UM}} = \boldsymbol{x}^{\text{UM}}) \mathbf{1}_{\boldsymbol{x}^\ell = d, \boldsymbol{y} = \boldsymbol{x}^{\ell \leftarrow d}} \tag{1}$$

where $\gamma(t)$ is a noise schedule, $\boldsymbol{x}, \boldsymbol{y} \in \mathcal{X}^L$. Due to this special structure, MDM often adopts a neural network $p^{u_\theta}(\cdot | \boldsymbol{x}) \in \mathbb{R}^{N \times D}$ to parametrize the unknown conditional data distribution, where the $(\ell, d)$th entry of $p^{u_\theta}(\cdot | \boldsymbol{x})$ approximates $\Pr_{\boldsymbol{X} \sim p_{\text{data}}}(\boldsymbol{X}^\ell = d | \boldsymbol{X}^{\text{UM}} = \boldsymbol{x}^{\text{UM}})$. MDMs are often trained by optimizing the **denoising cross-entropy (DCE)** loss (Ou et al., 2024; Sahoo et al., 2024; Shi et al., 2024), defined as

$$\min_\theta \mathbb{E}_{\boldsymbol{x} \sim p_{\text{data}}}[\mathcal{L}(\theta; \boldsymbol{x})], \ \mathcal{L}(\theta; \boldsymbol{x}) := \mathbb{E}_{\lambda \sim \text{Unif}(0,1)} \left[ \frac{1}{\lambda} \mathbb{E}_{\mu_\lambda(\tilde{\boldsymbol{x}} | \boldsymbol{x})} \sum_{\ell : \tilde{\boldsymbol{x}}^\ell = \boldsymbol{M}} -\log p^{u_\theta}(\tilde{\boldsymbol{x}})_{\ell, \boldsymbol{x}^\ell} \right] \tag{2}$$

where $\mu_\lambda(\cdot | \boldsymbol{x})$ is a transition kernel that independently turns tokens in $\boldsymbol{X}$ with probability $\lambda$.

**Reinforcement Learning for Discrete Diffusion Models**    Although discrete diffusion models are capable of accurately capturing the distribution of training data, they often fail in specialized downstream tasks that aim to generate sequences that optimize custom reward functions. Reinforcement learning (RL) can be used to align the marginal of the pre-trained model with some desired terminal reward. Given a pre-trained discrete diffusion model that can sample from $p_{\text{data}}$, and a reward

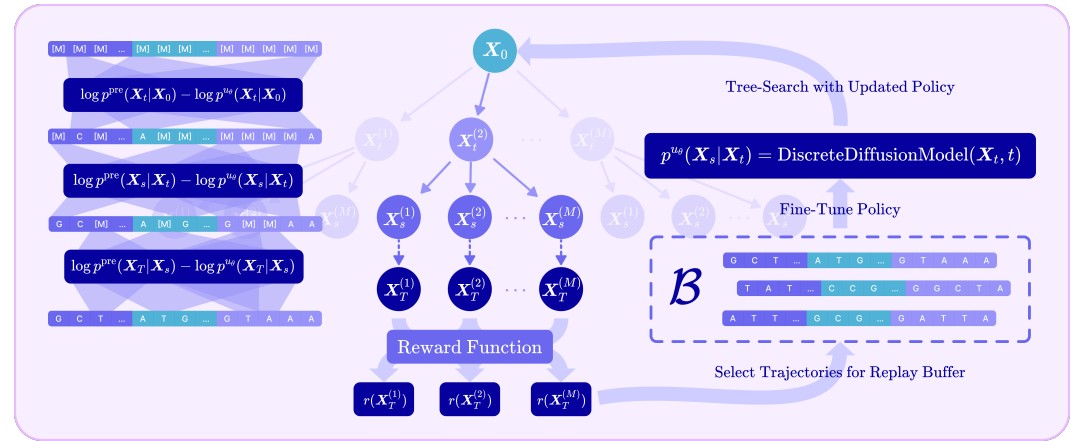

Figure 1: **Tree Search Guided Trajectory-Aware Fine-Tuning for Discrete Diffusion.** Our framework has two key components: **(1) a tree search algorithm** to generate a replay buffer of diffusion trajectories optimized for one or more reward functions using the current policy and **(2) an off-policy RL algorithm** for discrete diffusion fine-tuning using the optimized replay buffer.

function $r(\boldsymbol{X}) : \mathcal{X}^L \to \mathbb{R}$, RL can be used to align the $\theta$-parameterized policy model to the desired reward-tilted distribution by solving an **entropy-regularized reward optimization** problem (Uehara et al., 2024a).

$$\max_\theta \mathbb{E}_{\boldsymbol{X}_{0:T} \sim \mathbb{P}^{u_\theta}} \left[ r(\boldsymbol{X}_T) \right] - \alpha \, \mathrm{KL}(\mathbb{P}^{u_\theta} || \mathbb{P}^{\mathrm{pre}}) \tag{3}$$

where $\mathbb{P}^{u_\theta}$ and $\mathbb{P}^{\mathrm{pre}}$ correspond to the path measure of the CTMCs associated with the finetuned and pre-trained diffusion models, respectively, and $\alpha$ controls the strength of the KL-divergence regularization, with a smaller $\alpha$ value indicating greater tolerance to deviation from the pre-trained model. Zhu et al. (2025c) shows that the fine-tuned model that optimally solves (3) produces the following path measure that reaches a reward-tilted target distribution

$$\mathbb{P}^*(\boldsymbol{X}_{0:T}) = \mathbb{P}^{\mathrm{pre}}(\boldsymbol{X}_{0:T}) \frac{1}{Z} \exp\left( \frac{r(\boldsymbol{X}_T)}{\alpha} \right), \quad \mathbb{P}_T^*(\boldsymbol{X}) \propto p_{\mathrm{data}}(\boldsymbol{X}) \exp\left( \frac{r(\boldsymbol{X})}{\alpha} \right) =: p_{\mathrm{target}}(\boldsymbol{X}) \tag{4}$$

Therefore, the finetuning process naturally connects to solving a **stochastic optimal control (SOC)** problem for CTMC (Wang et al., 2025; Zhu et al., 2025c), and the reward optimization problem can be solved by matching the path measure $\mathbb{P}^{u_\theta}$ produced by the finetuned policy to the optimal path measure $\mathbb{P}^*$ through optimizing a loss that takes the general form $\min_\theta \mathcal{F}(\mathbb{P}^*, \mathbb{P}^{u_\theta})$. Common choices for $\mathcal{F}$ include log-variance (Nüsken & Richter, 2021), relative entropy (Wang et al., 2025; Zekri & Boullé, 2025; Cao et al., 2025), weighted denoising cross-entropy (Zhu et al., 2025c), among others. We refer to Appendix B.3 for a detailed discussion of the connection between SOC and RL fine-tuning.

## 3 ENHANCING REINFORCEMENT LEARNING WITH STRUCTURED SEARCH

The effectiveness of RL in optimizing customized reward functions is **largely dependent on the ability of the pre-trained model to generate high-reward samples**. The model can then learn to reinforce these "positive" samples through RL and iteratively improve the quality of the next generation round. However, for a discrete state space $\mathcal{X} = \{1, \ldots D\}^L$ with $D$ states and $L$ token positions, the search space contains $D^L$ possible sequences, which becomes intractably large even for modest values of $D$ and $L$. When generating highly structured discrete data, such as biological sequences that optimize some reward function, it is common for high-reward sequences to lie in low-density regions of the search space that are rarely sampled by the pre-trained model (De Santi et al., 2025).

To avoid sub-optimal RL rounds due to low-quality trajectories, **supervised fine-tuning (SFT)** is commonly used to warm up the model by adapting it to generate favorable sequences through training on curated, specially-designed datasets. For example, when training LLMs into powerful

---

**Algorithm 1** Framework for **Enhanced Reinforcement Learning with Structured Search**

---

1: **Input:** pre-trained model $p^{\text{pre}}$, finetune policy model $p^{u_\theta}$, reward function $r$, and off-policy RL algorithm
2: Initalize finetune policy model $p^{u_\theta} = p^{\text{pre}}$
3: **while** not converged **do**
4:     **while** buffer $\mathcal{B}$ is not full **do**
5:         `Generate` samples from current policy model $p^{u_\theta}$
6:         `Select` optimal samples that maximize the reward function and add to buffer
7:         `Explore` similar samples given previous selections using the `Search` algorithm
8:     **end while**
9:     `Update` $\theta$ using samples from $\mathcal{B}$ using off-policy RL algorithm for multiple epochs
10:     `Reset` the buffer $\mathcal{B}$
11: **end while**

---

math reasoners, it is standard practice to perform SFT on math-domain-related datasets to produce some level of reasoning capability, and enhance it using RL approaches such as GRPO (Shao et al., 2024). For general downstream tasks, such as biological sequence optimization, labeled datasets for specialized tasks are sparse, making RL finetuning for these problems challenging.

To address this challenge, we aim to provide the policy model with a pseudo-warm-up that utilizes **reward-guided inference time scaling** techniques for the discrete diffusion fine-tuning. While random samples from the model are not guaranteed to have high rewards, optimal samples can often be obtained by scaling the inference budget and performing an extensive search of the sample space. Using **structured search algorithms** like Monte Carlo Tree Search (MCTS; Coulom (2006)) that discover and bias towards highly optimal regions of the sample space implicitly aligns with several optimization tasks where high-reward samples follow a common structure or contain similar motifs. The sequences obtained from the search can approximately serve as a specialized dataset that guides the discrete diffusion policy model to produce similar high-quality samples, accelerating RL training.

To maximize the utility of the collection of sequences found through the search, we add them to a *replay buffer* $\mathcal{B}$ and adopt **off-policy RL** algorithms which amplify the signal of the highly optimal sequences found during the search by training repetitively on samples from $\mathcal{B}$ over multiple iterations. This can amortize the inference computation cost incurred during buffer curation and further enhance the training efficiency. Furthermore, integrating structured search to curate the buffer leverages the ability of off-policy RL to memorize and reinforce buffer samples, improving the next round of buffer generation.

We summarize the high-level idea of combining search algorithms with RL finetuning of discrete diffusion models in Algorithm 1. One **major benefit** is that the search and finetuning steps in this framework are **decoupled**, opening the design space to any pair of search and off-policy RL algorithms. We further discuss this unique characteristic in Appendix C.3. In Section 4, we provide a specific implementation of this general framework, specifically tailored to fine-tuning of masked discrete diffusion models.

## 4   TR2-D2: Tree Search Guided Trajectory-Aware Fine-Tuning

To implement the framework discussed in Sec 3, we introduce **TRee-Guided TRajectory Planning for Discrete Diffusion (TR2-D2)** that integrates an off-policy RL-based fine-tuning algorithm coupled with Monte-Carlo Tree Search for reward-guided buffer generation. Notably, our implementation uses an off-policy RL algorithm with a **scalable objective function** (Sec 4.1) and efficiently **balances exploration and exploitation** of arbitrary reward functions (Sec 4.2).

### 4.1   Off-Policy RL for Masked Discrete Diffusion Models

To perform RL with discrete diffusion models for sampling from a reward-tilted distribution $p_{\text{target}}(\boldsymbol{X}) \propto p_{\text{data}}(\boldsymbol{X}) \exp(r(\boldsymbol{X})/\alpha)$, it suffices to find a CTMC that produces a path measure $\mathbb{P}^{u_\theta}$ that matches to the optimal path measure as in (4) (Zhu et al., 2025c). In the case of fine-tuning masked discrete diffusion models (MDM), the optimal CTMC is fully characterized by the conditional

---

**Algorithm 2** `TR2-D2`: Tree Search Guided Trajectory-Aware Fine-Tuning for Discrete Diffusion

---

1: **Input:** pre-trained model $p^{\text{pre}}(\cdot|\boldsymbol{X}_t^{\text{UM}})$, finetuned policy model $p^{u_\theta}(\cdot|\boldsymbol{X}_t^{\text{UM}})$, reward function $\boldsymbol{r} : \mathcal{X} \to \mathbb{R}^K$, number of finetuning epochs $N_{\text{epoch}}$, number of WDCE repeats $R$
2: **for** epoch in $1, \ldots, N_{\text{epoch}}$ **do**
3:     $\{\boldsymbol{X}^i, W^{\bar{u}}\}_{i=1}^B \leftarrow \texttt{MCTS}(p^{\text{pre}}, p^{u_\theta})$          ▷ *see Alg 5*
4:     $\mathcal{B} \leftarrow \{\boldsymbol{X}^i, W^{\bar{u}}\}_{i=1}^B$          ▷ *optimize replay buffer*
5:     **for** step in $1, \ldots, N_{\text{step}}$ **do**
6:        $\{\tilde{\boldsymbol{X}}^i, W^{\bar{u}}\}_{i=1}^{B \times R} \leftarrow \texttt{ResampleWithMask}(\mathcal{B}; R)$
7:        Compute $\mathcal{F}_{\text{WDCE}}$ from (7) with $\{\tilde{\boldsymbol{X}}^i, W^{\bar{u}}\}_{i=1}^{B \times R}$
8:        Update $\theta$ with $\nabla_\theta \mathcal{F}_{\text{WDCE}}$
9:     **end for**
10: **end for**

---

probability $p^*$, defined as

$$p^*(\cdot|\boldsymbol{x})_{\ell,d} = \Pr_{\boldsymbol{X} \sim p_{\text{target}}} (\boldsymbol{X}^\ell = d | \boldsymbol{X}^{\text{UM}} = \boldsymbol{x}^{\text{UM}}) \tag{5}$$

We note that the optimal solution stated in (5) shares a similar form to the pre-trained MDM, that outputs the conditional distribution with respect to $p_{\text{data}}$ as in (1) and can be learned with the denoising cross entropy objective in (2) when i.i.d. samples from $p_{\text{data}}$ are available. Therefore, we can naively learn $p^*$ by minimizing the following loss,

$$\min_\theta \mathbb{E}_{\boldsymbol{x} \sim p_{\text{target}}}[\mathcal{L}(\theta; \boldsymbol{x})] \tag{6}$$

where $\mathcal{L}(\theta; \boldsymbol{x})$ is the data-conditioned denoising cross entropy term defined in (2). In the case of diffusion fine-tuning, we lack access to i.i.d. samples from the desired distribution $p_{\text{target}} \propto p_{\text{data}} \exp(r(\boldsymbol{X}_T)/\alpha)$, making (6) an intractable objective. Recently, MDNS (Zhu et al., 2025c) introduced the **weighted denoising cross-entropy (WDCE)**, a tractable implementation of (6) that leverages importance sampling over the space of trajectories to simulate $p_{\text{target}}$. As shown in Appendix C.1, we can derive the WDCE objective by rewriting (6) using $\mathbb{P}^*$ since its marginal at time $T$ is exactly $p_{\text{target}}$,

$$\mathbb{E}_{p_{\text{target}(\boldsymbol{x})}}[\mathcal{L}(\theta; \boldsymbol{x})] = \mathbb{E}_{\boldsymbol{X}_{0:T} \sim \mathbb{P}^*}[\mathcal{L}(\theta; \boldsymbol{X}_T)] = \mathbb{E}_{\boldsymbol{X}_{0:T} \sim \mathbb{P}^v} \left[ \frac{\mathrm{d}\mathbb{P}^*}{\mathrm{d}\mathbb{P}^v}(\boldsymbol{X}_{0:T}) \mathcal{L}(\theta; \boldsymbol{X}_T) \right] := \mathcal{F}_{\text{WDCE}} \tag{7}$$

where $\mathbb{P}^v$ is a reference path measure that does not track the gradient with respect to $\theta$, and $\frac{\mathrm{d}\mathbb{P}^*}{\mathrm{d}\mathbb{P}^v}$ is the **Radon-Nikodým (RN) derivative** between the CTMC path measures $\mathbb{P}^*$ and $\mathbb{P}^v$, and can be interpreted as an importance weight that measures how closely the two path measures are aligned. We remark that the loss $\mathcal{F}_{\text{WDCE}}$ is considered **off-policy** as the reference policy $v$ used to generate training samples does not need to be updated as the fine-tuned policy $u_\theta$ is updated.

In practice, we periodically align the reference policy with the current model policy $u_\theta$ to control the variance of the importance weights for enhanced numerical stability. As derived in Appendix C.1, the RN derivative for MDM can be computed as,

$$\log \frac{\mathrm{d}\mathbb{P}^\star}{\mathrm{d}\mathbb{P}^v}(\boldsymbol{X}_{0:T}) = \underbrace{\frac{r(\boldsymbol{X}_T)}{\alpha} + \sum_{t:\boldsymbol{X}_s \neq \boldsymbol{X}_t} \sum_{\ell:\boldsymbol{X}_s^\ell \neq \boldsymbol{X}^\ell} \log \frac{p^{\text{pre}}(\boldsymbol{X}_s^\ell|\boldsymbol{X}_t^{\text{UM}})}{p^v(\boldsymbol{X}_s^\ell|\boldsymbol{X}_t^{\text{UM}})} - \log Z}_{:=W^v(\boldsymbol{X}_{0:T})} \tag{8}$$

where the normalizing constant $Z$ is approximated with $\mathbb{E}_{\boldsymbol{X}_{0:T} \sim \mathbb{P}^v} \exp(W^v(\boldsymbol{X}_{0:T}))$. In practice, we take the `softmax` over the importance weights $W^v$ in the batch, which approximates the expectation. Since this objective requires only the clean sequence $\boldsymbol{X}_T$ and the corresponding log-RND weight $W^v$, we store the generated rollouts in the replay buffer $\mathcal{B}$ in the form of pairs $(\boldsymbol{X}_T, W^v)$. We use the notation $v$ to emphasize the off-policy nature of the loss function. In practice, we always choose the non-gradient tracking policy $v = \bar{u} := \texttt{stopgrad}(u_\theta)$ to generate the sample batch.

## 4.2 STRUCTURED TREE SEARCH FOR BUFFER GENERATION

Given its success in inference-time guidance of MDMs (Tang et al., 2025), we leverage Monte-Carlo Tree Search (MCTS) (Coulom, 2006) as the **structured tree search algorithm** used to

optimize the buffer of unmasking trajectories for off-policy RL. The algorithm iterates over four steps (selection, expansion, rollout, and backpropagation), which effectively balances **exploration of diverse unmasking steps** and **exploitation of optimal rollouts**.

**Initialization** We define a tree $\mathcal{T}$, where each node is represented by a partially unmasked sequence $\boldsymbol{X}_s^i \in \mathcal{X}$, a *total reward* $R(\boldsymbol{X}_s^i) \in \mathbb{R}$ that determines the potential of the node to generate a high-reward sequence, the number of times the node was visited $N_{\text{visits}}(\boldsymbol{X}_s^i)$, and a set of children nodes $\text{children}(\boldsymbol{X}_s^i)$. Each node also stores the log-probability of sampling it given its parent $\boldsymbol{X}_t$ under the pre-trained model $p^{\text{pre}}(\boldsymbol{X}_s^i|\boldsymbol{X}_t^{\text{UM}})$ where $\boldsymbol{X}_t = \text{parent}(\boldsymbol{X}_s^i)$. The tree is stored as a set of nodes linked together by child and parent references. A node is considered *expandable* if it has no child nodes and is not fully unmasked (i.e. $t \neq T$). At initialization, the tree has a single root node defined as the fully masked sequence $\boldsymbol{X}_0 = [\boldsymbol{M}]^L$ with the number of visits set to $N_{\text{visits}}(\boldsymbol{X}_0) = 1$ and an empty set of child nodes.

**Selection** Starting from the root node, we traverse the existing unmasking steps defined in the tree by selecting from the $M$ child nodes at each intermediate node. To do this, we define the *selection reward* which guides exploration as

$$U(\boldsymbol{X}_t, \boldsymbol{X}_s^i) = \frac{R(\boldsymbol{X}_s^i)}{N_{\text{visits}}(\boldsymbol{X}_s^i)} + c \cdot p^{u_\theta}(\boldsymbol{X}_s^i|\boldsymbol{X}_t)\frac{\sqrt{N_{\text{visit}}(\boldsymbol{X}_t)}}{1 + N_{\text{visit}}(\boldsymbol{X}_s^i)} \quad (9)$$

Then, a child node is selected by sampling from the nodes with optimal selection rewards. In practice, we take the $\text{softmax}$ over $U(\boldsymbol{X}_t, \boldsymbol{X}_s^i)$ for the top-$k$ child nodes, where $k$ is a tunable hyperparameter, to avoid the chance of selecting nodes with low rewards.

**Expansion** After reaching an *expandable* node at time $t$, we sample $M$ *child* sequences $\{\boldsymbol{X}_s^i\}_{i=1}^M$ corresponding to the time $s = t + \Delta t$ by unmasking tokens using the condition probability of the current policy $p^{u_\theta}$. To ensure diversity in the samples, we perturb the predicted distribution with i.i.d. Gumbel noise before sampling each child sequence.

$$\boldsymbol{X}_s^i \leftarrow \text{SingleReverseStep}\left(\log p^{u_\theta}(\cdot|\boldsymbol{X}_t^{\text{UM}}) + \boldsymbol{G}_i, t\right)$$
$$\text{where } \boldsymbol{G}_i \sim -\log(-\log\mathcal{U}), \ \mathcal{U} \sim \text{Unif}(0, 1) \quad (10)$$

For each expanded node $\boldsymbol{X}_s^i$, we compute the log-probability of sampling the token under the pre-trained model and the current policy to get the log-RND weight of the step as

$$\text{log\_rnd}_i = \log\frac{p^{\text{pre}}(\boldsymbol{X}_s^i|\boldsymbol{X}_t^{\text{UM}})}{p^{u_\theta}(\boldsymbol{X}_s^i|\boldsymbol{X}_t^{\text{UM}})} = \sum_{\boldsymbol{X}_s^{i,\ell} \neq \boldsymbol{X}_t^\ell} \log\frac{p^{\text{pre}}(\boldsymbol{X}_s^{i,\ell}|\boldsymbol{X}_t^{\text{UM}})}{p^{u_\theta}(\boldsymbol{X}_s^{i,\ell}|\boldsymbol{X}_t^{\text{UM}})} \quad (11)$$

**Rollout** For each expanded node $\boldsymbol{X}_s^i$, we iteratively unmask the remaining masked tokens for the remaining timesteps until reaching a fully unmasked sequence $\boldsymbol{X}_T^i$. At each step, we track the running log-RND of the trajectory, which will be used in the training objective.

$$\boldsymbol{X}_s^i \leftarrow \text{SingleReverseStep}\left(\log p^{u_\theta}(\cdot|\boldsymbol{X}_t^{\text{UM}}), t\right) \quad (12)$$

$$W^{u_\theta}(\boldsymbol{X}_{0:T}^i) \leftarrow W^{u_\theta}(\boldsymbol{X}_{0:T}^i) + \sum_{\boldsymbol{X}_s^{i,\ell} \neq \boldsymbol{X}_t^\ell} \log\frac{p^{\text{pre}}(\boldsymbol{X}_s^{i,\ell}|\boldsymbol{X}_t^{\text{UM}})}{p^{u_\theta}(\boldsymbol{X}_s^{i,\ell}|\boldsymbol{X}_t^{\text{UM}})} \quad (13)$$

After the sequence is fully unmasked, the final reward is added to the total log-RND of the trajectory $r(\boldsymbol{X}_T^i)$ and the buffer is updated, such that it contains the top-$B$ sequences $(\boldsymbol{X}^i, W^{\bar{u}})$ with the highest reward with every iteration.

**Backpropagation** For each newly expanded child node $\boldsymbol{X}_s^i$, we initialize the total reward with the reward $R(\boldsymbol{X}_s^i) \leftarrow r(\boldsymbol{X}_T^i)$ and the number of visits to $N_{\text{visits}}(\boldsymbol{X}_s^i) \leftarrow 1$. Then, we sum the terminal rewards of the clean sequence generated at each child node $r(\boldsymbol{X}_T^i)$ and update the total reward of all predecessor nodes.

Compared to standard buffer generation, this method offers the following advantages: **(1) high-reward trajectories sampled are exploited**, and **(2) the log-probabilities of each node in the tree under the pre-trained model are pre-computed** and remain unchanged during selection.

## 5 MULTI-OBJECTIVE FINE-TUNING WITH TR2-D2

While several works have explored multi-objective guidance for test-time scaling of discrete diffusion (Gruver et al., 2023; Tang et al., 2025), **multi-objective fine-tuning** of discrete diffusion models remains largely unexplored. Since the Pareto optimal distribution is not known before training, multi-objective fine-tuning requires a framework that efficiently moves toward the Pareto optimal distribution *during* the fine-tuning process without sacrificing performance in any one objective. Here, we extend our approach from Sec 4 to multiple reward functions.

**Pareto Optimization**    When optimizing a multi-objective reward function $\boldsymbol{r} = (r_1, \ldots, r_K) : \mathcal{X} \to \mathbb{R}^K$, their critical points often conflict, resulting in tradeoffs. Rather than a single optimal reward value, there exists a **Pareto frontier** denoted $\mathcal{P}^\star$ of *reward vectors*.

> **Definition 5.1** (Pareto Frontier of Rewards). *Given a feasible solution space $\mathcal{X}$ and a set of $K$ rewards $\boldsymbol{r} = (r_k)_{k=1}^K$, the Pareto frontier is the set $\mathcal{P}^\star$ defined as*
>
> $$\mathcal{P}^\star = \left\{ \boldsymbol{r}(\boldsymbol{X}_T^i) \mid \boldsymbol{X}_T^i \in \mathcal{X}, \nexists \boldsymbol{X}_T^j \in \mathcal{X} \text{ s.t. } \left(\forall k : r_k^j \geq r_k^i\right) \wedge \left(\exists k : r_k^j > r_k^i\right) \right\} \tag{14}$$
>
> *where each reward $\boldsymbol{r}(\boldsymbol{X}_T^i) \in \mathcal{P}^\star$ is non-dominated, such that no other reward in the set is better than or equal to it across all objectives and strictly better in at least one objective.*

In practice, multi-objective optimization algorithms typically search for a finite approximation of the Pareto-frontier $\mathcal{P}$ by sufficiently exploring the solution space $\mathcal{X}$ and inserting items into $\mathcal{P}$ if it is non-dominated by any existing solution in $\mathcal{P}$.

**Multi-Objective Selection**    During the selection process, rather than selecting with a scalar selection score, we compute a vector of $K$ reward values for each objective $\boldsymbol{U}(\boldsymbol{X}_t, \boldsymbol{X}_s^i) \in \mathbb{R}^K$ where the scalar reward in the first term of (9) is replaced with a reward vector $\boldsymbol{R}(\boldsymbol{X}_s^i) \in \mathbb{R}^K$ that measures the estimated *future reward* of the selection step $\boldsymbol{X}_t \to \boldsymbol{X}_s^i$.

$$\mathcal{P}_{\text{select}}^\star = \left\{ \boldsymbol{X}_s^i \mid \nexists \boldsymbol{X}_s^j \in \texttt{children}(\boldsymbol{X}_t) \text{ s.t. } \boldsymbol{U}(\boldsymbol{X}_t, \boldsymbol{X}_s^j) \succ \boldsymbol{U}(\boldsymbol{X}_t, \boldsymbol{X}_s^j) \right\} \tag{15}$$

where $\succ$ indicates strictly better values across all objectives (Pareto dominance).

**Generating a Buffer of Pareto-Optimal Sequences**    To decide when to update the buffer with a newly generated trajectory $(\boldsymbol{X}_T, W^{\bar{u}})$, we consider the Pareto-optimality of its reward vector $\boldsymbol{r}(\boldsymbol{X}_T) \in \mathbb{R}^K$. At each iteration of MCTS, we compare the $M$ rolled out sequences $\{\boldsymbol{X}_T^i\}_{i=1}^M$ with the current buffer $\mathcal{B}$, and add it to the buffer if it is non-dominated by the sequences in the buffer.

$$\mathcal{B} \leftarrow \mathcal{B} \cup \left\{ \boldsymbol{X}_T^i \mid \nexists \tilde{\boldsymbol{X}}_T \in \mathcal{B} \text{ s.t. } \forall k, r_k(\tilde{\boldsymbol{X}}_T) \geq r_k(\boldsymbol{X}_T^i) \wedge \exists k, r_k(\tilde{\boldsymbol{X}}_T) > r_k(\boldsymbol{X}_T^i) \right\} \tag{16}$$

While the true Pareto frontier is intractable in practice, it holds that each iteration of the tree search moves the buffer set closer to the Pareto-optimal set.

> **Proposition 5.1** (Pareto Optimization of Buffer). *With each iteration of the search, the buffer $\mathcal{B}$ approaches the Pareto front $\mathcal{P}^\star$, where the hypervolume generated by the rewards in the set is maximized.*

The proof is provided in Appendix C.4. While this statement holds for any search algorithm that sufficiently explores the solution space and discovers $\varepsilon$-Pareto solutions with non-negative probability with each search iteration, MCTS efficiently balances tradeoffs between objectives by **(1)** exploiting Pareto-optimal sampling paths with rewards stored as vectors without scalarization and **(2)** further exploring Pareto-optimal nodes to ensure all tradeoffs are maximized.

## 6 EXPERIMENTS

We evaluate **TR2-D2** on several diffusion fine-tuning tasks for biological sequences. Specifically, we fine-tune a pre-trained regulatory DNA sequence model to optimize enhancer activity (Sec 6.1) and a peptide SMILES generator for multi-objective fine-tuning (Sec 6.2).

Table 1: **Comparison of TR2-D2 for regulatory DNA generation optimized on enhancer activity.** Metrics were computed for 640 sequences across 3 seeds, with standard deviations reported. Best values are **bolded**. Evaluation metrics are detailed in Appendix D.

| Method | Pred-Activity (median; ↑) | ATAC-Acc (%; ↑) | 3-mer Corr (↑) | App-Log-Lik (median; ↑) |
|---|---|---|---|---|
| Pre-trained | $0.17_{\pm 0.04}$ | $1.5_{\pm 0.2}$ | $-0.061_{\pm 0.034}$ | $-261_{\pm 0.6}$ |
| CG | $3.30_{\pm 0.00}$ | $0.0_{\pm 0.0}$ | $-0.065_{\pm 0.001}$ | $-266_{\pm 0.6}$ |
| SMC | $4.15_{\pm 0.33}$ | $39.9_{\pm 8.7}$ | $0.840_{\pm 0.045}$ | $-259_{\pm 2.5}$ |
| TDS | $4.64_{\pm 0.21}$ | $45.3_{\pm 16.4}$ | $0.848_{\pm 0.008}$ | $-257_{\pm 1.5}$ |
| CFG | $5.04_{\pm 0.06}$ | $92.1_{\pm 0.9}$ | $0.746_{\pm 0.001}$ | $-265_{\pm 0.6}$ |
| DRAKES | $5.61_{\pm 0.07}$ | $92.5_{\pm 0.6}$ | $0.887_{\pm 0.002}$ | $-264_{\pm 0.6}$ |
| SEPO | $7.55_{\pm 0.01}$ | $99.5_{\pm 0.2}$ | $0.500_{\pm 0.004}$ | $-243.8_{\pm 0.5}$ |
| GLID$^2$E | $7.35_{\pm 0.07}$ | $90.6_{\pm 0.3}$ | $0.490_{\pm 0.074}$ | $\mathbf{-239.9_{\pm 1.4}}$ |
| TR2-D2 w/o MCTS ($\alpha = 0.1$) | $6.00_{\pm 0.02}$ | $76.9_{\pm 1.60}$ | $0.910_{\pm 0.004}$ | $-269.9_{\pm 0.05}$ |
| TR2-D2 w/o MCTS ($\alpha = 0.001$) | $9.13_{\pm 0.02}$ | $95.1_{\pm 0.52}$ | $0.054_{\pm 0.005}$ | $-277.1_{\pm 0.20}$ |
| **TR2-D2** ($\alpha = 0.1$) | $6.56_{\pm 0.02}$ | $86.9_{\pm 1.18}$ | $\mathbf{0.925_{\pm 0.002}}$ | $-259.4_{\pm 0.2}$ |
| **TR2-D2** ($\alpha = 0.001$) | $\mathbf{9.74_{\pm 0.01}}$ | $\mathbf{99.9_{\pm 0.01}}$ | $0.548_{\pm 0.001}$ | $-271.8_{\pm 0.1}$ |

## 6.1 Regulatory DNA Sequence Design

**Setup and Baselines**   We fine-tune the pre-trained DNA enhancer MDM trained on ∼700k HepG2 sequences with measured activity and use the reward oracles from Wang et al. (2025). We compare **TR2-D2** against both discrete diffusion guidance and fine-tuning baselines. Guidance baselines include classifier guidance (CG) (Nisonoff et al., 2025), Sequential Monte Carlo with the pre-trained model as proposal (**SMC**) and with classifier guidance as proposal (**TDS**) (Wu et al., 2023), and classifier-free guidance (CFG) (Ho & Salimans, 2022). Fine-tuning baselines include DRAKES (Wang et al., 2025), which applies the Gumbel-Softmax trick for reward gradients, SEPO (Zekri & Boullé, 2025), which uses REINFORCE with importance sampling, and GLID$^2$E (Cao et al., 2025), which imposes a clipped likelihood constraint for gradient-free RL. We evaluate four metrics: (1) median predicted activity (**Pred-Activity**) by the evaluation oracle (Wang et al., 2025), (2) predicted chromatin accessibility (**ATAC-Acc**; %), (3) 3-mer Pearson correlation with the top 0.5% HepG2 sequences (**3-mer Corr**), and (4) log-likelihood under the pre-trained model (**App-Log-Lik**). Further details are provided in App D, with hyperparameters and ablations in App F.

**Results**   We demonstrate that **TR2-D2** with $\alpha = 0.001$ outperforms *all* benchmarks on predicted activity and chromatin accessibility, achieving a median Pred-Activity of $\mathbf{9.78}$ compared to 7.64 of the closest benchmark and a near-perfect ATAC-Acc score of 100% (Table 1). Alongside the high reward, we maintain relatively high 3-mer correlation and log-likelihood, indicating that the generated sequences still resemble natural enhancers. Furthermore, we find that by increasing KL regularization with $\alpha = 0.1$, we can achieve the highest 3-mer correlation to the top 0.1% sequences in the dataset with the highest HepG2 activity, while maintaining higher predicted activity and chromatin accessibility than DRAKES, with the second-highest 3-mer correlation (Table 1).

## 6.2 Multi-Objective Peptide Sequence Design

In this experiment, we aim to fine-tune a peptide MDM to optimize multiple therapeutic properties using the algorithm in Sec 5. Notably, we show that generation with one diffusion pass of the fine-tuned policy outperforms inference-time multi-objective guidance, marking a significant advancement in multi-objective fine-tuning.

**Setup and Baselines**   We fine-tune the pre-trained peptide MDM from Tang et al. (2025), built on the MDLM framework (Sahoo et al., 2024) and trained on 11M peptide SMILES sequences. Multi-objective rewards are defined by the classifiers from Tang et al. (2025) for binding affinity, solubility, non-hemolysis, non-fouling, and membrane permeability. Binding affinity is optimized for multiple therapeutically relevant targets described in App E. We compare the multi-objective rewards of generated sequences from the fine-tuned model against sequences from the unconditional pre-trained model and from inference-time multi-objective guidance with PepTune (Tang et al., 2025). Further experimental details are in App E.

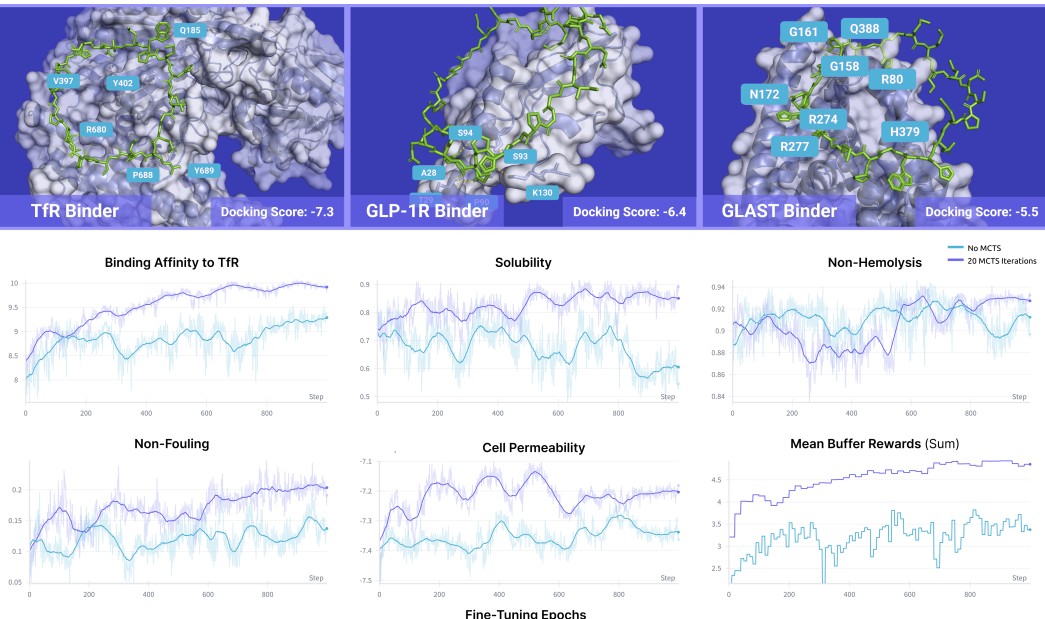

Figure 2: **Peptide docking results and comparison of multi-objective fine-tuning with and without MCTS. (Top)** Docked peptides to TfR, GLP-1R, and GLAST with docking scores (↓) and polar contacts within 3.5 Å annotated. **(Bottom)** Average multi-reward scores for TR2-D2 with no MCTS (**blue**) and with 20 iterations of MCTS at each buffer generation (**purple**). Scores are evaluated on 50 sequences sampled from the fine-tuned model after each fine-tuning epoch and plotted over a total of 1000 epochs, and a running average is shown with the smooth line.

Table 2: **Multi-objective peptide design results (full results in Table 5).** All values are averaged over 100 generated peptides. Best values are **bolded**. **Pre-trained** indicates unconditional sampling with the pre-trained peptide SMILES model from PepTune (Tang et al., 2025). **PepTune** indicates samples from 100 iterations of inference-time Monte-Carlo Tree Guidance conditioned on all objectives. **TR2-D2** indicates unconditional sampling after 1000 epochs of fine-tuning of the pre-trained model with our multi-objective fine-tuning approach.

| Target Protein | Method | Binding Affinity (↑) | Solubility (↑) | Non-hemolysis (↑) | Non-fouling (↑) | Permeability (↑) |
|---|---|---|---|---|---|---|
| TfR | Pre-trained | $8.008_{\pm 0.673}$ | $0.742_{\pm 0.166}$ | $0.874_{\pm 0.063}$ | $0.102_{\pm 0.083}$ | $-7.470_{\pm 0.120}$ |
| | PepTune | $8.216_{\pm 0.703}$ | $0.789_{\pm 0.144}$ | $0.902_{\pm 0.051}$ | $0.121_{\pm 0.081}$ | $-7.389_{\pm 0.119}$ |
| | **TR2-D2** w/o MCTS | $9.336_{\pm 0.325}$ | $0.548_{\pm 0.173}$ | $\mathbf{0.908_{\pm 0.034}}$ | $0.122_{\pm 0.044}$ | $-7.323_{\pm 0.076}$ |
| | **TR2-D2** (Ours) | $\mathbf{10.098_{\pm 0.050}}$ | $\mathbf{0.838_{\pm 0.066}}$ | $0.896_{\pm 0.012}$ | $\mathbf{0.271_{\pm 0.038}}$ | $\mathbf{-7.168_{\pm 0.024}}$ |
| GLP-1R | Pre-trained | $8.233_{\pm 0.367}$ | $0.742_{\pm 0.166}$ | $0.874_{\pm 0.063}$ | $0.102_{\pm 0.083}$ | $-7.470_{\pm 0.120}$ |
| | PepTune | $8.403_{\pm 0.365}$ | $0.774_{\pm 0.170}$ | $\mathbf{0.907_{\pm 0.057}}$ | $0.125_{\pm 0.082}$ | $-7.388_{\pm 0.128}$ |
| | **TR2-D2** (Ours) | $\mathbf{9.426_{\pm 0.035}}$ | $\mathbf{0.841_{\pm 0.043}}$ | $0.849_{\pm 0.016}$ | $\mathbf{0.499_{\pm 0.037}}$ | $\mathbf{-7.263_{\pm 0.020}}$ |
| GLAST | Pre-trained | $7.830_{\pm 0.420}$ | $0.742_{\pm 0.166}$ | $0.874_{\pm 0.063}$ | $0.102_{\pm 0.083}$ | $-7.470_{\pm 0.120}$ |
| | PepTune | $8.400_{\pm 0.353}$ | $0.815_{\pm 0.139}$ | $\mathbf{0.937_{\pm 0.029}}$ | $0.137_{\pm 0.086}$ | $-7.311_{\pm 0.106}$ |
| | **TR2-D2** (Ours) | $\mathbf{9.703_{\pm 0.072}}$ | $\mathbf{0.884_{\pm 0.038}}$ | $0.930_{\pm 0.007}$ | $\mathbf{0.364_{\pm 0.083}}$ | $\mathbf{-7.238_{\pm 0.020}}$ |

**Results** Compared to inference-time multi-objective guidance with PepTune (Tang et al., 2025), **TR2-D2** consistently yields higher scores across nearly *all properties* for each protein target, requiring only a single diffusion pass (Table 2 and 5). Furthermore, we demonstrate that using tree search in the buffer generation step significantly enhances performance across multiple rewards over fine-tuning iterations compared to optimizing the scalarized reward with just the off-policy fine-tuning strategy (Fig 2 and 4; Table 8). Finally, we highlight that **TR2-D2** outperforms PepTune with only 200 epochs of fine-tuning, while minimizing the trade-off between achieving reward optimality and the diversity in the generated sequences observed with increasing fine-tuning iterations (Table 5). We include a detailed discussion of hyperparameters and ablations in App F.

## 7 CONCLUSION

In this work, we introduce **TR**ee Search Guided **TR**ajectory-Aware Fine-Tuning for **D**iscrete **D**iffusion (**TR2-D2**), a general framework for enhancing the efficiency and reliability of RL with structured search. We apply this framework for discrete diffusion fine-tuning by curating a buffer of optimized sequences with MCTS for off-policy RL, demonstrating success in single and multi-objective fine-tuning. Looking ahead, TR2-D2 can be applied to broader classes of biological sequences, such as full-length proteins and mRNA (Wang et al., 2024; Peng et al., 2025; Vincoff et al., 2025; Patel et al., 2025), where optimizing for multiple structural and functional constraints is essential. The framework also lends itself naturally to integration with high-throughput wet-lab pipelines (Zhao et al., 2024; 2025a;b), where experimentally validated feedback can be incorporated into the replay buffer to accelerate closed-loop sequence discovery. From a theoretical perspective, TR2-D2 opens new directions for studying discrete stochastic optimal control, variance-reduction strategies for trajectory weighting, and multi-objective optimization under Pareto efficiency, offering deeper insight into how reinforcement learning and search interact with discrete diffusion processes. Together, these directions highlight TR2-D2 as both a practical platform for therapeutic design and a step toward a broader theory of reward-guided discrete generative modeling.

## REPRODUCIBILITY STATEMENT

We have made significant efforts to ensure the reproducibility of our work. Complete experimental details are provided in Appendices D and E, including dataset descriptions, model architectures, training procedures, and evaluation metrics. All hyperparameters used in our experiments are documented in Table 6 with detailed discussion and ablation studies in Appendix F. The complete algorithmic implementation is provided as pseudocode in Appendix G, including the main TR2-D2 algorithm (Algorithm 2), MCTS implementation (Algorithm 5), and all supporting functions. For the regulatory DNA experiments, we use pre-trained models from prior work, with clear references to enable replication. The peptide experiments use the pre-trained weights from the PepTune framework (Tang et al., 2025), which is provided in our codebase. We provide details on hardware specifications to facilitate the reproduction of our results. Code is provided in the anonymous repository: https://anonymous.4open.science/r/TR2-D2anon.

## ETHICS STATEMENT

This work adheres to the ICLR Code of Ethics. Our research focuses on computational methods for biological sequence design, which has potential applications in drug discovery and biotechnology. We acknowledge that biological sequence design technologies could have dual-use implications. The peptide design application demonstrated in our work targets therapeutic proteins for treating diseases such as diabetes and rare genetic disorders, representing beneficial applications. However, we recognize that sequence design methods could potentially be misused for harmful purposes. We have designed our experiments to focus on clearly beneficial applications and have not explored potentially harmful uses. All datasets used in this work are publicly available and have been previously published with appropriate ethical considerations. No human subjects were involved in this research. The computational methods developed here are general frameworks that require domain expertise and appropriate oversight for real-world applications. We encourage responsible use of these methods and recommend that practitioners consider ethical implications and potential risks when applying sequence design technologies in practice.

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

OVERVIEW OF APPENDIX

In App A, we present a detailed discussion of related works in diffusion fine-tuning, discrete diffusion, inference-time scaling of diffusion models, and multi-objective optimization. In App B, we provide the theoretical foundation of our work. In App C, we present the theoretical proofs and justifications for Sec 4. The experiment details for enhancer DNA generation are given in App D and experiment details for multi-objective peptide generation are given in App E. We include a discussion on hyperparameters and present ablation results for the enhancer DNA and peptide experiments in App F. Finally, the pseudo-code for our algorithms are given in App G.

**Notation** We denote the discrete state space of sequences of length $L$ with a vocabulary of size $D$ as $\mathcal{X} \in \{1, \ldots, D\}^L$ where a probability distribution for a single token is on the $(D-1)$-dimensional simplex $\Delta^{D-1}$. We denote a path measure as $\mathbb{P}$ with the pre-trained path measure as $\mathbb{P}^{\text{pre}}$, the path measure produced by the fine-tuned policy as $\boldsymbol{Q}^u$ as $\mathbb{P}^u$, and the path measure produced by the optimal generator $\boldsymbol{Q}^\star$ as $\mathbb{P}^\star$. We denote a sequence at time $t$ in the diffusion process as $\boldsymbol{X}_t \in \mathcal{X}$ and the following step at time $s = t + \Delta t$ as $\boldsymbol{X}_s$ and a trajectory of as $\boldsymbol{X}_{0:T} := (\boldsymbol{X}_t)_{t \in [0,T]}$. We index each token in the sequence with $\ell \in \{1, \ldots, L\}$ and denote an update of the masked state with the token at position $\ell$ as $\boldsymbol{X}_s^\ell = \boldsymbol{x}_s^\ell$. We further consider a tree of unmasking trajectories denoted $\mathcal{T}$, where each node is defined as a partially masked sequence. Given a node $\boldsymbol{X}_t$ in the tree, we denote the $M$ unmasking steps derived from $\boldsymbol{X}_t$ as $\{\boldsymbol{X}_s^i\}_{i=1}^M$. We denote the conditional probability distribution of the token $\ell$ given the unmasked tokens $\boldsymbol{X}_t^{\text{UM}}$ as $p^{\text{pre}}(\tilde{\boldsymbol{x}})_{\ell, \boldsymbol{X}_T^\ell} \in \Delta^{D-1}$ under the pre-trained model and $p^{u_\theta}(\tilde{\boldsymbol{x}})_{\ell, \boldsymbol{X}_T^\ell} \in \Delta^{D-1}$ under the current policy.

# A RELATED WORKS

**Fine-Tuning Diffusion Models with Reinforcement Learning** RL fine-tuning of diffusion has been used to train the model to generate data samples that optimize a reward function (Black et al., 2023; Wallace et al., 2024; Domingo-Enrich et al., 2024; Uehara et al., 2024a; Fan et al., 2023a; Clark et al., 2023; Blessing et al., 2025). Specifically, fine-tuning has been widely explored for text-to-image generation (Lu et al., 2023; Ruiz et al., 2023; Gupta et al., 2025; Yuan et al., 2024a; Fan et al., 2023b; Liu et al., 2025; Zheng et al., 2025b) and biomolecular sequence design (Wang et al., 2025; Zekri & Boullé, 2025; Cao et al., 2025). The fine-tuning problem has commonly been framed as an entropy-regularized control problem (Uehara et al., 2024b; Han et al., 2024b; Tang, 2024; Zhu et al., 2025c), which seeks to find an optimal sampling trajectory that maximizes some terminal reward. Fine-tuning methods have also been developed specifically for discrete diffusion, with approaches that optimize differentiable rewards (Wang et al., 2025), non-differentiable rewards (Zekri & Boullé, 2025; Cao et al., 2025; Su et al., 2025; Zhu et al., 2025c), and those tailored to diffusion language models (Zhao et al., 2025c; Gong et al., 2025).

**Discrete Diffusion Models** Diffusion models have achieved state-of-the-art performance on generating various data modalities (Zhu et al., 2025b; Esser et al., 2024; Zhu et al., 2025a; Rojas et al., 2025b; Zheng et al., 2025c). Discrete diffusion models (Austin et al., 2021; Campbell et al., 2022; Lou et al., 2023), as a natural generalization of diffusion models to finite state space, have emerged as powerful generative frameworks for sequence data, among which the most effective variant is Masked discrete diffusion models (MDM) (Sahoo et al., 2024; Wang et al., 2024; Shi et al., 2024; Peng et al., 2025; Tang et al., 2025; Nisonoff et al., 2025; Rector-Brooks et al., 2025; Bai et al., 2025; Shi et al., 2025). These models operate by progressively denoising masked inputs, enabling them to capture long-range dependencies without relying on autoregressive factorization. Within biology, masked discrete diffusion models have been successfully applied to peptide (Tang et al., 2025; Vincoff et al., 2025), protein (Wang et al., 2024; Goel et al., 2025; Nisonoff et al., 2025; Rector-Brooks et al., 2025; Wang et al., 2025), and nucleic acid design (Wang et al., 2025; Patel et al., 2025). Furthermore, recent extensions have introduced blockwise discrete diffusion architectures that interpolate between autoregressive and diffusion models to improve training efficiency and sequence length generalization (Arriola et al., 2025), as well as simplified formulations of masked diffusion that provide tighter likelihood bounds and more effective training objectives (Schiff et al., 2025).

**Inference-Time Scaling of Diffusion Models**   Inference-time scaling of diffusion models aims to efficiently leverage additional compute during sampling to improve output quality and controllability. One line of work steers continuous diffusion processes using Feynman–Kac guidance, which is theoretically guaranteed to sample from a reward-tilted distribution by reweighting trajectories at each denoising step (Skreta et al., 2025; Singhal et al., 2025; Chen et al., 2025a). Search-based approaches apply combinatorial optimization over diffusion trajectories to identify high-reward sequences (Sun et al., 2022), while reward-gradient methods adapt score-function estimators to steer sampling (Song et al., 2020b; Bansal et al., 2023). Importance sampling techniques can also be used to bias toward rare high-reward generations, but require large sample sizes to ensure coverage (Chatterjee & Diaconis, 2018). Soft value-based decoding has been proposed as a derivative-free approach for steering both continuous and discrete diffusion processes (Li et al., 2024b). More recently, classical search methods have been incorporated into continuous diffusion sampling as a scaling technique during inference time Jain et al. (2025); Zhang et al. (2025a).

Classifier-based and classifier-free guidance methods have been adapted from continuous diffusion into the discrete domain (Nisonoff et al., 2025; Rector-Brooks et al., 2025; Wang et al., 2024; Schiff et al., 2025; Rojas et al., 2025a; Guo et al., 2024). Recent strategies for post-hoc optimization include classifier-free guidance (CFG) (Ho & Salimans, 2022), LaMBO-2 and NOS guidance (Gruver et al., 2023), and MCTS-guided sampling as in PepTune (Tang et al., 2025) and SOAPIA (Vincoff et al., 2025), which adapt pretrained models to specific objectives strictly at inference time.

**Multi-Objective Optimization**   Optimizing multiple, potentially conflicting, reward and constraint functions while balancing tradeoffs has significant applications across engineering and biology applications (Marler & Arora, 2004; Jain et al., 2017; Tagasovska et al., 2022; Zhu et al., 2023; Janson et al., 2008). For molecular drug design, the objectives include affinity to the drug target, bioavailability, potency, solubility for efficient drug loading, non-toxicity, synthesizability, among others (Nicolaou et al., 2007; Fromer & Coley, 2023; Sun et al., 2022; Winter et al., 2019; Jin et al., 2020; Xie et al., 2021). Due to tradeoffs between objectives, there often does not exist a single solution that dominates across all objectives, but rather a set of optimal solutions where no objective can be improved without sacrificing another objective (Censor, 1977). To reduce the multi-objective problem into a more tractable single-objective problem, hypervolume (HV) has been used to quantify the optimality of a solution with respect to a reference point (Yang et al., 2019; Daulton et al., 2020; Ament et al., 2023; Daulton et al., 2022; Konakovic Lukovic et al., 2020). To sample from the Pareto-frontier, several approaches have been proposed, including active learning (Zuluaga et al., 2016; Belakaria et al., 2020), entropy-based multi-objective Bayesian optimization (Wang & Jegelka, 2017; Suzuki et al., 2020; Hernández-Lobato et al., 2016; Fernández-Sánchez et al., 2020), cumulative distribution function optimization (Park et al., 2023), and constrained multi-objective optimization (Gelbart et al., 2014; Li et al., 2024a). More recently, multi-objective guidance frameworks have been used to steer generative models like LLM (Ren et al., 2024b;a), diffusion (Gruver et al., 2023; Yao et al., 2024; Han et al., 2023; Yuan et al., 2024c; Annadani et al., 2025; Zhang et al., 2025b), discrete diffusion (Tang et al., 2025), and flow matching (Jain et al., 2023; Yuan et al., 2024b; Chen et al., 2025b) toward optimizing multiple objectives.

## B   EXTENDED THEORETICAL BACKGROUND

In this section, we provide relevant theoretical backgrounds that connect the RL fine-tuning of discrete diffusion models with the stochastic optimal control of CTMCs. Some relevant results are first proved in Uehara et al. (2024b); Wang et al. (2025); Zhu et al. (2025c); we include the proof there for self-consistency and a more coherent reading experience.

### B.1   CONTINUOUS-TIME MARKOV CHAINS (CTMCS)

Here, we derive the RND of two CTMCs, which will be used to define our fine-tuning objective. Throughout the theoretical proofs, we will use subscript $t-$ to be the instantaneous timestep following a discrete jump of the CTMC at time $t$, $\mathbb{P}^0$ and $Q^0$ to denote the reference path measure and corresponding generator, which are the same as the path measure $\mathbb{P}^{pre}$ and generator $Q^{pre}$ of the pre-trained diffusion model in the case of fine-tuning.

**Lemma 1** (Kolmogorov Forward Equation). *The forward-time dynamics of the probability distribution $p_t(\cdot) = Pr(\boldsymbol{X}_t = \cdot)$ of a CTMC $\boldsymbol{X}_{0:T}$ with generator $\boldsymbol{Q}_t$ satisfies the Kolmogorov forward equation:*

$$\forall x, \ \ \partial_t p_t(x) = \sum_y \boldsymbol{Q}_t(y, x) p_t(y) = \sum_{y \neq x} (\boldsymbol{Q}_t(y, x) p_t(y) - \boldsymbol{Q}_t(x, y) p_t(x)) \qquad (17)$$

*where given an endpoint condition at $t \in \{0\}$, the solution is a unique proabbility measure $p$ given that $\boldsymbol{Q}_t$ is continuous over time $t \in [0, T]$.*

*Proof.* We prove this by taking the conditional probability for a forward step $[t, t + \Delta t]$ and taking the limit as $\Delta t \to 0$. From (21), we have

$$p_{t+\Delta t}(x) = \sum_y \Pr(\boldsymbol{X}_{t+\Delta t} = x | \boldsymbol{X}_t = y) p_t(y)$$

$$= \sum_y (\mathbf{1}_{x=y} + \Delta t \boldsymbol{Q}_t(y, x) + \mathcal{O}(\Delta t^2)) p_t(y)$$

$$= p_t(x) + \Delta t \sum_y \boldsymbol{Q}_t(y, x) p_t(y) + \mathcal{O}(\Delta t^2) \qquad (18)$$

Taking the limit as $\Delta t \to 0$, we have

$$\partial_t p_t(x) = \lim_{\Delta t \to 0} \left[ \Delta t \sum_y \boldsymbol{Q}_t(y, x) p_t(y) + \mathcal{O}(\Delta t^2) \right]$$

$$= \sum_y \boldsymbol{Q}_t(y, x) p_t(y)$$

$$= \sum_{y \neq x} \boldsymbol{Q}_t(y, x) p_t(y) + \boldsymbol{Q}_t(x, x) p_t(x)$$

$$= \sum_{y \neq x} \boldsymbol{Q}_t(y, x) p_t(y) - \sum_{y \neq x} \boldsymbol{Q}_t(x, y) p_t(x) \qquad (19)$$

which concludes our proof. $\qquad \square$

**Lemma 2** (Radon-Nikodym Derivative (RND)). *Consider two CTMCs $\boldsymbol{Q}$ and $\boldsymbol{Q}'$ with path measures $\mathbb{P}$ and $\mathbb{P}'$ and initial distributions $\pi_0$ and $\pi_0'$. Then, the Radon-Nikodym derivative over a trajectory $\boldsymbol{X}_{0:T} = (\boldsymbol{X}_t)_{t \in [0, T]}$ is defined as*

$$\log \frac{\mathrm{d}\mathbb{P}'}{\mathrm{d}\mathbb{P}}(\boldsymbol{X}_{0:T}) = \log \frac{\mathrm{d}\pi_0'}{\mathrm{d}\pi_0}(\boldsymbol{X}_0) + \sum_{t: \boldsymbol{X}_{t-} \neq \boldsymbol{X}_t} \log \frac{\boldsymbol{Q}_t'(\boldsymbol{X}_{t-}, \boldsymbol{X}_t)}{\boldsymbol{Q}_t(\boldsymbol{X}_{t-}, \boldsymbol{X}_t)} + \int_0^T \sum_{y \neq \boldsymbol{X}_t} (\boldsymbol{Q}_t - \boldsymbol{Q}_t')(\boldsymbol{X}_t, y) \mathrm{d}t$$

*Proof.* First, we compute the RND in the discrete-time case, where $\Delta t = \frac{T}{N}$ is the discrete time interval and $t_n = n\Delta t$ is the time at the $n$th step. The RND of the discretized path can be written as

$$\log \frac{\mathrm{d}\mathbb{P}'}{\mathrm{d}\mathbb{P}}(\boldsymbol{X}_{0:T}) = \log \frac{\mathrm{d}\pi_0'}{\mathrm{d}\pi_0}(\boldsymbol{X}_0) + \sum_{n=0}^{N-1} \log \frac{\mathbb{P}'(\boldsymbol{X}_{t_{n+1}} | \boldsymbol{X}_{t_n})}{\mathbb{P}(\boldsymbol{X}_{t_{n+1}} | \boldsymbol{X}_{t_n})} + \mathcal{O}(\Delta t) \qquad (20)$$

where $\mathcal{O}(\Delta t)$ accounts for the probability of multiple jumps within the time interval. The probability of a single jump under a CTMC $\mathbb{P}$ can be decomposed into the probability of remaining in the same state and the probability of transitioning to a different state $y$ at time $t_n$.

$$\mathbb{P}(\boldsymbol{X}_{t_{n+1}} = y | \boldsymbol{X}_{t_n} = x) = \begin{cases} 1 - \Delta t \sum_{z \neq x} \boldsymbol{Q}_{t_n}(x, z) + \mathcal{O}(\Delta t^2) & y = x \\ \Delta t \boldsymbol{Q}_{t_n}(x, y) + \mathcal{O}(\Delta t^2) & y \neq x \end{cases} \qquad (21)$$

First, expanding the log-ratio for the case where a jump is made in the interval $[t_n, t_{n+1}]$, we have

$$\log \frac{\mathbb{P}'(\boldsymbol{X}_{t_{n+1}}|\boldsymbol{X}_{t_n})}{\mathbb{P}(\boldsymbol{X}_{t_{n+1}}|\boldsymbol{X}_{t_n})} = \log \frac{\Delta t \boldsymbol{Q}'_{t_n}(\boldsymbol{X}_{t_n}, \boldsymbol{X}_{t_{n+1}}) + \mathcal{O}(\Delta t^2)}{\Delta t \boldsymbol{Q}_{t_n}(\boldsymbol{X}_{t_n}, \boldsymbol{X}_{t_{n+1}}) + \mathcal{O}(\Delta t^2)}$$

$$= \log \frac{\boldsymbol{Q}'_{t_n}(\boldsymbol{X}_{t_n}, \boldsymbol{X}_{t_{n+1}})}{\boldsymbol{Q}_{t_n}(\boldsymbol{X}_{t_n}, \boldsymbol{X}_{t_{n+1}})} + \mathcal{O}(\Delta t) \tag{22}$$

Next, expanding the log-ratio for the case where no jump is made in the interval $[t_n, t_{n+1}]$, we use the Taylor expansion of $\log(1 - x) = w + \mathcal{O}(w^2)$ to get

$$\log \frac{\mathbb{P}'(\boldsymbol{X}_{t_{n+1}}|\boldsymbol{X}_{t_n})}{\mathbb{P}(\boldsymbol{X}_{t_{n+1}}|\boldsymbol{X}_{t_n})} = \log \frac{1 - \Delta t \sum_{z \neq x} \boldsymbol{Q}'_{t_n}(\boldsymbol{X}_{t_n}, z) + \mathcal{O}(\Delta t^2)}{1 - \Delta t \sum_{z \neq x} \boldsymbol{Q}_{t_n}(\boldsymbol{X}_{t_n}, z) + \mathcal{O}(\Delta t^2)}$$

$$= \Delta t \sum_{z \neq \boldsymbol{X}_{t_n}} (\boldsymbol{Q}_{t_n}(\boldsymbol{X}_{t_n}, z) - \boldsymbol{Q}'_{t_n}(\boldsymbol{X}_{t_n}, z)) + \mathcal{O}(\Delta t^2) \tag{23}$$

Finally, putting it all together and taking the limit as $N \to \infty$ and $\Delta t \to 0$, we have

$$\log \frac{\mathrm{d}\mathbb{P}'}{\mathrm{d}\mathbb{P}}(\boldsymbol{X}_{0:T}) = \lim_{\Delta t \to 0} \left\{ \log \frac{\mathrm{d}\pi'_0}{\mathrm{d}\pi_0}(\boldsymbol{X}_0) + \sum_{n=0}^{N-1} \log \frac{\boldsymbol{Q}'_{t_n}(\boldsymbol{X}_{t_n}, \boldsymbol{X}_{t_{n+1}})}{\boldsymbol{Q}_{t_n}(\boldsymbol{X}_{t_n}, \boldsymbol{X}_{t_{n+1}})} \right.$$

$$\left. + \Delta t \sum_{z \neq x} (\boldsymbol{Q}_{t_n}(\boldsymbol{X}_{t_n}, z) - \boldsymbol{Q}'_{t_n}(\boldsymbol{X}_{t_n}, z)) + \mathcal{O}(\Delta t) \right\}$$

$$= \log \frac{\mathrm{d}\pi'_0}{\mathrm{d}\pi_0}(\boldsymbol{X}_0) + \sum_{t: \boldsymbol{X}_s \neq \boldsymbol{X}_t} \log \frac{\boldsymbol{Q}'_t(\boldsymbol{X}_s, \boldsymbol{X}_t)}{\boldsymbol{Q}_t(\boldsymbol{X}_s, \boldsymbol{X}_t)} + \int_0^T \sum_{z \neq \boldsymbol{X}_t} (\boldsymbol{Q}_t(\boldsymbol{X}_t, z) - \boldsymbol{Q}'_t(\boldsymbol{X}_t, z)) \mathrm{d}t \tag{24}$$

which concludes the proof. □

Now, we can easily extend this result to derive the KL-divergence $D_{\mathrm{KL}}(\mathbb{P}' \| \mathbb{P})$ by taking the expectation with respect to $\mathbb{P}'$ on either side of the equality.

**Corollary 1.** *The KL-divergence between two CTMCs $\mathbb{P}', \mathbb{P}$ with generators $\boldsymbol{Q}', \boldsymbol{Q}$*

$$D_{KL}(\mathbb{P}' \| \mathbb{P}) = D_{KL}(\pi'_0 \| \pi_0) + \mathbb{E}_{\boldsymbol{X}_{0:T} \sim \mathbb{P}'} \int_0^T \sum_{y \neq \boldsymbol{X}_t} \boldsymbol{Q}'_t \log \frac{\boldsymbol{Q}'_t}{\boldsymbol{Q}_t}(\boldsymbol{X}_t, y) \mathrm{d}t \tag{25}$$

*Proof.* For the first term on the RHS of (24), we have

$$\mathbb{E}_{\boldsymbol{X}_{0:T} \sim \mathbb{P}'} \left[ \log \frac{\mathrm{d}\pi'_0}{\mathrm{d}\pi_0}(\boldsymbol{X}_0) \right] = \mathbb{E}_{\boldsymbol{X}_0 \sim \pi'_0} \left[ \log \frac{\mathrm{d}\pi'_0}{\mathrm{d}\pi_0}(\boldsymbol{X}_0) \right] = D_{\mathrm{KL}}(\pi'_0 \| \pi_0) \tag{26}$$

For the second term, we apply the expectation to the discrete-time case and take the limit as $\Delta t \to 0$ given by

$$\mathbb{E}_{\boldsymbol{X}_{0:T} \sim \mathbb{P}'} \left[ \sum_{n=0}^{N-1} \mathbf{1}_{\boldsymbol{X}_{t_{n+1}} \neq \boldsymbol{X}_{t_n}} \log \frac{\boldsymbol{Q}'_{t_n}(\boldsymbol{X}_{t_n}, \boldsymbol{X}_{t_{n+1}})}{\boldsymbol{Q}_{t_n}(\boldsymbol{X}_{t_n}, \boldsymbol{X}_{t_{n+1}})} \right]$$

$$= \sum_{n=0}^{N-1} \mathbb{E}_{\mathbb{P}'(\boldsymbol{X}_{t_n}), \mathbb{P}'(\boldsymbol{X}_{t_{n+1}}|\boldsymbol{X}_{t_n})} \left[ \mathbf{1}_{\boldsymbol{X}_{t_{n+1}} \neq \boldsymbol{X}_{t_n}} \log \frac{\boldsymbol{Q}'_{t_n}(\boldsymbol{X}_{t_n}, \boldsymbol{X}_{t_{n+1}})}{\boldsymbol{Q}_{t_n}(\boldsymbol{X}_{t_n}, \boldsymbol{X}_{t_{n+1}})} \right]$$

$$= \sum_{n=0}^{N-1} \mathbb{E}_{\mathbb{P}'(\boldsymbol{X}_{t_n})} \sum_{y \neq \boldsymbol{X}_{t_n}} \mathbb{P}'(y|\boldsymbol{X}_{t_n}) \log \frac{\boldsymbol{Q}'_{t_n}(\boldsymbol{X}_{t_n}, y)}{\boldsymbol{Q}_{t_n}(\boldsymbol{X}_{t_n}, y)}$$

$$= \sum_{n=0}^{N-1} \mathbb{E}_{\mathbb{P}'(\boldsymbol{X}_{t_n})} \sum_{y \neq \boldsymbol{X}_{t_n}} \left[ \Delta t \boldsymbol{Q}'_{t_n}(\boldsymbol{X}_{t_n}, y) \log \frac{\boldsymbol{Q}'_{t_n}(\boldsymbol{X}_{t_n}, y)}{\boldsymbol{Q}_{t_n}(\boldsymbol{X}_{t_n}, y)} + \mathcal{O}(\Delta t^2) \right]$$

$$\underset{\Delta t \to 0}{=} \mathbb{E}_{\boldsymbol{X}_{0:T} \sim \mathbb{P}'} \int_0^T \sum_{y \neq \boldsymbol{X}_t} \boldsymbol{Q}'_t \log \frac{\boldsymbol{Q}'_t}{\boldsymbol{Q}_t}(\boldsymbol{X}_t, y) \mathrm{d}t \tag{27}$$

which concludes the proof. □

## B.2 Entropy-Regularized Diffusion Fine-Tuning

The standard entropy-regularized diffusion fine-tuning problem (Black et al., 2023; Fan et al., 2023a; Clark et al., 2023; Uehara et al., 2025) involves a maximization objective with two terms: **(1)** a reward function and **(2)** a KL regularization term that ensures the fine-tuned model does not diverge significantly from the pre-trained model. Formally, a parameterized policy $u_\theta$ that generates a diffusion path distribution $p^{u_\theta}$ aims to minimize the following objective

$$\arg\min_\theta \left\{ D_{\text{KL}} \left( p^{u_\theta}(\boldsymbol{X}_{0:T}) \| p^{\text{pre}}(\boldsymbol{X}_{0:T}) \right) - \mathbb{E}_{\boldsymbol{X}_{0:T} \sim \mathbb{P}^{u_\theta}} \left[ \frac{r(\boldsymbol{X}_T)}{\alpha} \right] \right\} \tag{28}$$

The first term maximizes the expected terminal reward under the policy model $u_\theta$ and the second term minimizes the KL divergence between the path measure under the policy model $\mathbb{P}^{u_\theta}$ and the pre-trained model $\mathbb{P}^{\text{pre}}$. The scalar $\alpha > 0$ is a regularization factor that determines how closely the policy model follows the pre-trained model, where a smaller $\alpha$ allows greater divergence from the pre-trained model and a larger $\alpha$ constrains the policy model to follow closer to the pre-trained model.

In discrete diffusion, the KL divergence term can be written in terms of the CTMC generators of the pre-trained model $\boldsymbol{Q}^{\text{pre}}$ and the policy model $\boldsymbol{Q}^{u_\theta}$ given by

$$D_{\text{KL}} \left( \mathbb{P}^{u_\theta} \| \mathbb{P}^{\text{pre}} \right) = \mathbb{E}_{\boldsymbol{X}_{0:T} \sim \mathbb{P}^{u_\theta}} \left[ \int_0^T \sum_{y \neq \boldsymbol{X}_t} \left( \boldsymbol{Q}_t^{u_\theta} \log \frac{\boldsymbol{Q}_t^{u_\theta}}{\boldsymbol{Q}_t^{\text{pre}}} - \boldsymbol{Q}_t^{u_\theta} + \boldsymbol{Q}_t^{\text{pre}} \right) (\boldsymbol{X}_t, y) \mathrm{d}t \right] \tag{29}$$

Then, the discrete diffusion fine-tuning objective can be written as

$$\arg\min_\theta \left\{ \mathbb{E}_{\boldsymbol{X}_{0:T} \sim \mathbb{P}^{u_\theta}} \left[ \int_0^T \sum_{y \neq \boldsymbol{X}_t} \left( \boldsymbol{Q}_t^{u_\theta} \log \frac{\boldsymbol{Q}_t^{u_\theta}}{\boldsymbol{Q}_t^{\text{pre}}} - \boldsymbol{Q}_t^{u_\theta} + \boldsymbol{Q}_t^{\text{pre}} \right) (\boldsymbol{X}_t, y) \mathrm{d}t - \frac{r(\boldsymbol{X}_T)}{\alpha} \right] \right\} \tag{30}$$

In the next section, we describe a method of minimizing this objective with stochastic optimal control theory to derive an **off-policy** objective that avoids taking the expectation with respect to the current policy $\mathbb{P}^{u_\theta}$.

## B.3 Fine-Tuning with Stochastic Optimal Control

Here, we will frame the entropy-regularized diffusion fine-tuning framework defined in App B.2 as a stochastic optimal control (SOC) problem that aims to find the optimal generator $\boldsymbol{Q}^\star$ that produces the optimal *reward-tilted* path measure $\mathbb{P}^\star$.

First, we define the **value function** $V_t(\boldsymbol{x})$ which gives the *cost-to-go* from a state $\boldsymbol{x}$ to a final state $\boldsymbol{X}_T$ under a controlled path measure $\mathbb{P}^u$. We define the cost minimization objective with terminal reward $r(\boldsymbol{X}_T)$ as

$$J_t(\boldsymbol{x}, u) = \mathbb{E}_{\boldsymbol{X} \sim \mathbb{P}^u} \left[ \int_t^T \sum_{y \neq \boldsymbol{X}_s} C(\boldsymbol{X}_s, y) ds - r(\boldsymbol{X}_T) \Big| \boldsymbol{X}_t = \boldsymbol{x} \right] \tag{31}$$

where the cost is defined as $C_t(x, y) = \left( \boldsymbol{Q}_t^u \log \frac{\boldsymbol{Q}^u}{\boldsymbol{Q}^0} - \boldsymbol{Q}^u + \boldsymbol{Q}^0 \right)(x, y)$, and the optimal cost-to-go is $J_t^\star(\boldsymbol{x}, u) = \inf_u J_t(\boldsymbol{x}, u)$. Then, in the case of reward-optimization, we define the value function as the *negative cost-to-go*, $V_t(\boldsymbol{x}) := -J_t^\star(\boldsymbol{x})$. In the case when the path measure is a discrete CTMC, the cost to go is determined by the number of jumps that occur in the interval $[t, T]$. We further expand the value function to the following form,

$$-V_t(\boldsymbol{x}) = \inf_u \mathbb{E}_{\boldsymbol{X} \sim \mathbb{P}^u} \left[ \left( \int_t^{t+\Delta t} + \int_{t+\Delta t}^T \right) \sum_{y \neq \boldsymbol{X}_s} C_t(\boldsymbol{X}_s, y) ds - r(\boldsymbol{X}_T) \Big| \boldsymbol{X}_T = \boldsymbol{x} \right]$$

$$= \left[ \Delta t \inf_u \sum_{y \neq x} C_t(x, y) + O(\Delta t^2) \right] + \inf_u \mathbb{E}_{\boldsymbol{X} \sim \mathbb{P}^u} \left[ -V_{t+\Delta t}(\boldsymbol{X}_{t+\Delta t}) \big| \boldsymbol{X}_t = \boldsymbol{x} \right] \tag{32}$$

Using the value function, we can derive the expression for the optimal generator $\boldsymbol{Q}^\star$.

**Lemma 3** (Optimal Generator). *Given a base generator $\boldsymbol{Q}^0$ and the value function $V_t$, the optimal generator $\boldsymbol{Q}^\star$ takes the form*

$$\boldsymbol{Q}_t^\star(x,y) = \boldsymbol{Q}_t^0(x,y)\exp(V_t(y) - V_t(x)) \tag{33}$$

*Proof.* Expanding the second term in (32), we have

$$\inf_u \mathbb{E}_{\boldsymbol{X} \sim \mathbb{P}^u}\left[-V_{t+\Delta t}(\boldsymbol{X}_{t+\Delta t})\big|\boldsymbol{X}_T = \boldsymbol{x}\right]$$

$$\overset{(21)}{=} \inf_u \left[-\sum_y V_{t+\Delta t}(y)\left(\mathbf{1}_{x=y} + \Delta t \boldsymbol{Q}_t^u(x,y) + \mathcal{O}(\Delta t^2)\right)\right]$$

$$= \inf_u \left[-V_{t+\Delta t}(x) - \Delta t \sum_{x\neq y} V_{t+\Delta t}(y)\boldsymbol{Q}_t^u(x,y) + \Delta t \sum_{x\neq y} V_{t+\Delta t}(x)\boldsymbol{Q}_t^u(x,y) + \mathcal{O}(\Delta t^2)\right]$$

$$= -V_{t+\Delta t}(x) + \Delta t \inf_u \left[\sum_{x\neq y} \boldsymbol{Q}_t^u(x,y)(V_{t+\Delta t}(x) - V_{t+\Delta t}(y))\right] + \mathcal{O}(\Delta t^2) \tag{34}$$

Now, substituting back into (32) and defining $C_t(x,y) = \left(\boldsymbol{Q}_t^u \log \frac{\boldsymbol{Q}^u}{\boldsymbol{Q}^0} - \boldsymbol{Q}^u + \boldsymbol{Q}^0\right)(x,y)$, we have

$$\partial_t V_t = \inf_u \left[\sum_{y\neq x} \left(\boldsymbol{Q}_t^u \log \frac{\boldsymbol{Q}^u}{\boldsymbol{Q}^0} - \boldsymbol{Q}^u + \boldsymbol{Q}^0\right)(x,y) + (V_t(x) - V_t(y))\boldsymbol{Q}_t^u(x,y)\right] \tag{35}$$

The infimum can be achieved by minimizing a convex scalar function for each pair $x \neq y$ defined as

$$f(\boldsymbol{Q}^u) = \boldsymbol{Q}^u \log \frac{\boldsymbol{Q}^u}{\boldsymbol{Q}^0} - \boldsymbol{Q}^u + \boldsymbol{Q}^0 + (V_t(x) - V_t(y))\boldsymbol{Q}^u$$

$$f'(\boldsymbol{Q}^u) = \log \frac{\boldsymbol{Q}^u}{\boldsymbol{Q}^0} + (V_t(x) - V_t(y)) \tag{36}$$

Setting $f'(\boldsymbol{Q}^u) = 0$, we get

$$\log \frac{\boldsymbol{Q}^\star}{\boldsymbol{Q}^0} = V_t(y) - V_t(x) \implies \boldsymbol{Q}_t^\star(x,y) = \boldsymbol{Q}_t^0(x,y)\exp(V_t(y) - V_t(x)) \tag{37}$$

which concludes our proof. $\qquad\square$

**Corollary 2** (Hamilton-Jacobi Bellman (HJB) Equation). *The value function $V_t(x) = \mathbb{E}[r(\boldsymbol{X}_T)|\boldsymbol{X}_t = x]$ satisfies the HJB equation given by*

$$\partial_t V_t(x) = \sum_{y\neq x} \boldsymbol{Q}_t^0(x,y)\left(1 - e^{V_t(y)-V_t(x)}\right) \iff \partial_t e^{V_t(x)} \sum_{y\neq x} \boldsymbol{Q}_t^0(x,y)\left(e^{V_t(x)} - e^{V_t(y)}\right) \tag{38}$$

*Proof.* The proof follows from substituting the optimal $\boldsymbol{Q}_t^\star(x,y) = \boldsymbol{Q}_t^0(x,y)\exp(V_t(y) - V_t(x))$ into equation (35) and the second equation follows immediately after. $\qquad\square$

**Lemma 4** (Optimal Path Measure). *Given the value function $V_t(\boldsymbol{x})$, the optimal path measure $\mathbb{P}^\star$ takes the form*

$$\mathbb{P}_t^\star(x) = \frac{1}{Z}\mathbb{P}_t^0(x)e^{V_t(\boldsymbol{x})}, \quad Z := \mathbb{E}_{x \sim \mathbb{P}_T^0}[e^{r(x)}] \tag{39}$$

*Proof.* Let $h_t(x) := \frac{1}{Z}\mathbb{P}_t^0(x)e^{V_t(x)}$. By definition, we have $h_T = \mathbb{P}_T^\star$. Now, we aim to show that $h_t$ satisfies the Kolmogorov forward equation for the optimal generator $\boldsymbol{Q}_t^\star$. First, we restate the Kolmogorov forward equation from Lemma 1 for $\boldsymbol{Q}^0$ as

$$\partial_t \mathbb{P}_t^0(\boldsymbol{x}) = \sum_{y\neq x}(\boldsymbol{Q}_t^0(y,x)\mathbb{P}_t^0(y) - \boldsymbol{Q}_t^0(x,y)\mathbb{P}_t^0(x)) \tag{40}$$

Furthermore, by Corollary 2, we have

$$\partial_t e^{V_t(x)} = \sum_{y \neq x} \boldsymbol{Q}_t^0(x, y) \left( e^{V_t(x)} - e^{V_t(y)} \right) \tag{41}$$

Now, taking the partial derivative of $h_t$, we get

$$\partial_t h_t(x) = \frac{1}{Z} \left[ \partial_t \mathbb{P}_t^0(x) e^{V_t(x)} + \mathbb{P}_t^0 \partial_t e^{V_t(x)} \right]$$

$$= \frac{1}{Z} \left[ e^{V_t(x)} \sum_{y \neq x} \left( \boldsymbol{Q}_t^0(y, x) \mathbb{P}_t^0(y) - \boldsymbol{Q}_t^0(x, y) \mathbb{P}_t^0(x) \right) + \mathbb{P}_t^0(x) \sum_{y \neq x} \boldsymbol{Q}_t^0(x, y) \left( e^{V_t(x)} - e^{V_t(y)} \right) \right]$$

$$= \sum_{y \neq x} \left( \boldsymbol{Q}_t^0(y, x) \frac{1}{Z} \mathbb{P}_t^0(y) e^{V_t(x)} - \boldsymbol{Q}_t^0(x, y) \frac{1}{Z} \mathbb{P}_t^0(x) e^{V_t(y)} \right)$$

$$= \sum_{y \neq x} \left( \boldsymbol{Q}_t^0(y, x) e^{V_t(x) - V_t(y)} h_t(y) - \boldsymbol{Q}_t^0(x, y) e^{V_t(x) - V_t(y)} h_t(x) \right)$$

$$\overset{(3)}{=} \sum_{y \neq x} \left( \boldsymbol{Q}_t^\star(y, x) h_t(y) - \boldsymbol{Q}_t^\star(x, y) h_t(x) \right)$$

which is the Kolmogorov forward equation for the tilted distribution $\mathbb{P}^\star$ with the optimal generator $\boldsymbol{Q}^\star$ in (3). By uniqueness of solutions to the Kolmogorov forward equation, we have $\mathbb{P}_t^\star(x) = \frac{1}{Z} \mathbb{P}_t^0(x) e^{V_t(x)}$. $\qquad \square$

Now, we derive the expression for the Radon-Nikodym derivative between the optimal and reference path measures $\frac{d\mathbb{P}^\star}{d\mathbb{P}^0}(\boldsymbol{X}_{0:T})$ in the following Lemma.

**Lemma 5** (Radon-Nikodym Derivative of Optimal and Reference Path Measure). *Given the optimal form of the path measure $\mathbb{P}^\star$ and generator $\boldsymbol{Q}^\star$ from Lemmas 3 and 4, the RND for any $\boldsymbol{X}_{0:T}$ can be expressed as*

$$\frac{d\mathbb{P}^\star}{d\mathbb{P}^0}(\boldsymbol{X}_{0:T}) = \frac{1}{Z} e^{r(\boldsymbol{X}_T)}, \quad where \quad Z = \mathbb{E}_{\mathbb{P}_T^0}[e^r] \tag{42}$$

*Proof.* Using Lemmas 2, 3, and 4, we have

$$\log \frac{d\mathbb{P}^\star}{d\mathbb{P}^0}(\boldsymbol{X}_{0:T}) \overset{(2)}{=} \log \frac{d\mathbb{P}_0^\star}{d\mathbb{P}_0^0}(\boldsymbol{X}_0) + \sum_{t: \boldsymbol{X}_{t-} \neq \boldsymbol{X}_t} \log \frac{\boldsymbol{Q}_t^\star(\boldsymbol{X}_{t-}, \boldsymbol{X}_t)}{\boldsymbol{Q}_t^0(\boldsymbol{X}_{t-}, \boldsymbol{X}_t)} + \int_0^T \sum_{y \neq \boldsymbol{X}_t} (\boldsymbol{Q}_t^0 - \boldsymbol{Q}_t^\star)(\boldsymbol{X}_t, y) dt$$

$$\overset{(4,3)}{=} V_0(\boldsymbol{X}_0) - \log Z + \sum_{t: \boldsymbol{X}_{t-} \neq \boldsymbol{X}_t} (V_t(\boldsymbol{X}_t) - V_t(\boldsymbol{X}_{t-}) + \int_0^T \sum_{y \neq \boldsymbol{X}_t} \boldsymbol{Q}_t^0(\boldsymbol{X}_t, y)(1 - e^{V_t(y) - V_t(\boldsymbol{X}_t)}) dt$$

The CTMC process $\boldsymbol{X}_{0:T}$ is a piecewise càdlàg function and $t \mapsto V_t(x)$ is continuous for all $x$, we can define discrete jump times at $0 < t_1 < \cdots < t_n < \cdots < t_{N-1} < T$ and write

$$V_T(\boldsymbol{X}_T) - V_0(\boldsymbol{X}_0) = \sum_{n=0}^{N-1} (V_{t_{n+1}}(\boldsymbol{X}_{t_n}) - V_{t_n}(\boldsymbol{X}_{t_n}) + \sum_{n=1}^{N-1} (V_{t_n}(\boldsymbol{X}_{t_n}) - V_{t_n}(\boldsymbol{X}_{t_{n-1}}))$$

$$= \sum_{n=0}^{N-1} \int_{t_n}^{t_{n+1}} \partial_t V_t(\boldsymbol{X}_{t_n}) dt + \sum_{t: \boldsymbol{X}_{t-} \neq \boldsymbol{X}_t} (V_t(\boldsymbol{X}_t) - V_t(\boldsymbol{X}_{t-}))$$

$$= \int_0^T \partial_t V_t(\boldsymbol{X}_t) dt + \sum_{t: \boldsymbol{X}_{t-} \neq \boldsymbol{X}_t} (V_t(\boldsymbol{X}_t) - V_t(\boldsymbol{X}_{t-}))$$

$$\implies V_0(\boldsymbol{X}_0) = V_T(\boldsymbol{X}_T) - \int_0^T \partial_t V_t(\boldsymbol{X}_t) dt - \sum_{t: \boldsymbol{X}_{t-} \neq \boldsymbol{X}_t} (V_t(\boldsymbol{X}_t) - V_t(\boldsymbol{X}_{t-})) \tag{43}$$

Using Lemma 3, we also have

$$\int_0^T \sum_{y \neq \boldsymbol{X}_t} (\boldsymbol{Q}_t^0 - \boldsymbol{Q}_t^\star)(\boldsymbol{X}_t, y)\mathrm{d}t = \int_0^T \sum_{y \neq \boldsymbol{X}_t} \boldsymbol{Q}_t^0(\boldsymbol{X}_t, y)\left(1 - e^{V_t(y) - V_t(\boldsymbol{X}_t)}\right)\mathrm{d}t$$

$$= \int_0^T \partial_t V_t(\boldsymbol{X}_t)\mathrm{d}t \tag{44}$$

Substituting the expressions for $V_0(\boldsymbol{X}_0)$ and $\int_0^T \sum_{y \neq \boldsymbol{X}_t} \boldsymbol{Q}_t^0(\boldsymbol{X}_t, y)\left(1 - e^{V_t(y) - V_t(\boldsymbol{X}_t)}\right)\mathrm{d}t$, the RND reduces to

$$\log \frac{\mathrm{d}\mathbb{P}^\star}{\mathrm{d}\mathbb{P}^0}(\boldsymbol{X}_{0:T}) = V_T(\boldsymbol{X}_T) - \log Z \implies \frac{\mathrm{d}\mathbb{P}^\star}{\mathrm{d}\mathbb{P}^0}(\boldsymbol{X}_{0:T}) = \frac{1}{Z}V_T(\boldsymbol{X}_T) \tag{45}$$

Given the terminal reward $V_T(\boldsymbol{X}_T) = r(\boldsymbol{X}_T)$, we conclude our proof. $\qquad\square$

## C   THEORETICAL PROOFS

### C.1   OFF-POLICY LEARNING FOR MASKED DISCRETE DIFFUSION FINE-TUNING

Here, we will derive the WDCE objective in (7) used for our off-policy RL fine-tuning algorithm, which matches the optimal path measure $\mathbb{P}^\star$. We note that this objective was derived in Zhu et al. (2025c) for the training of Masked Diffusion Neural Samplers (MDNS).

**Lemma 6.** *The RND between the optimal path measure $\mathbb{P}^\star$ defined in (4) and the current path measure of the fine-tuned model $\mathbb{P}^v$ under the masked discrete diffusion model formulation can be written as*

$$\log \frac{\mathrm{d}\mathbb{P}^\star}{\mathrm{d}\mathbb{P}^v}(\boldsymbol{X}_{0:T}) = \frac{r(\boldsymbol{X}_T)}{\alpha} + \underbrace{\sum_{t:\boldsymbol{X}_s \neq \boldsymbol{X}_t} \sum_{\ell:\boldsymbol{X}_s^\ell \neq \boldsymbol{X}^\ell} \log \frac{p^{pre}(\boldsymbol{X}_s^\ell | \boldsymbol{X}_t^{UM})}{p^v(\boldsymbol{X}_s^\ell | \boldsymbol{X}_t^{UM})} - \log Z}_{:= W^v(\boldsymbol{X}_{0:T})} \tag{46}$$

Recall the special form of the optimal generator for MDM from (1) as

$$\boldsymbol{Q}_t(\boldsymbol{x}, \boldsymbol{y}) = \gamma(t) \Pr_{\boldsymbol{X} \sim p_{\text{data}}}(\boldsymbol{X}^\ell = d | \boldsymbol{X}^{\text{UM}} = \boldsymbol{x}^{\text{UM}})\mathbf{1}_{\boldsymbol{x}^\ell = d, \boldsymbol{y} = \boldsymbol{x}^{\ell \leftarrow d}}$$

Now, we can write the exit rate from $\boldsymbol{x}$ as

$$\sum_{y \neq x} \boldsymbol{Q}_t^v(x, y) = \sum_{d:\boldsymbol{x}^\ell = \boldsymbol{M}} \sum_d \boldsymbol{Q}_t^u(\boldsymbol{x}, \boldsymbol{x}^{\ell \leftarrow d}) = \gamma(t) \sum_{d:\boldsymbol{x}^\ell = \boldsymbol{M}} 1 = \gamma(t)\big|\{\ell : \boldsymbol{x}^\ell = \boldsymbol{M}\}\big| \tag{47}$$

Since the pre-trained model is trained with the same noise schedule $\gamma$, we can also write

$$\sum_{y \neq x} \boldsymbol{Q}_t^0(x, y) = \gamma(t)\big|\{\ell : \boldsymbol{x}^\ell = \boldsymbol{M}\}\big| \tag{48}$$

Therefore, the last term in the RND cancels, and we derive a simplified form of the RND specific to MDMs as

$$\log \frac{d\mathbb{P}^{\star}}{d\mathbb{P}^v}(\boldsymbol{X}_{0:T}) = \log \frac{d\mathbb{P}^{\star}}{d\mathbb{P}^0} \frac{d\mathbb{P}^0}{d\mathbb{P}^v}$$

$$= \log \frac{d\mathbb{P}^{\star}}{d\mathbb{P}^0} + \log \frac{d\mathbb{P}^0}{d\mathbb{P}^v}$$

$$= \frac{r(\boldsymbol{X}_T)}{\alpha} - \log Z + \sum_{t: \boldsymbol{X}_{t-} \neq \boldsymbol{X}_t} \log \frac{\boldsymbol{Q}_t^0(\boldsymbol{X}_{t-}, \boldsymbol{X}_t)}{\boldsymbol{Q}_t^v(\boldsymbol{X}_{t-}, \boldsymbol{X}_t)} + \int_0^T \sum_{y \neq \boldsymbol{X}_t} (\boldsymbol{Q}_t^v - \boldsymbol{Q}_t^0)(\boldsymbol{X}_t, y)dt$$

$$= \frac{r(\boldsymbol{X}_T)}{\alpha} - \log Z + \sum_{t: \boldsymbol{X}_{t-} \neq \boldsymbol{X}_t} \log \frac{\boldsymbol{Q}_t^0(\boldsymbol{X}_{t-}, \boldsymbol{X}_t)}{\boldsymbol{Q}_t^v(\boldsymbol{X}_{t-}, \boldsymbol{X}_t)}$$

$$= \frac{r(\boldsymbol{X}_T)}{\alpha} - \log Z + \sum_{t: \boldsymbol{X}_{t-} \neq \boldsymbol{X}_t} \log \frac{\gamma(t)p^{\mathrm{pre}}(\boldsymbol{X}^\ell = d|\boldsymbol{X}^{\mathrm{UM}} = \boldsymbol{x}^{\mathrm{UM}})\mathbf{1}_{\boldsymbol{x}^\ell = d, \boldsymbol{y} = \boldsymbol{x}^{\ell \leftarrow d}}}{\gamma(t)p^v(\boldsymbol{X}^\ell = d|\boldsymbol{X}^{\mathrm{UM}} = \boldsymbol{x}^{\mathrm{UM}})\mathbf{1}_{\boldsymbol{x}^\ell = d, \boldsymbol{y} = \boldsymbol{x}^{\ell \leftarrow d}}}$$

$$= \frac{r(\boldsymbol{X}_T)}{\alpha} + \underbrace{\sum_{t: \boldsymbol{X}_s \neq \boldsymbol{X}_t} \sum_{\ell: \boldsymbol{X}_s^\ell \neq \boldsymbol{X}^\ell} \log \frac{p^{\mathrm{pre}}(\boldsymbol{X}_s^\ell|\boldsymbol{X}_t^{\mathrm{UM}})}{p^v(\boldsymbol{X}_s^\ell|\boldsymbol{X}_t^{\mathrm{UM}})}}_{:=W^v(\boldsymbol{X}_{0:T})} - \log Z \tag{49}$$

where we denote the log-RND excluding the normalization term as $W^v$. $\qquad\square$

---

**Corollary 3** (Weighted Denoising Cross-Entropy (WDCE) Loss). *The solution to the Weighted Denoising Cross-Entropy (WDCE) loss defined as*

$$\mathcal{F}_{WDCE}(\mathbb{P}^u, \mathbb{P}^{\star}) = \mathop{\mathbb{E}}_{\boldsymbol{X} \sim \mathbb{P}^v} \left[ \frac{1}{Z} e^{W^v(\boldsymbol{X}_{0:T})} \mathbb{E}_{\lambda \sim \mathrm{Unif}(0,1)} \left[ \frac{1}{\lambda} \mathbb{E}_{\mu_\lambda(\tilde{\boldsymbol{x}}|\boldsymbol{x})} \sum_{\ell: \tilde{\boldsymbol{x}}^\ell = \boldsymbol{M}} -\log p^{u_\theta}(\tilde{\boldsymbol{x}})_{\ell, \boldsymbol{x}^\ell} \right] \right]$$

*is the optimal generator $\boldsymbol{Q}^{\star}$ of $\mathbb{P}^{\star}$.*

*Proof.* First, we recall the definition of the cross-entropy loss between the optimal and controlled path measure, defined as

$$\mathcal{F}_{\mathrm{CE}}(\mathbb{P}^{\star}, \mathbb{P}^u) := \mathbb{E}_{\mathbb{P}^{\star}} \left[ \log \frac{d\mathbb{P}^{\star}}{d\mathbb{P}^u} \right] = \mathbb{E}_{\mathbb{P}^v} \left[ \frac{d\mathbb{P}^{\star}}{d\mathbb{P}^v} \log \frac{d\mathbb{P}^{\star}}{d\mathbb{P}^u} \right] \tag{50}$$

Then, writing the objective with respect to the log-RND of $\mathbb{P}^v$ and $\mathbb{P}^u$, we have

$$\mathcal{F}_{\mathrm{CE}}(\mathbb{P}^{\star}, \mathbb{P}^u) = \mathbb{E}_{\boldsymbol{X}_{0:T} \sim \mathbb{P}^v} \left[ \frac{1}{Z} e^{W^v(\boldsymbol{X}_{0:T})} W^u(\boldsymbol{X}_{0:T}) \right] \tag{51}$$

To further simplify $W^u$, we can discard the terms independent to $u$ in $W^u$ to get

$$W^u(\boldsymbol{X}_{0:T}) = \sum_{t: \boldsymbol{X}_s \neq \boldsymbol{X}_t} \sum_{\ell: \boldsymbol{X}_s^\ell \neq \boldsymbol{X}^\ell} -\log p^v(\boldsymbol{X}_s^\ell|\boldsymbol{X}_t^{\mathrm{UM}}) \tag{52}$$

Instead of computing the loss only with respect to the trajectory that generates $\boldsymbol{X}_T$, Zhu et al. (2025c) proposes to compute a loss over many potential trajectories for each single clean sample $\boldsymbol{X}_T$ by remasking $\boldsymbol{X}_T$ and computing the DCE loss in (2) with respect to each of the masked tokens.

$$W^u(\boldsymbol{X}_{0:T}) = \mathbb{E}_{\lambda \sim \mathrm{Unif}(0,1)} \left[ \frac{1}{\lambda} \mathbb{E}_{\mu_\lambda(\tilde{\boldsymbol{x}}|\boldsymbol{x})} \sum_{\ell: \tilde{\boldsymbol{x}}^\ell = \boldsymbol{M}} -\log p^{u_\theta}(\tilde{\boldsymbol{x}})_{\ell, \boldsymbol{x}^\ell} \right] \tag{53}$$

where $p^{u_\theta}(\tilde{\boldsymbol{x}})_{\ell, \boldsymbol{x}^\ell}$ takes the probability of the $\ell$th token being in state $\boldsymbol{x}^\ell$. This gives us the **weighted denoising cross-entropy** (WDCE) loss defined as

$$\mathcal{F}_{\mathrm{WDCE}}(\mathbb{P}^u, \mathbb{P}^{\star}) = \mathop{\mathbb{E}}_{\boldsymbol{X} \sim \mathbb{P}^v} \left[ \frac{1}{Z} e^{W^v(\boldsymbol{X}_{0:T})} \mathbb{E}_{\lambda \sim \mathrm{Unif}(0,1)} \left[ \frac{1}{\lambda} \mathbb{E}_{\mu_\lambda(\tilde{\boldsymbol{x}}|\boldsymbol{x})} \sum_{\ell: \tilde{\boldsymbol{x}}^\ell = \boldsymbol{M}} -\log p^{u_\theta}(\tilde{\boldsymbol{x}})_{\ell, \boldsymbol{x}^\ell} \right] \right]$$

where we define $v = \bar{u} := \mathtt{stopgrad}(u_\theta)$ and $W^{\bar{u}}$ is computed with respect to the optimal measure $\mathbb{P}^{\star} = \mathbb{P}^{\mathrm{pre}} \exp(r(\boldsymbol{X}_T))$ given the pre-trained generator $\boldsymbol{Q}^{\mathrm{pre}}$ that produces the path measure $\mathbb{P}^{\mathrm{pre}}$. $\qquad\square$

## C.2 Justification for Off-Policy Reinforcement Learning

While there exist several well-known challenges associated with off-policy reinforcement learning relative to other RL methods, we highlight that **TR2-D2** overcomes all of these challenges which we rigorously justify in this section.

**Distribution Mismatch** Our framework is rigorously derived using stochastic optimal control theory, which is theoretically guaranteed to converge to the optimal reward-tilted distribution of the initial model $\mathbb{P}^0$ defined as $\mathbb{P}^\star \propto \frac{1}{Z}\mathbb{P}^0 e^r$ with proof in App C.1.

**Performance Comparison to On-Policy RL** TThe difference in performance of on-policy and off-policy RL depends on the objective we use. In general, if we use an objective which is **computed under the expectation of the current policy** $\mathbb{E}_{\mathbb{P}^{u_\theta}}[\cdot]$, then on-policy will outperform off-policy due to the variance in the importance weights. However, our objective takes the expectation under optimal policy $\mathbb{E}_{\mathbb{P}^\star}[\mathcal{L}(\theta; \boldsymbol{X}_T)]$ as defined in (7), which is **not dependent on the current policy**. For this objective, on-policy is not necessarily better than off-policy, and off-policy is much more efficient, as it amortizes the search cost by reusing high-reward samples from the buffer without recalculating the log RND weights of the current policy.

**High Variance of Importance Sampling** Unlike PPO/GRPO-type policy gradient methods (Schulman et al., 2017; Shao et al., 2024), which use importance sampling to debias the distribution to the current policy distribution at a token level and suffer from high variance, TR2-D2 uses importance sampling to debias to the optimal policy distribution at a sequence level, reducing importance weight variance. This advantage is also discussed in GSPO (Zheng et al., 2025a), which uses sequence-level importance weight for AR-LLM finetuning, and we believe that TR2-D2 enjoys a similar benefit.

**Periodic Policy Realignment** Since we are not debiasing with respect to the current policy distribution, the debiased target remains fixed throughout training. Therefore, we avoid the typical concern that importance-weight variance may explode as the policy evolves because the target distribution itself is not moving. Instead, we perform **debiasing relative to an optimal, stationary target**, which in theory removes the need to frequently update the sampling policy to maintain numerical stability in the training objective. In practice, we still update the sampling policy periodically to improve sample efficiency and convergence speed, although such updates are not strictly required for stability.

## C.3 Justification for the Decoupling of Tree Search and Fine-Tuning

Our framework relies on the fact that the tree search algorithm used to populate the replay buffer and the off-policy RL algorithm are **decoupled**, enabling integration of any pair of search and off-policy RL algorithms. The effectiveness of our approach is grounded in two key properties of off-policy RL: **(1)** it trains on trajectories generated from an *arbitrary* reference measure in a frozen replay buffer to inform the update to the current policy and **(2)** it fits the buffer distribution with theoretical guarantees which **amortizes** the cost of searching by letting the fine-tuned policy inherit the high-quality samples generated from the search algorithm.

## C.4 Achieving Pareto-Optimality with Multi-Objective Fine-Tuning

To prove that our multi-objective fine-tuning framework from Sec 5 enables the fine-tuned model to generate samples that approach Pareto-optimality, we first establish the following Lemma.

> **Lemma 7** (Non-Decreasing Hypervolume of Buffer). *Given a set $\mathcal{S}$ of candidate sequence rewards $\mathcal{S} = \{r^i\}$ and the current set of rewards in the buffer $\mathcal{B} = \{r^\star\}$, the HV of the non-dominated rewards in the union of both sets $\mathcal{B} \cup \mathcal{S}$ is non-decreasing from the HV of the original set $\mathcal{B}$*
>
> $$HV(ND(\mathcal{B} \cup \mathcal{S})) \geq HV(\mathcal{B}) \tag{54}$$
>
> *wher $ND(\cdot)$ is takes the set of non-dominated solutions.*

*Proof.* Let $\bar{r}$ be a reference reward vector such that all feasible rewards dominate it (i.e., $r \succ \bar{r}$). Denoting the axis-aligned orthant between the coordinates $\bar{r}$ and $r$ as $[\bar{r}, r] = \{y \in \mathbb{R}^K : \forall k, \ \bar{r}_k \leq y_k \leq r_k\}$, we write the hypervolume (HV) as the Lebesgue measure $\mu(\cdot)$ of the union of the orthants generated from a set.

$$\text{HV}(\bar{r}; \mathcal{B}) = \mu(U(\mathcal{B})) := \mu \left( \bigcup_{r^\star \in \mathcal{B}} [\bar{r}, r^\star] \right) \tag{55}$$

It is straightforward to show that given $\mathcal{B} \subseteq \mathcal{B} \cup \mathcal{S}$ we have

$$U(\mathcal{B}) \subseteq U(\mathcal{B} \cup \mathcal{S}) \implies \text{HV}(\mathcal{B}) \leq \text{HV}(\mathcal{B} \cup \mathcal{S}) \tag{56}$$

Now, we want to show that the union of the *non-dominated* subset $\text{ND}(\mathcal{B} \cup \mathcal{S})$ does not shrink the union:

$$U(\text{ND}(\mathcal{B} \cup \mathcal{S})) = \bigcup_{r^\star \in \text{ND}(\mathcal{B} \cup \mathcal{S})} [\bar{r}, r^\star] = \bigcup_{r \in \mathcal{B} \cup \mathcal{S}} [\bar{r}, r] = U(\mathcal{B} \cup \mathcal{S}) \tag{57}$$

By definition of $[\bar{r}, \cdot]$, if a reward $r^\star$ *dominates* $y$ (i.e. $r^\star \succ y$), we have

$$[\bar{r}, y] \subseteq [\bar{r}, r^\star] \tag{58}$$

Let $y \in \mathcal{B} \cup \mathcal{S}$. If $y \in \text{ND}(\mathcal{B} \cup \mathcal{S})$, then clearly $[\bar{r}, y] \in U(\text{ND}(\mathcal{B} \cup \mathcal{S}))$. If $y \notin \text{ND}(\mathcal{B} \cup \mathcal{S})$, then by definition, there exists some $r^\star \in \text{ND}(\mathcal{B} \cup \mathcal{S})$ that dominates it such that $r^\star \succ y$. Then, it follows that $\forall y \in \mathcal{B} \cup \mathcal{S}$, we have $[\bar{r}, y] \subseteq [\bar{r}, r^\star]$ and

$$U(\mathcal{B} \cup \mathcal{S}) \subseteq U(\text{ND}(\mathcal{B} \cup \mathcal{S})) \tag{59}$$

Since $\text{ND}(\mathcal{B} \cup \mathcal{S}) \subseteq \mathcal{B} \cup \mathcal{S} \implies U(\text{ND}(\mathcal{B} \cup \mathcal{S})) \subseteq U(\mathcal{B} \cup \mathcal{S})$, we have shown that $U(\text{ND}(\mathcal{B} \cup \mathcal{S})) = U(\mathcal{B} \cup \mathcal{S})$. Since $U(\mathcal{B}) \subseteq U(\mathcal{B} \cup \mathcal{S})$, we get

$$\mu(U(\text{ND}(\mathcal{B} \cup \mathcal{S}))) \geq \mu(U(\mathcal{B} \cup \mathcal{S})) \implies \text{HV}(\text{ND}(\mathcal{B} \cup \mathcal{S}) \geq \text{HV}(\mathcal{B} \cup \mathcal{S}) \tag{60}$$

which concludes our proof. $\qquad\square$

> **Proposition 5.1** (Pareto Optimization of Buffer). *With each iteration of the search, the buffer $\mathcal{B}$ approaches the Pareto front $\mathcal{P}^\star$, where the hypervolume generated by the rewards in the set is maximized.*

First, we establish the following assumptions: **(A1)** Each node in the tree is sufficiently explored, such that $N_{\text{visits}} \to \infty$ as the number of iterations goes to infinity $N_{\text{iter}} \to \infty$. **(A2)** The reward function is bounded and defined over the feasible search space $\mathcal{X}$. **(A3)** There is a positive probability $p > 0$ of discovering a sequence that strictly increases the HV of $\mathcal{B}$ by $\Delta \geq \varepsilon$ with each search iteration.

For the purpose of this proof, we do not limit the size of the buffer set $\mathcal{B}$. Let $\mathcal{P}^\star$ denote the Pareto frontier of the feasible solution space $\mathcal{X}$ and multi-reward function $r$, such that $\text{HV}(\mathcal{P}^\star)$ is the maximum feasible hypervolume.

By Lemma 7, we have shown that the HV is non-decreasing with each search iteration. By our assumption, we have that for all iterations where $\text{HV}(\mathcal{B}) \leq \text{HV}(\mathcal{P}^\star) - \epsilon$, the expected HVI of $\mathcal{B}'$ after each iteration is proportional to the discovery probability $p > 0$ given by

$$\mathbb{E}\left[\text{HV}(\mathcal{B}') - \text{HV}(\mathcal{B})\right] \geq p\varepsilon \tag{61}$$

After $N_{\text{iter}}$ iterations of the search, the search-optimized buffer $\mathcal{B}^\star$ has an expected HV given by

$$\mathbb{E}\left[\text{HV}(\mathcal{B}^{N_{\text{iter}}})\right] \geq \text{HV}(\mathcal{B}^0) + N_{\text{iter}} p\varepsilon \tag{62}$$

which converges to $\text{HV}(\mathcal{P}^\star)$ as $N_{\text{iter}} \to \infty$. $\qquad\square$

This convergence guarantee holds for **any search algorithm** that satisfies **(A1)-(A3)**, that is, it sufficiently explores the solution space and discovers $\varepsilon$-Pareto solutions with non-negative probability with each search iteration. MCTS satisfies **(A1)** with the exploration constant $c$ so every path has non-zero probability of being sampled and **(A2)-(A3)** given that the reward oracle is trained on an empirical subset of the dataset used to train the pre-trained model. Furthermore, the MCTS algorithm exploits sampling paths based on an estimated future reward derived from previous iterations, which intuitively increases the probability of discovering a high-reward sample that contributes positively to HVI at each iteration.

## D    REGULATORY DNA EXPERIMENT DETAILS

We largely follow the experimental setup and evaluation metrics from Wang et al. (2025) to ensure fair benchmarking.

### D.1    EXPERIMENT SETUP

**Pre-trained Model**    We use the pre-trained masked discrete diffusion model from Wang et al. (2025) built on the Masked Discrete Language Model (MDLM) framework (Sahoo et al., 2024). The model is trained on 700k DNA enhancer sequences 200 base-pairs in length from the Gosai dataset (Gosai et al., 2023). The backbone architecture is a CNN with a linear noise schedule following Stark et al. (2024).

**Enhancer Activity Predictor**    We use the pre-trained reward oracles from Wang et al. (2025), which predict the enhancer activity in the HepG2 cell line. Following the procedure in (Lal et al., 2024), the Gosai dataset (Gosai et al., 2023) of 700K DNA sequences is split into two disjoint sets which each contains enhancers from half of the 23 human chromosomes. One is used to train the fine-tuning oracle for optimization during fine-tuning, while the other is used to train the evaluation oracle, which was used to compute the Pred-Activity reported in Table 1. Both models are built on the Enformer architecture (Avsec et al., 2021) and achieved Pearson correlations of $> 0.85$ on the held-out sets.

**Fine-Tuning Setup**    We load the pre-trained model with frozen weights for log-RND computation and load the model with unfrozen weights for fine-tuning. We set the buffer size to 128 and the number of diffusion steps to 128, to remain consistent with (Wang et al., 2025). We conducted ablations on various hyperparameters, including the regularization strength $\alpha$, the use of MCTS, the resampling frequency $N_{\text{resample}}$, and the number of MCTS iterations $N_{\text{iter}}$, with results reported in Table 7. All the DNA experiments were conducted on an NVIDIA H100 GPU. We used the AdamW optimizer with a learning rate of $\eta = 3 \times 10^{-4}$. For evaluation, we compute metrics for 640 sequences with three random seeds and report the mean and standard deviation, consistent with Wang et al. (2025); Zekri & Boullé (2025).

### D.2    ENHANCER EVALUATION METRICS

**Mean Predicted Activity (Pred-Activity)**    We use the fine-tuning and evaluation reward oracles from Wang et al. (2025), which are trained on disjoint splits of the Gosai dataset of 700k DNA enhancer sequences (Gosai et al., 2023) labeled with the measured expression of the sequence in the HepG2 cell line. We fine-tune the pre-trained generator to optimize the predicted activity by the fine-tuning oracle and report the **median predicted activity** by the evaluation oracle in Table 1 for comparison against baseline models.

**Binary Classification on Chromatin Accessibility (ATAC-Acc)**    We further validate the predicted enhancer activity from a classifier that is not directly optimized during fine-tuning. Specifically, we use the binary classification model (%) that predicts the chromatin accessibility of an enhancer sequence in the HepG2 cell line, where positive accessibility indicates increased enhancer activity Wang et al. (2025); Lal et al. (2024).

**3-mer Pearson Correlation (3-mer Corr)**    To measure whether the fine-tuned model generates sequences within the distribution of the pre-trained model, we evaluate the 3-mer Pearson correlation between the generated sequences with the fine-tuned model and the $0.1\%$ of sequences with the highest HepG2 enhancer activity from the Gosai dataset (Gosai et al., 2023) used to train the pre-trained generator.

**Approximated Log-Likelihood of Sequences (App-Log-Lik)**    We evaluate the log-likelihood of the sequences generated by the fine-tuned model under the pre-trained model, which indicates whether the fine-tuning method over-optimizes the pre-trained model to generate out-of-distribution sequences. Specifically, we compute the likelihood as the evidence lower bound (ELBO) (Sahoo

Table 3: **Docking results for TR2-D2 generated peptide binders.** Binding affinities calculated with AutoDock VINA (kJ/mol; ↓), where lower values indicate stronger binding affinity, are reported for two randomly selected binders generated with the fine-tuned peptide models optimized for TfR, GLP-1R, and GLAST binding affinity. Classifier scores for binding affinity, solubility, non-hemolysis, non-fouling, and permeability optimized during fine-tuning are also reported.

| Target Protein | VINA Docking Score (kJ/mol; ↓) | Binding Affinity (↑) | Solubility (↑) | Non-hemolysis (↑) | Non-fouling (↑) | Permeability (↑) |
|---|---|---|---|---|---|---|
| TfR Binder 1 | −7.3 | 9.485 | 0.901 | 0.940 | 0.197 | −7.283 |
| TfR Binder 2 | −7.2 | 9.276 | 0.941 | 0.908 | 0.133 | −7.195 |
| GLP-1R Binder 1 | −6.4 | 9.211 | 0.901 | 0.925 | 0.494 | −7.254 |
| GLP-1R Binder 2 | −5.9 | 9.177 | 0.822 | 0.864 | 0.411 | −7.388 |
| GLAST Binder 1 | −5.5 | 9.198 | 0.769 | 0.874 | 0.188 | −7.285 |
| GLAST Binder 2 | −5.4 | 9.578 | 0.746 | 0.927 | 0.084 | −7.223 |

et al., 2024), where a larger ELBO indicates a higher likelihood of the fine-tuned sequence under the pre-trained model.

# E  PEPTIDE EXPERIMENT DETAILS

## E.1  EXPERIMENT SETUP

**Pre-trained Model**   We use the pre-trained bond-dependent masked discrete diffusion model from Tang et al. (2025), which generates peptide sequences containing the 20 canonical amino acids, in addition to non-canonical amino acids with chemical modifications and cyclizations in SMILES notation (Weininger, 1988). The model is trained on 11 million peptide SMILES, containing 7451 cyclic peptides from the CycPeptMPDB database (Li et al., 2023), $825,632$ peptide sequences from SmProt (Li et al., 2021), and  10 million peptides with cyclizations and non-canonical amino acids generated from CycloPs (Duffy et al., 2011; Feller & Wilke, 2025). To tokenize the SMILES sequences, we use the SMILES Pair Encoding (SPE) tokenizer (Li & Fourches, 2021; Feller & Wilke, 2025) containing a vocabulary of 581 SMILES tokens and 5 special tokens including `[PAD]`, `[UNK]`, `[CLS]`, `[SEP]`, and `[MASK]`. The generator is built on the Masked Discrete Language Model (MDLM) framework with a masking schedule that promotes early unmasking of peptide bond tokens (Tang et al., 2025). The backbone architecture is a RoFormer (Su et al., 2024) with 8 Transformer layers and 8 attention heads.

**Fine-Tuning Setup**   We load two versions of the pre-trained weights, one as the frozen pre-trained model for calculating the log-RND of the trajectory, and one with all weights unfrozen for fine-tuning. We also load the pre-trained classifiers for binding affinity, given a protein sequence input, solubility, non-hemolysis, non-fouling, and membrane permeability into a joint function that outputs a 5-dimensional vector of scores. We perform ablations on several hyperparameters as shown in Table 8 and choose the hyperparameters in Table 6 as default, given their superior performance. We trained for a total of 1000 epochs for each protein target and hyperparameter set. All peptide experiments were conducted on an NVIDIA A6000 GPU with a learning rate of $\eta = 10^{-4}$ with the AdamW optimizer (Loshchilov & Hutter, 2017) and gradient clipping. For evaluation, we generate 100 sequences i.i.d. with a single generation pass with 128 diffusion steps and report the mean and standard deviation of the predicted rewards.

**Target Proteins**   We evaluate the ability of TR2-D2 to generate peptide binders to therapeutically relevant protein targets, including Transferrin receptor (**TfR**), a common organ-specific drug-delivery target (Han et al., 2024a); glucagon-like peptide-1 receptor (**GLP-1R**), relevant for type-2 diabetes and obesity (Alfaris et al., 2024); glutamate-aspartate transporter (**GLAST**) protein abundant on the surface of astrocytes, a type of glial cell in the brain relevant to neurological disorders (Pajarillo et al., 2019); glial fibrillary acidic protein (**GFAP**), associated with Alexander disease (Eng, 1985; Quinlan et al., 2007); anti-Müllerian hormone type-2 receptor (**AMHR2**) which is a relavent target for polycystic ovarian syndrome (PCOS) therapy (Singh et al., 2023); and finally, neural cell adhesion molecule 1 (**NCAM1**), a transmembrane protein that is expressed on the surface of neurons and glial cells and facilitates neuronal migration and synaptogenesis.

## E.2 Therapeutic Property Classifiers

We use the pre-trained classifiers from Tang et al. (2025) for the prediction of target-protein binding affinity, solubility, non-hemolysis, non-fouling, and membrane permeability, which serve as the multi-objective reward functions.

**Protein Target-Binding Predictor**  The target-protein binding affinity classifier that embeds the target protein amino acid sequence using ESM-2-650M (Lin et al., 2023) and the peptide SMILES sequence with PeptideCLM (Feller & Wilke, 2025) and feeds the sequences to a cross multi-head attention Transformer architecture. The model is trained on 1806 protein-peptide pairs from the PepLand dataset (Zhang et al., 2023) containing canonical and non-canonical peptides with experimentally-validated $Kd/Ki/IC50$ binding affinity scores to various protein sequences, achieving a strong Spearman correlation coefficient of 0.869 on the training data and 0.633 on the held-out validation data. We classify scores as indicating weak binding ($< 6.0$), medium binding ($6.0 - 7.5$), and high binding ($\geq 7.5$).

**Solubility and Toxicity Predictors**  For solubility, non-hemolysis, and non-fouling, we used the XGBoost (Chen & Guestrin, 2016) logistic regression classifiers trained on binary data collected from the PepLand (Zhang et al., 2023) and PeptideBERT (Guntuboina et al., 2023) datasets, with 1 indicating the positive class and 0 indicating the negative class, and values ranging from $[0, 1]$. Positive solubility means a higher concentration of peptides can be dissolved in water, indicating enhanced drug loading. Positive non-hemolysis and non-fouling indicate lower destruction of red blood cells and lower off-target binding, respectively, which is essential for the non-toxicity of peptide drugs. The optimal positive thresholds for each score are 0.500 for solubility, 0.800 for non-hemolysis, and 0.450 for non-fouling.

**Membrane Permeability Predictor**  For membrane permeability, the classifier is an XGBoost regression model trained on 34,853 experimentally validated peptide SMILES with labeled PAMPA lipophilicity scores from the ChEMBL (Gaulton et al., 2012) and CycPeptMPDB (Li et al., 2023) databases, where less negative scores indicate stronger membrane permeability.

## E.3 Baselines and Evaluation

**Baseline Setup**  For the **pre-trained baseline**, we generate 100 sequences unconditionally from a single generation pass with 128 diffusion steps of the pre-trained model and compute the binding affinity to each of the protein targets as well as the other properties for comparison. For the **PepTune** baseline (Tang et al., 2025), we run inference-time guidance on the pre-trained model by running 100 iterations of Monte-Carlo Tree Guidance (MCTG) with 128 denoising steps on the set of five reward functions with the number of children set to $M = 50$.

**VINA Docking**  To visualize the binding position of generated peptides on the target protein, we used Autodock VINA (Eberhardt et al., 2021) for *in silico* confirmation of binding affinity. We processed the target proteins with MGITools (Morris et al., 2009) and the peptide SMILES with ETKDG from RDKit (Wang et al., 2020), and visualized the final protein-peptide complex in PyMol (DeLano et al., 2002).

## F Hyperparameter Discussion and Ablations

In this section, we provide a detailed analysis of the hyperparameters for TR2-D2. We include results of ablation studies for the number of epochs in Table 5, for enhancer DNA design in Table 7, and for multi-objective peptide design in Table 8 and Figures 3, 4, 5, and 6. In addition, we discuss the effect of MCTS search in App F.1, the number of fine-tuning epochs in App F.2, and all other hyperparameters in App F.3. We suggest tuning hyperparameters when adapting the **TR2-D2** framework to new modalities and tasks, and provide further intuition on each hyperparameter and their role in App F.3.

## F.1 Ablation on MCTS Search

To show the impact of MCTS on the effectiveness of fine-tuning, we conduct an ablation study that removes the use of MCTS to generate the buffer. Instead, we populate the buffer with sequences and their log-RND weights $(\boldsymbol{X}_T, W^{\bar{u}})$ using independent forward diffusion passes through the current policy model without gradient tracking. We maintain the same non-MCTS hyperparameters and show that removing the MCTS search results in worse metrics for enhancer DNA design (Table 7) and consistently lower rewards across all five objectives for multi-objective therapeutic peptide design (Table 8; Fig 2).

With regards to the computational cost of TR2-D2, we emphasize that TR2-D2 amortizes the cost of tree search by fine-tuning on the high-reward sequences stored in the replay buffer, allowing subsequent searches to sample with an increasingly optimized policy. This amortization effect substantially reduces redundant exploration and makes the overall training procedure more efficient. All DNA experiments take at most 16 GPU hours on an NVIDIA H100 GPU. For the peptide experiments, we present the GPU hours on a single NVIDIA A6000 GPU for 1000 fine-tuning epochs in Table 4, demonstrating that only a minor increase in compute time is required to obtain substantial gains in multi-objective generation performance.

Table 4: **Compute time on a single NVIDIA A6000 GPU for peptide fine-tuning with increasing number of MCTS iterations.**

| Number of MCTS Iterations | Compute Time (1 NVIDIA A6000 GPU) |
|---|---|
| $N_{\text{iter}} = 0$ (No MCTS) | 5h 52m |
| $N_{\text{iter}} = 5$ | 6h 42m |
| $N_{\text{iter}} = 20$ | 10h 1m |
| $N_{\text{iter}} = 50$ | 16h 0m |

## F.2 Ablation on Number of Fine-Tuning Epochs

As shown in Table 5, we show that the TR2-D2 outperforms PepTune across almost all objectives for each protein target with 200 epochs and 1000 epochs of fine-tuning ($N_{\text{resample}} = 20$, $N_{\text{iter}} = 20$, and $M = 50$). After 200 epochs of fine-tuning, we observe increased performance across most rewards compared to the PepTune baseline, while maintaining sequence diversity. After 1000 epochs of fine-tuning, we observe that the mean reward values plateau to optimality across all objectives but result in lower sequence diversity. We conclude that there is a trade-off between the reward optimality and sequence diversity, with $N_{\text{epochs}} = [200, 1000]$ epochs being a suitable range for multi-objective peptide sequence generation. We also note that tuning other hyperparameters can also affect the diversity of generated sequences, specifically setting $N_{\text{resample}} = 10$ instead of 20 significantly increases diversity, even after 1000 epochs.

## F.3 Hyperparameter Discussion

**Number of Children** $M$    This determines the number of partially unmasked sequences independently sampled from the fine-tuned model at the expansion step in each MCTS loop. These will become the child nodes of the expanded node at each iteration. Increasing $M$ increases the number of sequences explored at each step, which widens the optimal search space covered during buffer generation. We found that increasing the number of children improves performance across multiple objectives (Table 8).

**Number of MCTS Iterations** $N_{\text{iter}}$    This determines the number of MCTS loops of selection, expansion, rollout, and backpropagation at each buffer resampling step, where each iteration begins by selecting an optimal trajectory from the root node (fully masked sequence) to a leaf node (unexpanded partially masked sequence). If the selected leaf node is fully unmasked, the selection process restarts from the root without increasing the iteration count. Each iteration generates a new batch of $M$ sequences that could be added to the buffer. We found that even $N_{\text{iter}} = 5$ improves fine-tuning across multiple rewards, which steadily increases with larger $N_{\text{iter}}$ (Figure 4).

Table 5: **Full multi-objective peptide design results.** Target proteins include TfR, GLP-1R, GLAST, GFAP, AMHR2, and NCAM1. All values are averaged over 100 generated peptides. Best values are **bolded**. Second-best values are underlined. **Pre-trained** indicates unconditional sampling with the pre-trained peptide SMILES model from PepTune (Tang et al., 2025). **PepTune** indicates samples from 100 iterations of inference-time Monte-Carlo Tree Guidance conditioned on all objectives. **TR2-D2** indicates unconditional sampling after 200 and 1000 epochs of fine-tuning of the pre-trained model with our multi-objective fine-tuning approach. Hyperparameters are set to $N_{\text{resample}} = 20$, $N_{\text{iter}} = 20$, and $M = 50$ across all runs.

| Target Protein | Method | Binding Affinity ($\uparrow$) | Solubility ($\uparrow$) | Non-hemolysis ($\uparrow$) | Non-fouling ($\uparrow$) | Permeability ($\uparrow$) |
|---|---|---|---|---|---|---|
| TfR | Pre-trained | $8.008_{\pm 0.673}$ | $0.742_{\pm 0.166}$ | $0.874_{\pm 0.063}$ | $0.102_{\pm 0.083}$ | $-7.470_{\pm 0.120}$ |
| | PepTune | $8.216_{\pm 0.703}$ | $0.789_{\pm 0.144}$ | $0.902_{\pm 0.051}$ | $0.121_{\pm 0.081}$ | $-7.389_{\pm 0.119}$ |
| | **TR2-D2** ($N_{\text{epochs}} = 200$) | $8.959_{\pm 0.796}$ | $\underline{0.732}_{\pm 0.145}$ | $\mathbf{0.904}_{\pm \mathbf{0.038}}$ | $0.229_{\pm 0.094}$ | $-7.300_{\pm 0.067}$ |
| | **TR2-D2** ($N_{\text{epochs}} = 1000$) | $\mathbf{10.098}_{\pm \mathbf{0.050}}$ | $\mathbf{0.838}_{\pm \mathbf{0.066}}$ | $0.896_{\pm 0.012}$ | $\mathbf{0.271}_{\pm \mathbf{0.038}}$ | $\mathbf{-7.168}_{\pm \mathbf{0.024}}$ |
| GLP-1R | Pre-trained | $8.233_{\pm 0.367}$ | $0.742_{\pm 0.166}$ | $\underline{0.874}_{\pm 0.063}$ | $0.102_{\pm 0.083}$ | $-7.470_{\pm 0.120}$ |
| | PepTune | $8.403_{\pm 0.365}$ | $\underline{0.774}_{\pm 0.170}$ | $\mathbf{0.907}_{\pm \mathbf{0.057}}$ | $0.125_{\pm .082}$ | $-7.388_{\pm 0.128}$ |
| | **TR2-D2** ($N_{\text{epochs}} = 200$) | $9.059_{\pm 0.329}$ | $0.700_{\pm 0.084}$ | $0.839_{\pm 0.037}$ | $0.385_{\pm .095}$ | $\underline{-7.288}_{\pm 0.047}$ |
| | **TR2-D2** ($N_{\text{epochs}} = 1000$) | $\mathbf{9.426}_{\pm \mathbf{0.035}}$ | $\mathbf{0.841}_{\pm \mathbf{0.043}}$ | $0.849_{\pm 0.016}$ | $\mathbf{0.499}_{\pm \mathbf{0.037}}$ | $\mathbf{-7.263}_{\pm \mathbf{0.020}}$ |
| GLAST | Pre-trained | $7.830_{\pm 0.420}$ | $0.742_{\pm 0.166}$ | $0.874_{\pm 0.063}$ | $0.102_{\pm 0.083}$ | $-7.470_{\pm 0.120}$ |
| | PepTune | $8.400_{\pm 0.353}$ | $0.815_{\pm 0.139}$ | $\mathbf{0.937}_{\pm \mathbf{0.029}}$ | $0.137_{\pm 0.086}$ | $-7.311_{\pm 0.106}$ |
| | **TR2-D2** ($N_{\text{epochs}} = 200$) | $8.842_{\pm 0.274}$ | $0.822_{\pm 0.122}$ | $\underline{0.906}_{\pm 0.031}$ | $0.268_{\pm 0.086}$ | $-7.316_{\pm 0.048}$ |
| | **TR2-D2** ($N_{\text{epochs}} = 1000$) | $\mathbf{9.703}_{\pm \mathbf{0.072}}$ | $\mathbf{0.884}_{\pm \mathbf{0.038}}$ | $0.930_{\pm 0.007}$ | $\mathbf{0.364}_{\pm \mathbf{0.083}}$ | $\mathbf{-7.238}_{\pm \mathbf{0.020}}$ |
| GFAP | Pre-trained | $7.084_{\pm 0.594}$ | $0.742_{\pm 0.166}$ | $0.874_{\pm 0.063}$ | $0.102_{\pm 0.083}$ | $-7.470_{\pm 0.120}$ |
| | PepTune | $7.256_{\pm 0.704}$ | $0.807_{\pm 0.167}$ | $\mathbf{0.907}_{\pm \mathbf{0.053}}$ | $0.124_{\pm 0.088}$ | $-7.374_{\pm 0.134}$ |
| | **TR2-D2** ($N_{\text{epochs}} = 200$) | $8.539_{\pm 0.463}$ | $0.820_{\pm 0.166}$ | $0.905_{\pm 0.020}$ | $\mathbf{0.154}_{\pm \mathbf{0.043}}$ | $-7.256_{\pm 0.071}$ |
| | **TR2-D2** ($N_{\text{epochs}} = 1000$) | $\mathbf{9.762}_{\pm \mathbf{0.123}}$ | $\mathbf{0.910}_{\pm \mathbf{0.032}}$ | $0.889_{\pm 0.010}$ | $\underline{0.137}_{\pm 0.011}$ | $\mathbf{-7.196}_{\pm \mathbf{0.030}}$ |
| AMHR2 | Pre-trained | $7.958_{\pm 0.253}$ | $0.742_{\pm 0.166}$ | $0.874_{\pm 0.063}$ | $0.102_{\pm 0.083}$ | $-7.470_{\pm 0.120}$ |
| | PepTune | $8.284_{\pm 0.186}$ | $\underline{0.789}_{\pm 0.144}$ | $\mathbf{0.930}_{\pm \mathbf{0.039}}$ | $0.156_{\pm 0.074}$ | $-7.346_{\pm 0.102}$ |
| | **TR2-D2** ($N_{\text{epochs}} = 200$) | $8.532_{\pm 0.117}$ | $0.710_{\pm 0.192}$ | $0.917_{\pm 0.047}$ | $0.534_{\pm 0.144}$ | $\mathbf{-7.159}_{\pm \mathbf{0.073}}$ |
| | **TR2-D2** ($N_{\text{epochs}} = 1000$) | $\mathbf{8.595}_{\pm \mathbf{0.029}}$ | $\mathbf{0.947}_{\pm \mathbf{0.0145}}$ | $0.923_{\pm 0.008}$ | $\mathbf{0.766}_{\pm \mathbf{0.023}}$ | $\underline{-7.164}_{\pm 0.031}$ |
| NCAM1 | Pre-trained | $6.438_{\pm 0.372}$ | $0.742_{\pm 0.166}$ | $0.874_{\pm 0.063}$ | $\mathbf{0.102}_{\pm \mathbf{0.083}}$ | $-7.470_{\pm 0.120}$ |
| | PepTune | $6.916_{\pm 0.240}$ | $0.877_{\pm 0.105}$ | $\underline{0.935}_{\pm 0.039}$ | $0.090_{\pm 0.075}$ | $-7.391_{\pm 0.133}$ |
| | **TR2-D2** ($N_{\text{epochs}} = 200$) | $7.333_{\pm 0.186}$ | $0.940_{\pm 0.065}$ | $0.932_{\pm 0.047}$ | $\underline{0.086}_{\pm 0.117}$ | $-7.123_{\pm 0.088}$ |
| | **TR2-D2** ($N_{\text{epochs}} = 1000$) | $\mathbf{7.541}_{\pm \mathbf{0.025}}$ | $\mathbf{0.972}_{\pm \mathbf{0.018}}$ | $\mathbf{0.974}_{\pm \mathbf{0.003}}$ | $0.067_{\pm 0.009}$ | $\mathbf{-6.930}_{\pm \mathbf{0.028}}$ |

**Exploration Constant** $c$   This determines the scaling factor of the second term in Equation (9) that determines the degree of exploration during MCTS. We determine that $c = 0.1$ optimally balances exploration and exploitation of optimal trajectories.

**Number of Replicates for WDCE** $R$   For each batch of $B$ fully unmasked sequence sampled from the replay buffer $\{(\boldsymbol{X}_T^i, W^{u_\theta})\}_{i=1}^B$, we calculate the WDCE loss $\mathcal{L}_{\text{WDCE}}$ from (7) using $R$ independently masked versions of $\boldsymbol{X}_T^i$. First, we sample a random variables $\{\lambda_{i,r}\}_{i \in \{1,...,B\}, r \in \{1,...,R\}}$ where $\lambda_{i,r} \sim \text{Unif}(0, 1)$ and generate a set of $R$ partially masked replicates $\{\boldsymbol{X}_t^i\}_{r=1}^R$ where each $\boldsymbol{X}_t^i$ is generated by masking each token of $\boldsymbol{X}_T^i$ with probability $\lambda_{i,r}$. Since we use a log-linear masking schedule, we derive $t = \lambda_{i,r}$ and $\sigma(t) = -\log(1 - (1 - \epsilon)t)$ for input to the policy model.

**Regularization Scaling** $\alpha$   For entropy-regularized diffusion fine-tuning, the KL regularization term in (28) is scaled by a small constant $\alpha > 0$, which determines the degree to which the fine-tuned model can diverge from the pre-trained model. For the DNA enhancer experiment, we found that setting $\alpha = 0.1$ achieved superior correlation to the 0.1% highest reward sequences in the dataset, while lower alpha $\alpha = 0.01$ achieved superior reward optimization against all benchmarks, indicating that alpha has a significant role in modulating how closely the fine-tuned distribution diverges from the data distribution and pre-trained model (Table 1). In the peptide experiment, we set $\alpha = 0.1$, which maintained high validity of generated sequences while optimizing the multi-objective rewards.

**Resampling Frequency** $N_{\text{resample}}$   This determines the number of epochs between each resampling of the replay buffer with tree search. For smaller $N_{\text{resample}}$, the buffer is resampled with greater frequency and the model is trained on each buffer for a lower number of epochs. For larger $N_{\text{resample}}$, the buffer is resampled less frequently and the same replay buffer is used for training over more epochs. We found that decreasing the resampling frequency to once per 20 epochs enhanced the multi-objective rewards but resulted in a decrease in diversity in generated sequences, whereas $N_{\text{resample}} = 10$ preserved diversity while optimizing all objectives (Table 8).

**Buffer Size $B$**    The buffer size $B$ is the number of sequences stored in the replay buffer for the WDCE loss computation during fine-tuning. At each buffer resampling step, the buffer is emptied and repopulated with optimal sequences and their corresponding log-RND weights using our tree search approach. While in most fine-tuning approaches, a larger buffer improves performance, our approach enables searching for optimal sequences to add to the buffer, thus improving the quality despite smaller buffer sizes. We also note that since MCTS generates $M$ sequences at each iteration for $N_{\text{iter}}$ iterations, the maximum buffer size is $M \times N_{\text{iter}}$.

**Number of Diffusion Steps $N_{\text{steps}}$**    This is the number of unmasking steps between the fully masked sequence at `timestep = 0` and the fully unmasked sequence at `timestep = N_steps − 1`. The MDLM framework (Sahoo et al., 2024) operates in continuous time $t \in [0, 1]$ with the log-linear noise schedule, where the probability of being masked at time $t$ is given by $t$ and the total probability of being masked over time $[0, t]$ is given by $\sigma(t) = -\log(1 - (1 - \epsilon)t)$. Following the standard setup in MDLM, we set $N_{\text{steps}} = 128$.

**Top $k$ Hyperparameter**    For single-reward fine-tuning, $k$ determines the number of child nodes that are candidates during the selection step of MCTS based on their selection reward value. At each selection step, we take the `softmax` of the top-$k$ selection scores and sample the next node from the categorical distribution. We find that setting $k$ equal to the number of children $k = M$ such that all child nodes have a chance of being explored yields good performance in DNA enhancer experiments.

**Resetting the MCTS Tree**    While it is possible to maintain the same MCTS tree for multiple buffer generation steps, we found that resetting to an empty tree before each buffer generation yields the best performance. This follows from the idea that after fine-tuning, the model inherits the ability to generate the optimal sequences from the previous tree, resulting in a more optimal tree in the next buffer generation.

Table 6: **Default hyperparameters for enhancer DNA and peptide experiments.** Discussion on hyperparameter choices and ablation studies are given in App F.

| Experiment | $M$ | $N_{\text{iter}}$ | $c$ | $R$ | $\alpha$ | $N_{\text{resample}}$ | $B$ | $N_{\text{steps}}$ | $k$ |
|---|---|---|---|---|---|---|---|---|---|
| **Enhancer DNA** | 32 | 5 | 0.1/0.001 | 16 | 0.1 | 5 | 160 | 128 | $B$ |
| **Peptides** | 50 | 20 | 0.1 | 16 | 0.1 | 20 | 20 | 128 | - |

Table 7: **Ablation study for fine-tuning DNA enhancer activity.** Metrics are computed for 640 sequences over 3 random seeds. Default settings are given in Table 6.

| Method | Pred Activity (median; ↑) | ATAC-Acc (%; ↑) | 3-mer Corr (↑) | App-Log-Lik (median; ↑) |
|---|---|---|---|---|
| **TR2-D2** ($\alpha = 0.1$) | $6.56_{\pm 0.02}$ | $86.9_{\pm 1.18}$ | $0.925_{\pm 0.002}$ | $-259.4_{\pm 0.20}$ |
| **TR2-D2** ($\alpha = 0.001$) | $9.74_{\pm 0.01}$ | $99.9_{\pm 0.01}$ | $0.548_{\pm 0.001}$ | $-271.8_{\pm 0.1}$ |
| TR2-D2 w/o MCTS ($\alpha = 0.1$) | $6.00_{\pm 0.02}$ | $76.9_{\pm 1.60}$ | $0.910_{\pm 0.004}$ | $-269.9_{\pm 0.05}$ |
| TR2-D2 w/o MCTS ($\alpha = 0.001$) | $9.13_{\pm 0.02}$ | $95.1_{\pm 0.52}$ | $0.054_{\pm 0.005}$ | $-277.1_{\pm 0.20}$ |
| **Resampling Frequency $N_{\text{resample}}$** | | | | |
| $N_{\text{resample}} = 10$ ($\alpha = 0.1$) | $6.73_{\pm 0.05}$ | $80.6_{\pm 1.23}$ | $0.900_{\pm 0.002}$ | $-254.2_{\pm 0.37}$ |
| **Number of MCTS Iterations $N_{\text{iter}}$** | | | | |
| $N_{\text{iter}} = 10$ ($\alpha = 0.1$) | $6.13_{\pm 0.11}$ | $85.0_{\pm 0.8}$ | $0.922_{\pm 0.001}$ | $-260.0_{\pm 0.16}$ |
| $N_{\text{iter}} = 20$ ($\alpha = 0.1$) | $5.49_{\pm 0.03}$ | $81.9_{\pm 2.6}$ | $0.921_{\pm 0.001}$ | $-262.0_{\pm 0.20}$ |
| $N_{\text{iter}} = 30$ ($\alpha = 0.1$) | $4.91_{\pm 0.02}$ | $79.8_{\pm 0.66}$ | $0.86_{\pm 0.003}$ | $-268.6_{\pm 0.40}$ |

Table 8: **Ablation study for multi-objective fine-tuning for therapeutic peptide design for targeting Transferrin receptor (TfR).** Metrics are computed for 100 i.i.d. generated sequences from a single forward pass through the fine-tuned model. Best scores within each hyperparameter group are **bolded**. Worst scores across all runs are underlined. Default settings are defined as $N_{\text{resample}} = 10$, $N_{\text{iter}} = 20$, $M = 20$ with MCTS.

| Method | Binding Affinity ($\uparrow$) | Solubility ($\uparrow$) | Non-hemolysis ($\uparrow$) | Non-fouling ($\uparrow$) | Permeability ($\uparrow$) |
|---|---|---|---|---|---|
| TR2-D2 w/o MCTS | $9.336_{\pm 0.325}$ | $\underline{0.548}_{\pm 0.173}$ | $0.908_{\pm 0.034}$ | $0.122_{\pm 0.044}$ | $\underline{-7.323}_{\pm 0.076}$ |
| **Resampling Frequency** $N_{\text{resample}}$ | | | | | |
| $N_{\text{resample}} = 5$ | $9.238_{\pm 0.684}$ | $0.645_{\pm 0.167}$ | $0.898_{\pm 0.039}$ | $0.186_{\pm 0.105}$ | $-7.273_{\pm 0.073}$ |
| $N_{\text{resample}} = 10$ | $9.324_{\pm 0.374}$ | $0.669_{\pm 0.166}$ | $0.901_{\pm 0.039}$ | $0.133_{\pm 0.052}$ | $-7.281_{\pm 0.067}$ |
| $N_{\text{resample}} = 20$ | $\mathbf{9.958}_{\pm \mathbf{0.120}}$ | $\mathbf{0.879}_{\pm \mathbf{0.052}}$ | $\mathbf{0.930}_{\pm \mathbf{0.010}}$ | $\mathbf{0.205}_{\pm \mathbf{0.041}}$ | $\mathbf{-7.204}_{\pm \mathbf{0.037}}$ |
| **Number of MCTS Iterations** $N_{\text{iter}}$ | | | | | |
| $N_{\text{iter}} = 5$ | $\underline{8.980}_{\pm 0.811}$ | $\mathbf{0.733}_{\pm \mathbf{0.154}}$ | $\mathbf{0.930}_{\pm \mathbf{0.024}}$ | $\mathbf{0.140}_{\pm \mathbf{0.052}}$ | $-7.262_{\pm 0.070}$ |
| $N_{\text{iter}} = 20$ | $9.324_{\pm 0.374}$ | $0.669_{\pm 0.166}$ | $0.901_{\pm 0.039}$ | $0.133_{\pm 0.052}$ | $-7.281_{\pm 0.067}$ |
| $N_{\text{iter}} = 50$ | $\mathbf{9.722}_{\pm \mathbf{0.347}}$ | $0.696_{\pm 0.120}$ | $0.909_{\pm 0.030}$ | $0.095_{\pm 0.034}$ | $\mathbf{-7.227}_{\pm \mathbf{0.067}}$ |
| **Number of Children** $M$ | | | | | |
| $M = 10$ | $9.271_{\pm 0.415}$ | $0.690_{\pm 0.156}$ | $\mathbf{0.907}_{\pm \mathbf{0.035}}$ | $0.151_{\pm 0.058}$ | $-7.290_{\pm 0.062}$ |
| $M = 20$ | $9.324_{\pm 0.374}$ | $0.669_{\pm 0.166}$ | $0.901_{\pm 0.039}$ | $0.133_{\pm 0.052}$ | $-7.281_{\pm 0.067}$ |
| $M = 50$ | $\mathbf{9.355}_{\pm \mathbf{0.573}}$ | $\mathbf{0.717}_{\pm \mathbf{0.141}}$ | $\underline{0.888}_{\pm 0.056}$ | $\mathbf{0.157}_{\pm \mathbf{0.074}}$ | $\mathbf{-7.256}_{\pm \mathbf{0.070}}$ |
| **Buffer Size** $B$ | | | | | |
| $B = 5$ | $\mathbf{9.680}_{\pm \mathbf{0.452}}$ | $0.695_{\pm 0.139}$ | $\mathbf{0.909}_{\pm \mathbf{0.027}}$ | $0.075_{\pm 0.027}$ | $\mathbf{-7.202}_{\pm \mathbf{0.058}}$ |
| $B = 10$ | $9.588_{\pm 0.359}$ | $\mathbf{0.704}_{\pm \mathbf{0.136}}$ | $0.903_{\pm 0.036}$ | $\mathbf{0.168}_{\pm \mathbf{0.073}}$ | $-7.223_{\pm 0.075}$ |
| $B = 20$ | $9.496_{\pm 0.357}$ | $0.640_{\pm 0.158}$ | $0.893_{\pm 0.043}$ | $0.143_{\pm 0.074}$ | $-7.249_{\pm 0.071}$ |

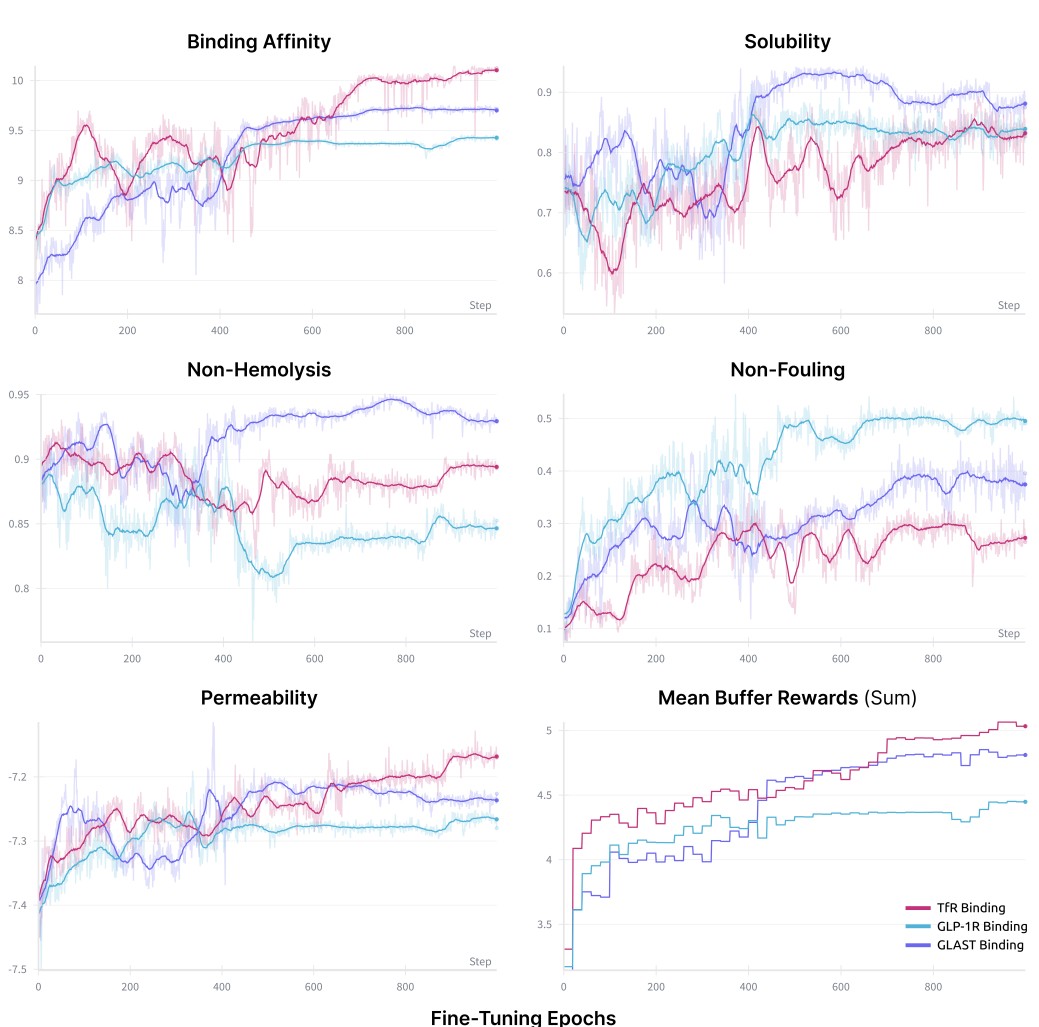

Figure 3: **Multi-objective reward curves for fine-tuning toward high binding affinity to proteins TfR, GLP-1R, and GLAST.** Average reward values of 50 sequences sampled from the fine-tuned model after each fine-tuning epoch are plotted over a total of 1000 epochs, and a running average is shown with the smooth line. The mean buffer reward is computed after every buffer resampling step (every 10 epochs). We observe that the multi-objective fine-tuning method effectively enables optimization of rewards for diverse therapeutic targets.

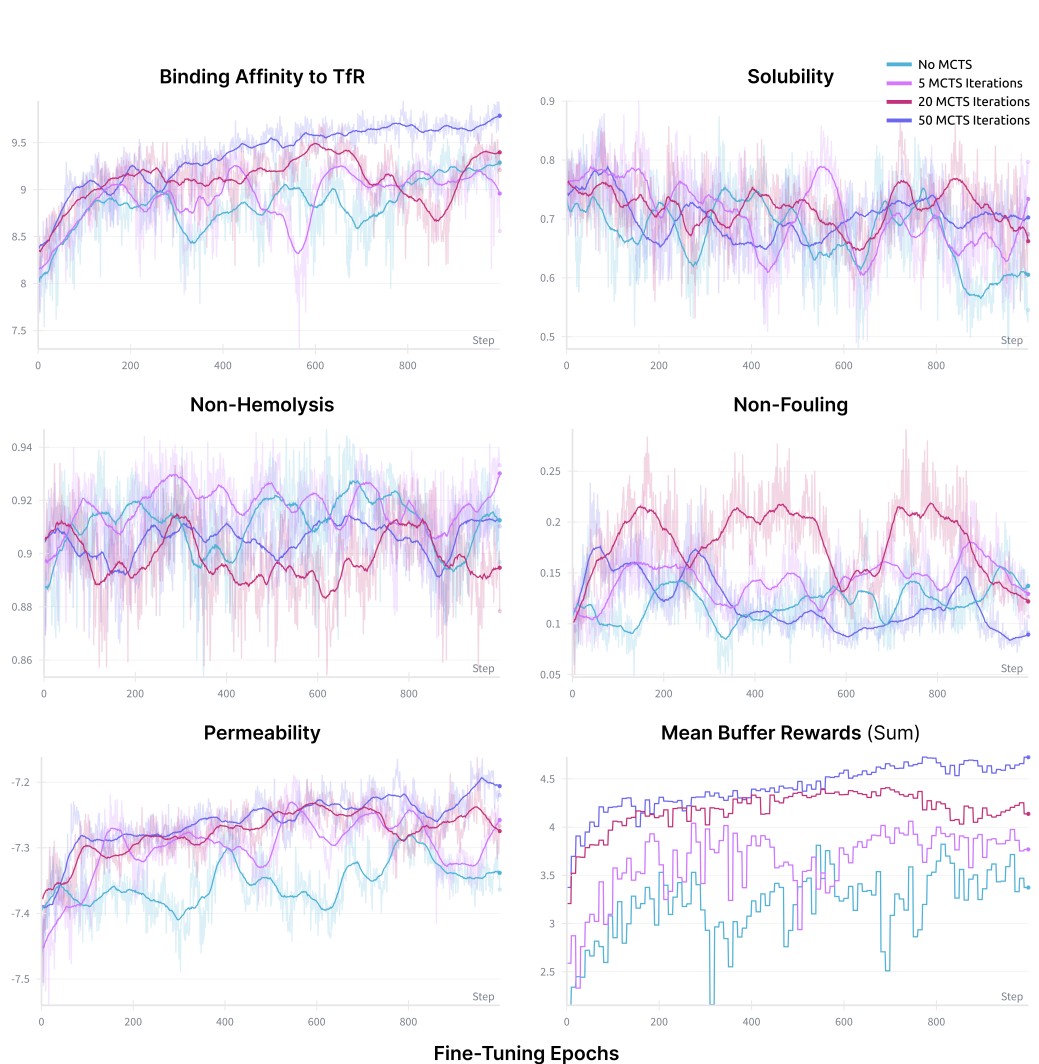

Figure 4: **Ablation study on the number of iterations of MCTS $N_{\text{iter}}$ per buffer generation step for multi-objective peptide generation.** Average reward values of 50 sequences sampled from the fine-tuned model after each fine-tuning epoch are plotted over a total of 1000 epochs, and a running average is shown with the smooth line. The mean buffer reward is computed after every buffer resampling step (every 10 epochs). We observe a steady increase in the mean rewards stored in the buffer with a larger number of iterations.

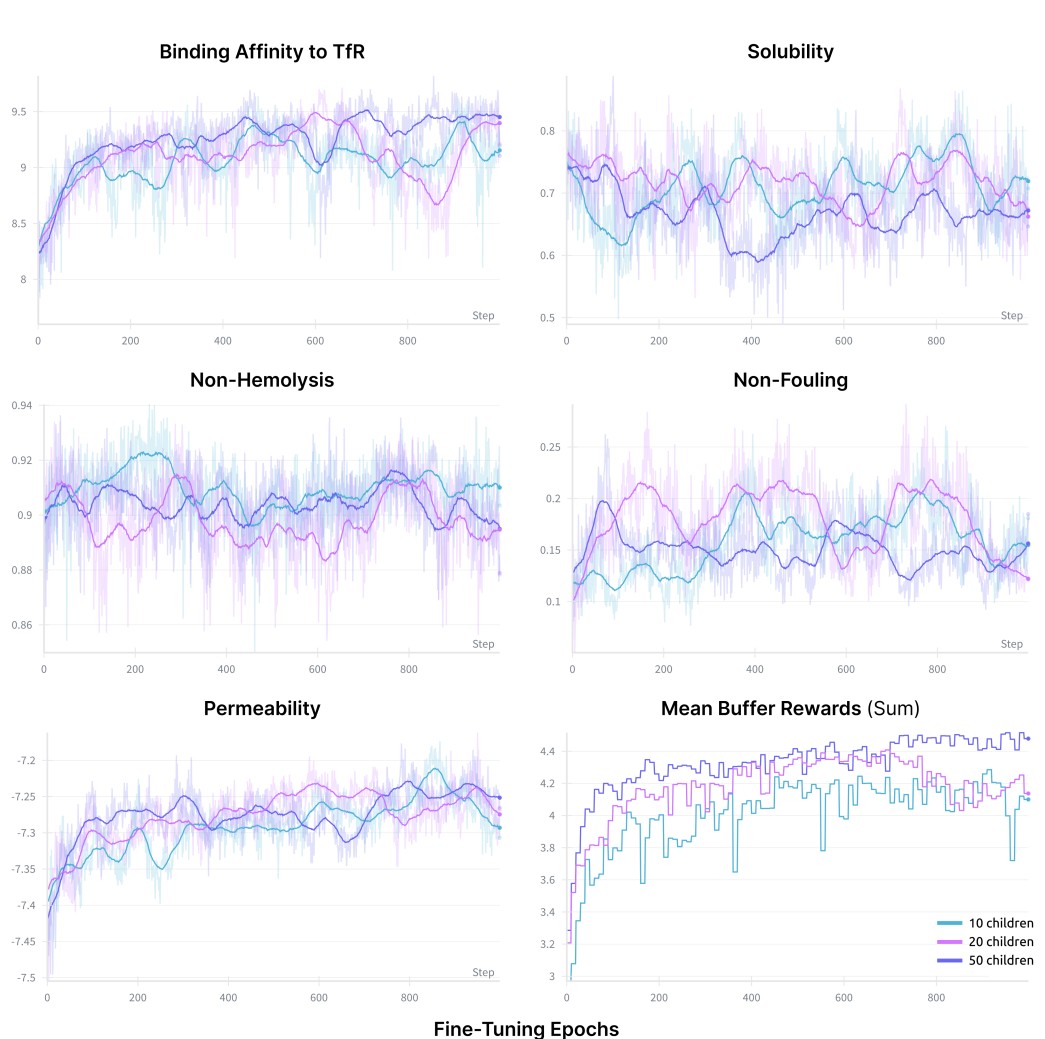

Figure 5: **Ablation study on the number of children nodes** $M$ **explored in each iteration of MCTS.** Average reward values of 50 sequences sampled from the fine-tuned model after each fine-tuning epoch are plotted over a total of 1000 epochs, and a running average is shown with the smooth line. The mean buffer reward is computed after every buffer resampling step (every 10 epochs). We observe a steady increase in the mean rewards stored in the buffer as the number of child sequences explored increases.

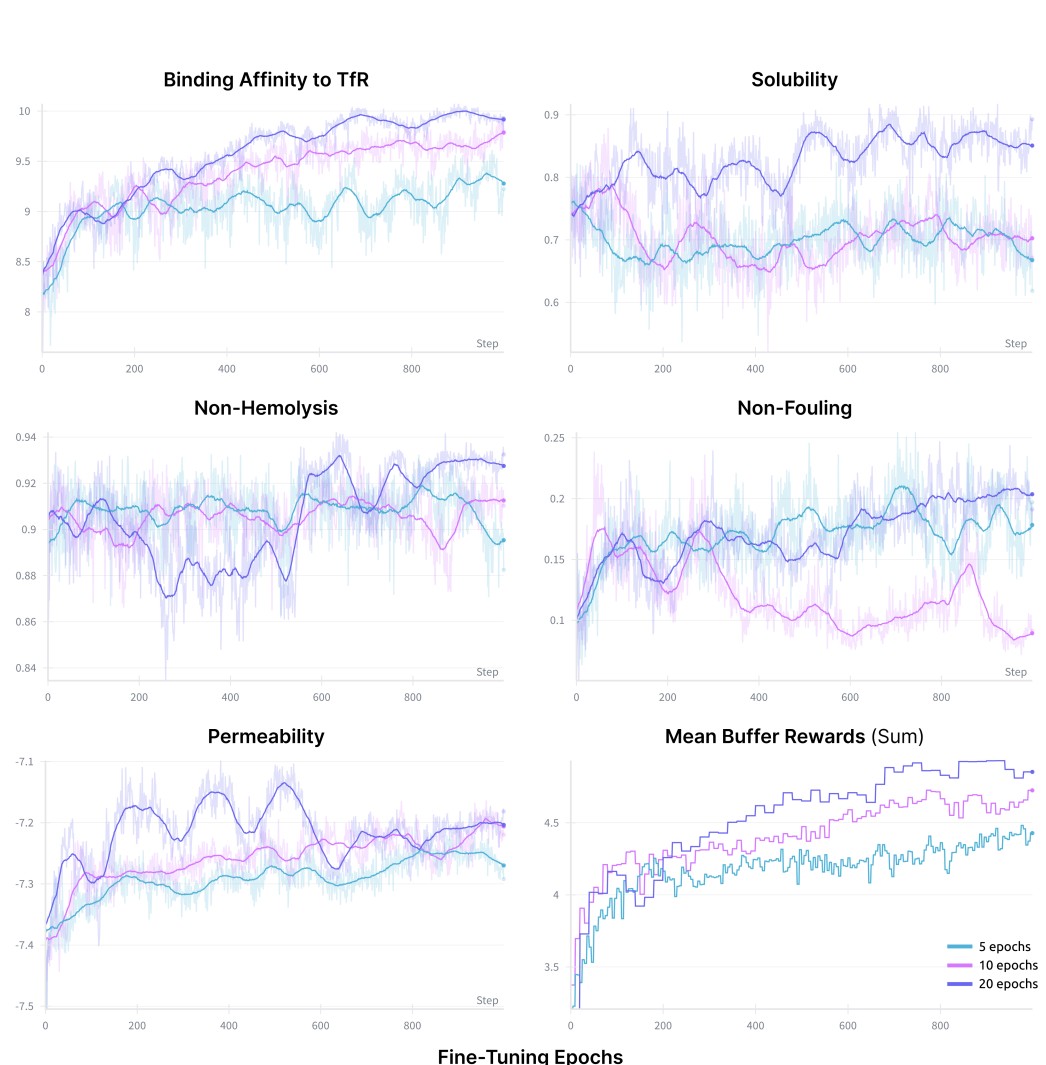

Figure 6: **Ablation study on the number of training epochs $N_{\text{resample}}$ between each buffer resampling step.** Average reward values of 50 sequences sampled from the fine-tuned model after each fine-tuning epoch are plotted over a total of 1000 epochs, and a running average is shown with the smooth line. The mean buffer reward is computed after every buffer resampling step (every 10 epochs). We observe a steady increase in the mean rewards as the $N_{\text{resample}}$ increases, indicating that the model can inherit the ability to generate high-reward sequences seen in the buffer with more training iterations.

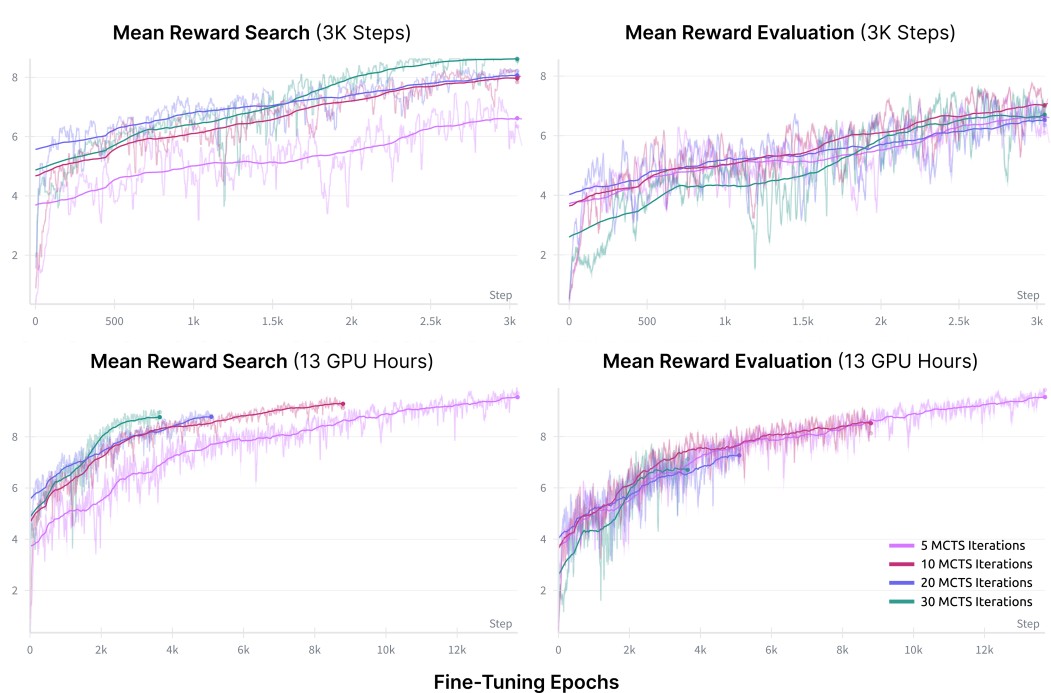

Figure 7: **Comparison of rewards after fixed fine-tuning steps (Top) and fixed compute time (Bottom) for different MCTS iterations.** The mean reward search is evaluated on samples generated during the MCTS search and the mean reward is evaluated on 50 sequences sampled from the fine-tuned model. $\alpha = 0.1$ for all experiments.

## G  ALGORITHMS

Here, we provide pseudo-code for the additional algorithms for single-reward and multi-reward fine-tuning of discrete diffusion models with **TR2-D2**. Algorithm 3 outlines the procedure for a single reverse unmasking step with log-RND tracking. Algorithm 4 describes the procedure for remasking clean samples from the replay buffer to compute the WDCE loss in (7). Algorithm 5 describes the MCTS algorithm for generating an optimal buffer $\mathcal{B}$ for the single and multi-reward case using the `Select` and `UpdateParetoFront` described in Algorithms 6 and 7, respectively.

---

**Algorithm 3** `SingleReverseStep`: Single diffusion inference step

---

1: **Input:** Partially masked sequence $\boldsymbol{X}_t$, timestep $t \in [0, 1]$, time increment $\Delta t$, pre-trained model $p^{\text{pre}}$, policy model $p^{u_\theta}$
2: $\sigma(t) \leftarrow \log(1 - (1 - \epsilon)t)$
3: `change_prob_t` $\leftarrow t$
4: `change_prob_s` $\leftarrow t - \Delta t$
5: $\log p^{u_\theta}(\cdot|\boldsymbol{X}_t) \leftarrow \texttt{Policy}(\boldsymbol{X}_t, \sigma(t))$
6: $\log p^{\text{pre}}(\cdot|\boldsymbol{X}_t) \leftarrow \texttt{pre-trained}(\boldsymbol{X}_t, \sigma(t))$
7: $q_s(\boldsymbol{X}_s|\boldsymbol{X}_t) \leftarrow p^{u_\theta}(\cdot|\boldsymbol{X}_t)(\texttt{change\_prob\_t} - \texttt{change\_prob\_s})$
8: $q_s(\boldsymbol{x}_s = \boldsymbol{M}|\boldsymbol{X}_t) \leftarrow 0$                  $\triangleright$ *zero-masking probability*
9: $\tilde{\boldsymbol{X}}_T \leftarrow \texttt{SampleCategorical}(q_s(\boldsymbol{X}_s|\boldsymbol{X}_t))$
10: $\boldsymbol{X}_s \leftarrow \tilde{\boldsymbol{X}}_T \cdot (1 - \mathbf{1}_{\boldsymbol{X}_t^\ell \neq \boldsymbol{M}}) + \boldsymbol{X}_t \cdot \mathbf{1}_{\boldsymbol{X}_t^\ell \neq \boldsymbol{M}}$
11: `log_policy` $\leftarrow \sum_{\ell: \boldsymbol{X}_s^\ell \neq \boldsymbol{X}_t^\ell} \log p^{u_\theta}(\cdot|\boldsymbol{X}_t)_{\ell, \boldsymbol{X}_s^\ell}$
12: `log_pre` $\leftarrow \sum_{\ell: \boldsymbol{X}_s^\ell \neq \boldsymbol{X}_t^\ell} \log p^{\text{pre}}(\cdot|\boldsymbol{X}_t)_{\ell, \boldsymbol{X}_s^\ell}$
13: **return** $\boldsymbol{X}_s$, `log_policy`, `log_pre`

---

---

**Algorithm 4** `ResampleWithMask`: Remasks a unmasked sequence to compute the WDCE loss

---

1: **Input:** Batch of sequences $\{\boldsymbol{X}_T^i\}_{i=1}^B$, number of replicates $R$
2: **for** $i = 1$ to $B$ **do**
3:     **for** $r = 1$ to $R$ **do**
4:        $\lambda_{i,r} \sim \text{Unif}(0, 1)$
5:        $\tilde{\boldsymbol{X}}_t^{i,r} \leftarrow \mu_\lambda(\tilde{\boldsymbol{x}}_t^\ell | \boldsymbol{X}_T^i)$           ▷ *mask each token with probability $\lambda_{i,r}$*
6:        $t \leftarrow \lambda_{i,r}$
7:     **end for**
8: **end for**
9: **return** $\{(\tilde{\boldsymbol{X}}_t^{i,r}, \lambda_{i,r})\}_{i \in \{1,\ldots,B\}, r \in \{1,\ldots R\}}$

---

---

**Algorithm 5** `MCTS`: Monte-Carlo Tree Search for Trajectory Optimization

---

1: **Input:** pre-trained model $p^{\text{pre}}(\cdot | \boldsymbol{X}_t)$, finetuned policy model $p^{u_\theta}(\cdot | \boldsymbol{X}_t)$, number of children $M$,
2: $\boldsymbol{X}_0 \leftarrow [\boldsymbol{M}]^L$
3: $\mathcal{B} \leftarrow \{\}$           ▷ *initialize empty buffer*
4: **for** `iter` in $1, \ldots, N_{\text{iter}}$ **do**
5:     `log_rnd` $\leftarrow 0$
6:     $\boldsymbol{X}_t$, `log_rnd` $\leftarrow$ `Select`$(\boldsymbol{X}_0)$           ▷ *select leaf node*
7:     $\{\boldsymbol{X}_s^i, \log p^{\text{pre}}(\boldsymbol{X}_s^i), \log p^{u_\theta}(\boldsymbol{X}_s^i)\}_{i=1}^M \leftarrow$ `BatchedReverseStep`$(\boldsymbol{X}_t)$
8:     **for** $i$ in $1, \ldots, M$ **do**           ▷ *rollout child nodes to fully unmasked*
9:        `log_rnd`$_i \leftarrow$ `log_rnd`$_i + \log p^{u_\theta}(\boldsymbol{X}_s^i) - \log p^{\text{pre}}(\boldsymbol{X}_s^i)$
10:        **for** $s$ in $\{t, \ldots, T\}$ **do**
11:           $\boldsymbol{X}_{s+\Delta t}^i$, `log_policy`$_i$, `log_pre`$_i \leftarrow$ `SingleReverseStep`$(\boldsymbol{X}_s, s)$
12:           $W^{\bar{u}}(\boldsymbol{X}_s^i) \leftarrow W^{\bar{u}}(\boldsymbol{X}_s^i) + (\text{`log\_pre`} - \text{`log\_policy`})$
13:        **end for**
14:        **if** $K > 1$ **then**           ▷ *multi-objective rewards*
15:           $W^{\bar{u}}(\boldsymbol{X}_s^i) \leftarrow W^{\bar{u}}(\boldsymbol{X}_s^i) + \frac{1}{\alpha} \sum_{k=1}^K r_k(\boldsymbol{X}_T^i)$
16:        **else**
17:           $W^{\bar{u}}(\boldsymbol{X}_s^i) \leftarrow W^{\bar{u}}(\boldsymbol{X}_s^i) + \frac{1}{\alpha} r(\boldsymbol{X}_T^i)$
18:        **end if**
19:        $\mathcal{B} \leftarrow$ `UpdateBuffer`$(\boldsymbol{X}_T^i, W^{\bar{u}}(\boldsymbol{X}_s^i))$
20:        `children`$(\boldsymbol{X}_t) \leftarrow \{\boldsymbol{X}_t^i, r(\boldsymbol{X}_T^i)\}$
21:        $R(\boldsymbol{X}_t) \leftarrow R(\boldsymbol{X}_t) + r(\boldsymbol{X}_T^i)$
22:     **end for**
23:     **while** `parent`$(\boldsymbol{X}_t)$ is not None **do**           ▷ *backpropogate*
24:        $\boldsymbol{X}^{\text{parent}} \leftarrow$ `parent`$(\boldsymbol{X}_t)$
25:        $R(\boldsymbol{X}^{\text{parent}}) \leftarrow R(\boldsymbol{X}^{\text{parent}}) + R(\boldsymbol{X}_s^i)$
26:        $N_{\text{visits}}(\boldsymbol{X}^{\text{parent}}) \leftarrow N_{\text{visits}}(\boldsymbol{X}^{\text{parent}}) + 1$
27:     **end while**
28: **end for**
29: **return**

---

---

**Algorithm 6** `Select`: Select Optimal Trajectory

---

1: **Input:** MCTS tree $\mathcal{T}$, root node $\boldsymbol{X}_0$
2: **while** True **do**
3:    **if** `children`$(\boldsymbol{X}_t)$ is not empty **and** $t \neq T$ **then**
4:       **if** $K > 1$ **then**              $\triangleright$ *multi-objective selection*
5:          $\mathcal{P}_{\text{select}} \leftarrow \{\}$              $\triangleright$ *Pareto-optimal children*
6:          **for** $\boldsymbol{X}_s^i$ in `children`$(\boldsymbol{X}_t)$ **do**
7:             $U(\boldsymbol{X}_t, \boldsymbol{X}_s^i) \leftarrow \frac{\boldsymbol{R}(\boldsymbol{X}_s^i)}{N_{\text{visits}}(\boldsymbol{X}_s^i)} + c \cdot p^{u_\theta}(\boldsymbol{X}_s^i | \boldsymbol{X}_t)\frac{\sqrt{N_{\text{visit}}(\boldsymbol{X}_t)}}{1+N_{\text{visit}}(\boldsymbol{X}_s^i)}$
8:             $\mathcal{P}_{\text{select}} \leftarrow$ `UpdateParetoFront`$(\mathcal{P}_{\text{select}}; (\boldsymbol{X}_s^i, \boldsymbol{U}(\boldsymbol{X}_t, \boldsymbol{X}_s^i)))$
9:             $\boldsymbol{X}_{\text{selected}} \sim \mathcal{P}_{\text{select}}$       $\triangleright$ *sample random child from $\mathcal{P}_{\text{select}}$*
10:          **end for**
11:       **else**
12:          `scores` $\leftarrow \{\}$
13:          **for** $\boldsymbol{X}_s^i$ in `children`$(\boldsymbol{X}_t)$ **do**
14:             $U(\boldsymbol{X}_t, \boldsymbol{X}_s^i) \leftarrow \frac{R(\boldsymbol{X}_s^i)}{M \cdot N_{\text{visits}}(\boldsymbol{X}_s^i)} + c \cdot p^{u_\theta}(\boldsymbol{X}_s^i | \boldsymbol{X}_t)\frac{\sqrt{N_{\text{visit}}(\boldsymbol{X}_t)}}{1+N_{\text{visit}}(\boldsymbol{X}_s^i)}$
15:             `scores.append`$\big(U(\boldsymbol{X}_t, \boldsymbol{X}_s^i)\big)$
16:          **end for**
17:          $\boldsymbol{X}_{\text{selected}} \sim$ `Cat`$\big(\text{softmax}(\text{top}k(\text{scores}))\big)$
18:       **end if**
19:    `Select`$(\boldsymbol{X}_{\text{selected}})$         $\triangleright$ *recursively call `Select` until expandable node*
20:    **else if** $t = 0$ **then**
21:       `Select`$(\boldsymbol{X}_0)$       $\triangleright$ *if leaf node is already fully unmasked, restart from root*
22:    **else**
23:       **return** $\boldsymbol{X}_t$         $\triangleright$ *if leaf node is expandable, return it*
24:    **end if**
25: **end while**

---

**Algorithm 7** `UpdateParetoFront`: Add sequences with Pareto-optimal reward vectors

---

1: **Input:** Current Pareto front containing unmasked sequences $\boldsymbol{X}_T^\star$ and their reward vectors $\boldsymbol{r}^\star \equiv \boldsymbol{r}(\boldsymbol{X}_T^\star)$ denoted $\mathcal{P} = \{(\boldsymbol{X}_T^\star, \boldsymbol{r}^\star)\}$, the candidate sequence and its reward vector $(\boldsymbol{X}_T^i, \boldsymbol{r}^i)$
2: **if** $\mathcal{P}$ is empty **then**
3:    $\mathcal{P} \leftarrow \{(\boldsymbol{X}_T^i, \boldsymbol{r}^i)\}$
4: **else**
5:    $\triangleright$ *if the candidate is dominated by any sequence in the set, return the set unchanged*    $\triangleleft$
6:    **for** $(\boldsymbol{X}_T^\star, \boldsymbol{r}^\star) \in \mathcal{P}$ **do**
7:       **if** `np.all`$(\boldsymbol{r}^\star \geq \boldsymbol{r}^i - \epsilon)$ and `np.any`$(\boldsymbol{r}^\star > \boldsymbol{r}^i + \epsilon)$ **then**
8:          **return** $\mathcal{P}$
9:       **end if**
10:    **end for**
11:    $\triangleright$ *initialize kept sequences with non-dominated candidate*    $\triangleleft$
12:    `keep` $\leftarrow \{(\boldsymbol{X}_T^i, \boldsymbol{r}^i)\}$
13:    $\triangleright$ *remove any sequence dominated by the candidate sequence*    $\triangleleft$
14:    **for** $(\boldsymbol{X}_T^\star, \boldsymbol{r}^\star) \in \mathcal{P}$ **do**
15:       **if** `np.all`$(\boldsymbol{r}^i \geq \boldsymbol{r}^\star - \epsilon)$ and `np.any`$(\boldsymbol{r}^i > \boldsymbol{r}^\star + \epsilon)$ **then**
16:          **continue**
17:       **end if**
18:       `keep.append`$(\boldsymbol{X}_T^\star, \boldsymbol{r}^\star)$
19:    **end for**
20:    $\mathcal{P} \leftarrow$ `keep`
21:    **return** $\mathcal{P}$
22: **end if**

---

