# OpenReview forum: "TR2-D2: Tree Search Guided Trajectory-Aware Fine-Tuning for Discrete Diffusion"
_ICLR.cc/2026/Conference — Submitted to ICLR 2026_

### Official Review · Reviewer_NznU · 2025-10-25

**Soundness:** 3
**Presentation:** 3
**Contribution:** 2
**Rating:** 4
**Confidence:** 4

**Summary:**

This paper proposes an iterative fine-tuning framework for discrete diffusion models that leverages Monte Carlo Tree Search (MCTS) to generate high-quality samples, combined with a WDCE loss to finetuning the model. The method is evaluated on biological sequence design tasks and demonstrates impressive empirical performance.

**Strengths:**

The authors provide comprehensive comparisons against many SOTA baselines for fine-tuning diffusion models in biological sequence design, with results that generally support their conclusions. The improvements are particularly striking. The motivation behind the proposed innovation is clear, and the methodology is technically sound. The paper is well-written and easy to follow. Moreover, the proposed approach can be extended to multi-objective optimization tasks, significantly enhancing its practical utility.

**Weaknesses:**

My primary concern lies in the experimental evaluation. Reinforcement learning–based fine-tuning methods are typically computationally expensive. While inference-time search techniques such as MCTS can indeed produce higher-quality samples, they come at the cost of additional computational overhead. Using MCTS-generated samples to construct datasets for fine-tuning compounds these costs. However, the paper does not provide any comparison of computational efficiency with baseline methods.

And theoretically, the more MCTS iterations you perform, the better samples you should get. However,
in Table 5, performance decreases as “N_iter” increases. This trend is also inconsistent with the results in Table 6.

Several critical hyperparameters—such as
$c,N_{\text{sample}},R,\alpha$ are not explained in terms of their selection criteria, even though they have a strong influence on performance.

In Figure 2, the curves are not clearly labeled.

In Equation (13), the $x$ should be $X$

**Questions:**

Mainly related to the weaknesses above.
Additionally, in Equation (9), why does $M$ appear in the denominator?

---

> ### Author Response · Authors · 2025-11-18
> **Author Response to Reviewer (1/3)**
>
> **Dear Reviewer NznU,**
>
> Thank you for taking the time to carefully review our paper. Please see the **general response** comment above and our revised manuscript to view the **key changes** that we have made to improve the manuscript and our responses to the common concerns raised. **Here, we highlight the changes that address each of your concerns one by one:**
>
> ---
>
> ### **Addressing Weaknesses**
>
> **W1: My primary concern lies in the experimental evaluation. Reinforcement learning–based fine-tuning methods are typically computationally expensive. While inference-time search techniques such as MCTS can indeed produce higher-quality samples, they come at the cost of additional computational overhead. Using MCTS-generated samples to construct datasets for fine-tuning compounds these costs. However, the paper does not provide any comparison of computational efficiency with baseline methods.**
>
> Thank you for pointing out the importance of efficiency and scalability. Regarding computational cost, we emphasize that TR2-D2 amortizes the cost of tree search by fine-tuning on high-reward sequences stored in the replay buffer, allowing subsequent searches to sample with an increasingly optimized policy. This amortization effect substantially reduces redundant exploration and makes the overall training procedure more efficient.
>
> Empirically, we demonstrate the significance of this amortized search effect in **Tables 1 and 2**, where both single- and multi-objective fine-tuning performance improve markedly over baselines without MCTS, even with as few as **5 and 20 MCTS iterations per buffer update**, respectively. To directly address efficiency, **Table A** below summarizes the **GPU hours required for 1000 epochs** of peptide fine-tuning under varying numbers of MCTS iterations on a single NVIDIA A6000 GPU.
>
> **Table A: Compute time on a single NVIDIA A6000 GPU for peptide fine-tuning with increasing number of MCTS iterations.**
>
> | Number of MCTS iterations | Compute Time (1 NVIDIA A6000 GPU) |
> | --- | --- |
> | **0 (No MCTS)** | 5h 52m |
> | **5** | 6h 42m |
> | **20** | 10h 1m |
> | **50** | 16h 0m |
>
> The results indicate that even **with a modest number of MCTS iterations (5-20)**, TR2-D2 achieves substantial performance gains with only a minor increase in wall-clock cost.
>
> For the regulatory DNA experiment, we maintain the same buffer size of 128 and number of diffusion steps of 128 as the baselines to ensure fair comparison. We assume that the baselines are fine-tuned until convergence to obtain the values reported in the paper. We further note that each DNA experiment takes **at most 16 GPU hours** on an NVIDIA H100 GPU. Since inference costs remain constant as a single diffusion pass with 128 steps, our evaluation ensures a fair comparison.
>
> For the multi-objective peptide experiment, we compare against the multi-objective inference-time guidance method PepTune, since there are no baselines (to our knowledge) for multi-objective fine-tuning. Since inference using the fine-tuned model requires only a **single diffusion pass with no additional classifier evaluations**, it **significantly reduces computational cost for generation** compared to inference-time guidance, which requires iterative reward optimization and classifier evaluation on sequences.

---

> ### Author Response · Authors · 2025-11-18
> **Author Response to Reviewer (2/3)**
>
> **W2: And theoretically, the more MCTS iterations you perform, the better samples you should get. However, in Table 5, performance decreases as** $N_{\text{iter}}$ **increases. This trend is also inconsistent with the results in Table 6.**
>
> In Table 7, the ablation study was done in a realistic setting where the total compute was kept fixed while varying the algorithm hyperparameter like MCTS iteration $N_{\text{iter}}$. This incurs an additional trade-off between the number of MCTS steps used in each rollout generation and the total RL fine-tuning steps completed during the use of such a fixed compute budget.
>
> For the **regulatory DNA design task,** which is relatively easy due to there being only a single objective, high-reward sequences can be discovered with only a few tree expansions. Under a fixed compute budget, it is therefore more beneficial to allocate resources toward **more RL updates with fewer MCTS iterations**, leading to the observed performance peak at a smaller number of iterations. **To demonstrate this, we add a plot showing reward curves for fixed fine-tuning steps compared to fixed compute time in Figure 7 of the revised manuscript**, demonstrating that lower MCTS iterations converge to the optimal reward faster. **We also highlight that both with and without MCTS, our novel fine-tuning approach exceeds performance against baseline fine-tuning methods, as shown in Table B below.**
>
> In contrast, Table 8 evaluates a multi-objective peptide optimization task with a more complex reward landscape. In this case, additional MCTS iterations yield better samples across all objectives, which directly improves the multi-objective performance of the policy, as shown in **Figure 4**.
>
> **Table B: Comparison of reward values for 640 sequences across 3 seeds from model fine-tuned with TR2-D2 for regulatory DNA generation.**
>
> | Method | Pred-Activity (median; ↑) |
> | --- | --- |
> | **Pre-trained** | 0.17 ± 0.04 |
> | **CG** | 3.30 ± 0.00 |
> | **SMC** | 4.15 ± 0.33 |
> | **TDS** | 4.64 ± 0.21 |
> | **CFG** | 5.04 ± 0.06 |
> | **DRAKES** | 5.61 ± 0.07 |
> | **SEPO** | 7.55 ± 0.01 |
> | **GLID2E** | 7.35 ± 0.07 |
> | **TR2-D2 w/o MCTS** | 9.13 ± 0.02 |
> | **TR2-D2** | **9.74 ± 0.01** |

---

> ### Author Response · Authors · 2025-11-18
> **Author Response to Reviewer (3/3)**
>
> **W3: Several critical hyperparameters—such as $c,N_{\text{samples}}, R, \alpha$ are not explained in terms of their selection criteria, even though they have a strong influence on performance.**
>
> We provide the explanations for **all hyperparameters in App. F, including $c, N_{\text{samples}}, R, \alpha$.** To clarify, we repeat those explanations here:
>
> > **Exploration Constant $c$:** This determines the scaling factor of the second term in Equation (9) that determines the degree of exploration during MCTS. We determine that $c = 0.1$ optimally balances exploration and exploitation of optimal trajectories.
> >
>
> > **Resampling Frequency $N\_{\text{resample}}$:** This determines the number of epochs between each resampling of the replay buffer with tree search. For smaller $N_{\text{resample}}$, the buffer is resampled with greater frequency and the model is trained on each buffer for a lower number of epochs. For larger $N_{\text{resample}}$, the buffer is resampled less frequently and the same replay buffer is used for training over more epochs. We found that decreasing the resampling frequency to once per 20 epochs enhanced the multi-objective rewards but resulted in a decrease in diversity in generated sequences, whereas $N\_{\text{resample}}=10$ preserved diversity while optimizing all objectives (Table 8).
> >
>
> > **Number of Replicates for WDCE $R$:** For each batch of $B$ fully unmasked sequence sampled from the replay buffer $\\{(\boldsymbol{X}_T^i, W^{u\_\theta})\\}\_{i=1}^B$, we calculate the WDCE loss $\mathcal{L}\_{\text{WDCE}}$ from (7) using $R$ independently masked versions of $\boldsymbol{X}\_{T}^i$. First, we sample a random variables $\\{\lambda\_{i,r}\\}\_{i \in \\{1, \dots, B\\}, r\in \\{1, \dots, R\\}}$ where $\lambda\_{i, r}\sim \text{Unif}(0,1)$ and generate a set of **$R$** partially masked replicates $\\{\boldsymbol{X}_t^i\\}\_{r=1}^R$ where each $\boldsymbol{X}_t^i$ is generated by masking each token of $\boldsymbol{X}_T^i$ with probability $\lambda\_{i, r}$. Since we use a log-linear masking schedule, we derive $t=\lambda\_{i, r}$ and $\sigma(t)=-\log (1-(1-\epsilon)t)$ for input to the policy model. We set $R=16$, which is shown to perform well for the WDCE loss in [1].
> >
>
> > **Regularization Scaling $\alpha$:** For entropy-regularized diffusion fine-tuning, the KL regularization term in (28) is scaled by a small constant $\alpha>0$, which determines the degree to which the fine-tuned model can diverge from the pre-trained model. For the DNA enhancer experiment, we found that setting $\alpha=0.1$ achieved superior correlation to the 0.1\% highest reward sequences in the dataset, while lower alpha $\alpha=0.01$ achieved superior reward optimization against all benchmarks, indicating that alpha has a significant role in modulating how closely the fine-tuned distribution diverges from the data distribution and pre-trained model (Table 1). In the peptide experiment, we set $\alpha=0.1$, which maintained high validity of generated sequences while optimizing the multi-objective rewards.
> >
>
> All remaining hyperparameters can be found in **App. F** in addition to ablation studies that evaluate the influence they have on fine-tuning performance.
>
> ---
>
> **W4: In Figure 2, the curves are not clearly labeled.**
>
> We have updated the revised manuscript with the labelled figure. The **blue curve** represents TR2-D2 with **no MCTS** and the **purple curve is TR2-D2 with 20 iterations of MCTS** at each buffer generation step.
>
> ---
>
> **W5: In Equation (13), the $\boldsymbol{x}$ should be $\boldsymbol{X}$**
>
> Thank you for catching our error, and we have fixed it in the revised manuscript.
>
> ---
>
> ### **Addressing Questions**
>
> **Q1: In Equation (9), why does $M$ appear in the denominator?**
>
> We add $M$ to the denominator because the exploration term depends on the number of visits to the **parent** and **child**, but in the case where each node can have a different number of children, nodes with more possible actions tend to accumulate higher total visit counts simply because they have more expansion opportunities, biasing exploration toward high-branching nodes rather than nodes with high reward potential.
>
> ---
>
> **Citations**
>
> [1] Zhu, Yuchen, et al. "MDNS: Masked diffusion neural sampler via stochastic optimal control." *NeurIPS 2025* (2025).
>
> ---
>
> If we have sufficiently addressed all of your concerns, we kindly ask that you support the acceptance of our paper for poster presentation, and if concerns remain, we are happy to continue the discussion!
>
> Thank you,
>
> The Authors

---

> > ### Comment · Reviewer_NznU · 2025-11-26
> >
> > Thank you for the author's reply. I think the author provided good supplementary information on the time consumption and explained the meaning of hyperparameters, which partially answered my question. After carefully reading the author's reply, I am still concerned about the MCTS part. When MCTS=0, the method modifies WDCE to an off-policy style, which is an incremental improvement in my view. Although the author's practical work is meaningful, the main argument and title of the paper still consider the combination with MCTS to be its main contribution. If my understanding is incorrect, I hope the author can provide a clear methodological distinction and comparison with WDCE. Regarding the hyperparameter part, can I assume that these hyperparameters($\alpha,N_{\text{sample}}$) cannot be used well across all tasks, and that the current method can only determine the final hyperparameters by testing its final performance? And for Eq 9, in terms of child node selection, M should be consistent for all child nodes, so M only affects the ratio between the reward term and the UCT term. Or does M refer to the number of child nodes of a child node? In that case, M should have a subscript i. I am still very confused about this and hope the author can provide further explanation.

---

> > > ### Author Response · Authors · 2025-11-27
> > > **Author Response to Follow-Up Questions (1/2)**
> > >
> > > Dear Reviewer NznU,
> > >
> > > We sincerely appreciate your thoughtful engagement with our paper and for bringing up these crucial points of clarification! We hope to clear up your remaining concerns by carefully addressing them in a point-by-point manner:
> > >
> > > ---
> > >
> > > > **When MCTS=0, the method modifies WDCE to an off-policy style, which is an incremental improvement in my view. Although the author's practical work is meaningful, the main argument and title of the paper still consider the combination with MCTS to be its main contribution. If my understanding is incorrect, I hope the author can provide a clear methodological distinction and comparison with WDCE.**
> > > >
> > >
> > > You’re correct, our main contribution is the **combination of off-policy RL fine-tuning with tree search**. We show in Table B above and Table 1 in our manuscript that while off-policy fine-tuning with WDCE outperforms prior baselines with a predicted activity of 9.13 and ATAC-Acc of 95.1, it performs even better with MCTS across **all metrics** with the highest predicted activity of 9.74 and ATAC-Acc of 99.9. We also show that TR2-D2 with MCTS performs better than without MCTS on the multi-objective peptide design task in Tables 2 and 5.
> > >
> > > Thank you for acknowledging our practical contributions, but we also want to **highlight three key methodological contributions of our approach in the following points:**
> > >
> > > 1. First of all, our work makes this insightful observation that the WDCE objective, while initially proposed for training neural samplers for discrete distributions like the Ising or Potts model (which is a completely different task), is suitable to use as a training loss for finetuning masked discrete diffusion models. We realized the high-level task equivalence between discrete neural sampler training and discrete diffusion model RL finetuning, which is a contribution of its own.
> > > 2. Another key contribution is leveraging the synergy of off-policy WDCE objective (Eq. 7) with structured search, since WDCE only requires the **log RND from the frozen policy weights to compute the importance weights**. Unlike other fine-tuning algorithms, like GRPO requires recomputing the log RND values of each sample from the tree, **TR2-D2 can re-use the samples generated from MCTS over an arbitrary number of iterations without having to recompute the log RND weights.** This saves significant computation and amortizes the additional computational overhead from the MCTS search in TR2-D2, allowing it to achieve superior performance compared to baseline methods with a similar order of compute. Our approach also overcomes the common limitations in off-policy RL methods, as detailed in App. C.2.
> > > 3. Besides demonstrating superior performance against baselines in single-objective fine-tuning, we emphasize that TR2-D2 is the **first** successful framework (to the best of our knowledge) for **multi-objective fine-tuning of discrete diffusion models** that balances tradeoffs between an arbitrary number of objectives. Instead of adopting a complicated multi-objective RL formulation, TR2-D2 leverages the perspective that WDCE loss is a weighted SFT loss, which can learn the Pareto frontier from approximately Pareto-optimal sequences obtained through multi-objective MCTS. This is an easy-to-implement, efficient approach, with efficacy that can be clearly observed in Tables 2 and 5. TR2-D2 achieves a significant improvement in multi-objective scores over **both fine-tuning without MCTS as well as inference-time multi-objective guidance.**

---

> > > > ### Author Response · Authors · 2025-11-27
> > > > **Author Response to Follow-Up Questions (2/2)**
> > > >
> > > > > **Regarding the hyperparameter part, can I assume that these hyperparameters $(\alpha, N_{\text{resample}})$ cannot be used well across all tasks, and that the current method can only determine the final hyperparameters by testing its final performance?**
> > > > >
> > > >
> > > > While the performance of fine-tuning does vary with different values of $\alpha$, this is the case for any fine-tuning method that aims to sample from the reward-tilted distribution defined in Eq. (4) as:
> > > >
> > > > $$
> > > > \mathbb{P}^{*}(\boldsymbol{X}\_{0:T}) = \mathbb{P}^{\text{pre}}(\boldsymbol{X}\_{0:T}) \frac{1}{Z} \exp\left(\frac{r(\boldsymbol{X}_T)}{\alpha}\right)
> > > > $$
> > > >
> > > > Intuitively, the $\alpha$ term determines the degree of divergence from the original pre-trained model distribution $\mathbb{P}^{\text{pre}}(\boldsymbol{X}_{0:T})$, where lower $\alpha$ results in greater divergence from the pre-trained and higher reward maximization. For tasks where divergence from the pre-trained model doesn’t significantly degrade generation quality (e.g., DNA sequences where all sequences are valid), then we show empirically that $\alpha=0.001$ gives consistently superior fine-tuning results. Whereas, for tasks like peptide generation with SMILES representation, where not all token combinations can be decoded into a valid peptide sequence, we believe that $\alpha=0.1$ maintains the generation quality while showing significantly greater reward values as demonstrated in Tables 2 and 5.
> > > >
> > > > For $N\_{\text{resample}}$, we agree that it is a hyperparameter that needs to be picked depending on task difficulty to optimally balance using each sampled buffer efficiently (larger $N\_{\text{resample}}$) and avoiding over-optimization on stale samples (smaller $N\_{\text{resample}}$). For simple tasks like regulatory DNA design (single-objective, short sequences, small vocabulary), the model can quickly capture all useful signal from each buffer, so setting $N\_{\text{resample}}$ too large can lead to overfitting and degraded performance and we find that $N_{\text{resample}}=5$ gives optimal performance. For more challenging tasks like peptide SMILES design (multi-objective, long sequences, large alphabet), each buffer contains much richer and more diverse optimization signal, so the model benefits from repeated optimization of the same buffer and larger $N\_{\text{resample}}$, and we observe an upward trend in performance for increasing $N\_{\text{resample}}\in \\{5, 10, 20\\}$ as shown in Table 8 and Figure 6. While we agree that these hyperparameters require some tuning, we believe there is sufficient intuition behind the hyperparameter choices that yield superior performance given the task.
> > > >
> > > > ---
> > > >
> > > > > **And for Eq 9, in terms of child node selection, M should be consistent for all child nodes, so M only affects the ratio between the reward term and the UCT term. Or does M refer to the number of child nodes of a child node? In that case, M should have a subscript i. I am still very confused about this and hope the author can provide further explanation.**
> > > > >
> > > >
> > > > Yes, you are correct! We apologize for the confusion and have removed $M$ from the denominator of Eq. 9 in the revised manuscript, which is consistent with our original code implementation in `/tr2d2-pep/peptide_mcts.py` found in our anonymous repository: https://anonymous.4open.science/r/TR2-D2anon.
> > > >
> > > > $N\_{\text{visits}}(X_s^i)$ fully accounts for the difference in rewards across child nodes since the number of child nodes (and corresponding rewards) generated for each visit is the same, so you are fully correct in your original assumption that $N\_{\text{visits}}(X_s^i)$ is a sufficient normalization factor that prevents biasing the search towards nodes that have been extensively visited.
> > > >
> > > > ---
> > > >
> > > > Thank you again for engaging deeply with our paper, and we hope our clarifications are helpful! If our responses have adequately addressed the raised concerns, we respectfully ask that you support our paper for acceptance.

---

### Official Review · Reviewer_fhok · 2025-10-29

**Soundness:** 3
**Presentation:** 3
**Contribution:** 3
**Rating:** 4
**Confidence:** 4

**Summary:**

This paper introduces TR2-D2, a novel framework for fine-tuning pre-trained discrete diffusion models. It is widely known that standard reinforcement learning (RL) fine-tuning can be unstable. To address this problem, TR2-D2 proposes combining MCTS-based search with off-policy RL for stable diffusion fine-tuning. The extension of this framework to multi-objective optimization is a plus, which the authors claim is the first for discrete diffusion models.

**Strengths:**

1. While off-policy learning is a known concept in RL, the core idea of using MCTS to explicitly curate a good buffer for off-policy learning, as far as I know, is new in trajectory-aware fine-tuning of diffusion models.
2. The extension to multi-objective fine-tuning, i.e., using MCTS to discover a set of Pareto-optimal trajectories, is a plus, due to the popularity of conflicting objectives in scientific applications.

**Weaknesses:**

1. As the decoupling to off-policy learning is a key methodological choice, the paper does not sufficiently describe all information related to solving the known challenges of off-policy RL (i.e., distribution mismatch, high variance of importance sampling, and hyperparameter sensitivity, especially the periodic model update frequency). Since off-policy RL is often difficult to implement, it is important to justify why this work is "uniquely" successful here.  Is the hyper-parameter selection sensitive in this method?

2. This work has a decent amount of equations. But no real theoretical insights are provided. For example, one important aspect is whether authors can provide any level of formal justifications on why the distribution mismatch can be mitigated by the MCTS buffer. Is this just for discrete diffusion, or is this just expecially useful for trajectory-aware learning?

3. Relevant to the real mechanism that makes MCTS work, one experimental weakness is revealed from the conflicting results regarding the number of MCTS iterations. For the DNA task (Table 5), larger iterations worsen enhancer Pred-Activity (this is concerning, as it suggests that MCTS may not promote learning as the authors propose, or at least that its interactions are more complex). In contrast, Figure 4 shows more iterations indeed lead to somewhat better performance (as we wish). In summary, the current version raises major concerns about the robustness of the MCTS component and suggests a highly sensitive, unexplained interaction with the reward landscape and/or learning hyperparameters. Thus, it remains confusing how practical and useful this method can be adapted elsewhere.

4. MCTS search, if applied in practice, is highly costly. The current version seems to ack an analysis of the computational overhead. If logging computational costs is tricky, it is widely acceptable to record such statistics by fixing a iteration of MCTS, e.g., fixing at 5 or 10.

**Questions:**

see above.

P.S. Table 2 shows this method is better than baselines, but it misses a key ablation to show whether this good performance indeed comes from MCTS. Please update Table 2 to include the "TR2-D2 w/o MCTS" baseline. If possible, it's crucial to ablate MCTS iterations in this experiment as well to show the empirical validity of introducing it.

---

> ### Author Response · Authors · 2025-11-18
> **Author Response to Reviewer (1/3)**
>
> **Dear Reviewer fhok,**
>
> Thank you for taking the time to carefully review our paper. Please see the **general response** comment above and our revised manuscript to view the **key changes** that we have made to improve the manuscript and our responses to the common concerns raised. **Here, we highlight the changes that address each of your concerns one by one:**
>
> ---
>
> ### **Addressing Weaknesses**
>
> **W1: As the decoupling to off-policy learning is a key methodological choice, the paper does not sufficiently describe all information related to solving the known challenges of off-policy RL (i.e., distribution mismatch, high variance of importance sampling, and hyperparameter sensitivity, especially the periodic model update frequency). Since off-policy RL is often difficult to implement, it is important to justify why this work is "uniquely" successful here. Is the hyper-parameter selection sensitive in this method?**
>
> Thank you for raising this important point of clarification. While there are known challenges with off-policy RL, we emphasize that **TR2-D2 overcomes all of these limitations**, which we provide a detailed justification for below and include in **App C.2** of the updated manuscript:
>
> **Distribution mismatch:** Our framework is rigorously derived using stochastic optimal control theory, which is theoretically guaranteed to converge to the optimal **reward-tilted distribution of the initial model $\mathbb{P}^0$ defined as $\mathbb{P}^\star\propto \frac{1}{Z}\mathbb{P}^0e^r$ with proof in App. C.1.**
>
> **Comparison between on-policy and off-policy RL:** The difference in performance of on-policy and off-policy RL depends on the objective we use. In general, if we use an objective which is computed **under the expectation of the current policy $\mathbb{E}\_{\mathbb{P}^{u_\theta}}[\cdot]$**, then on-policy will outperform off-policy due to the variance in the importance weights. However, our objective takes the expectation under **optimal policy $\mathbb{E}\_{\mathbb{P}^{\star}}[\mathcal{L}(\theta; \boldsymbol{X}_T)]$ as defined in Eq. (7)**, which is **not dependent on the current policy**. For this objective, on-policy is not necessarily better than off-policy, and off-policy is much more efficient, as it amortizes the search cost by reusing high-reward samples from the buffer without recalculating the log RND weights of the current policy.
>
> **High variance of importance sampling:** Unlike PPO/GRPO-type policy gradient methods [1, 2], which use importance sampling to debias to the current policy distribution at a token level (which suffers from high importance weight variance), **TR2-D2 uses importance sampling to debias to the optimal policy distribution at a sequence level** (where the importance weight variance are more benign). The advantage of sequence-level importance weighting is also discussed in GSPO [3], which uses sequence-level importance weight for AR-LLM finetuning, and we believe that TR2-D2 enjoys a similar benefit.
>
> **Periodic model update frequency:** Since we are not debiasing with respect to the current policy distribution, the debiased target remains fixed throughout training. Therefore, we avoid the typical concern that importance-weight variance may explode as the policy evolves because the target distribution itself is not moving. Instead, we perform **debiasing relative to an optimal, stationary target**, which in theory removes the need to frequently update the sampling policy to maintain numerical stability in the training objective. In practice, we still update the sampling policy periodically to improve sample efficiency and convergence speed, although such updates are not strictly required for stability.
>
> **We justify and provide ablations for hyperparameters in App. F, demonstrating robust performance across hyperparameter settings.**
>
> ---
>
> **W2: This work has a decent amount of equations. But no real theoretical insights are provided. For example, one important aspect is whether authors can provide any level of formal justifications on why the distribution mismatch can be mitigated by the MCTS buffer.**
>
> As mentioned previously, we provide formal proofs in App. C.1 that shows the optimized distribution is theoretically guaranteed to converge to the optimal **reward-tilted distribution $\mathbb{P}^\star\propto \frac{1}{Z}\mathbb{P}^0e^r$ with or without MCTS.** However, despite these theoretical guarantees, the performance in practice is limited by the quality of samples sampled and stored in the replay buffer, which could be limited in size due to memory constraints. Therefore, the integration with MCTS enhances the optimality of sequences in the buffer and **notably enables multi-objective reward optimization without scalarization.**

---

> ### Author Response · Authors · 2025-11-18
> **Author Response to Reviewer (2/3)**
>
> **Is this just for discrete diffusion, or is this just especially useful for trajectory-aware learning?**
>
> Although we introduce the MCTS buffer in the context of discrete diffusion, the underlying justification is more general. Any trajectory-based learning procedure that relies on importance weighting, stochastic optimal control objectives, or off-policy reinforcement learning can suffer from distribution mismatch when the reference trajectories fail to cover high-reward regions. **A search-augmented buffer directly reduces this mismatch by supplying the learner with trajectories that better approximate the support of the reward-tilted path measure**. As such, this can also apply to continuous-time diffusion, discrete flow matching, autoregressive credit-assignment methods, and broader SOC-based fine-tuning frameworks where trajectory quality governs variance and stability of the update.
>
> ---
>
> **W3: Relevant to the real mechanism that makes MCTS work, one experimental weakness is revealed from the conflicting results regarding the number of MCTS iterations. [...] Thus, it remains confusing how practical and useful this method can be adapted elsewhere.**
>
> In Table 7, the ablation study was done in a realistic setting where the total compute was kept fixed while varying the algorithm hyperparameter like MCTS iteration $N_{\text{iter}}$. This incurs an additional trade-off between the number of MCTS steps used in each rollout generation and the total RL fine-tuning steps completed during the use of such a fixed compute budget.
>
> For the **regulatory DNA design task,** which is relatively easy due to there being only a single objective, high-reward sequences can be discovered with only a few tree expansions. Under a fixed compute budget, it is therefore more beneficial to allocate resources toward **more RL updates with fewer MCTS iterations**, leading to the observed performance peak at a smaller number of iterations. **To demonstrate this, we add a plot showing reward curves for fixed fine-tuning steps compared to fixed compute time in Figure 7 of the revised manuscript**, demonstrating that lower MCTS iterations converge to the optimal reward faster. **We also highlight that both with and without MCTS, our novel fine-tuning approach exceeds performance against baseline fine-tuning methods, as shown in Table A below.**
>
> In contrast, Table 8 evaluates a multi-objective peptide optimization task with a more complex reward landscape. In this case, additional MCTS iterations yield better samples across all objectives, which directly improves the multi-objective performance of the policy, as shown in **Figure 4**.
>
> We conclude that for single-objective tasks, fewer iterations of MCTS are sufficient, and for multi-objective tasks, larger iterations of MCTS will significantly enhance performance.
>
> **Table A: Comparison of reward values for 640 sequences across 3 seeds from model fine-tuned with TR2-D2 for regulatory DNA generation.**
>
> | Method | Pred-Activity (median; ↑) |
> | --- | --- |
> | **Pre-trained** | 0.17 ± 0.04 |
> | **CG** | 3.30 ± 0.00 |
> | **SMC** | 4.15 ± 0.33 |
> | **TDS** | 4.64 ± 0.21 |
> | **CFG** | 5.04 ± 0.06 |
> | **DRAKES** | 5.61 ± 0.07 |
> | **SEPO** | 7.55 ± 0.01 |
> | **GLID2E** | 7.35 ± 0.07 |
> | **TR2-D2 w/o MCTS** | 9.13 ± 0.02 |
> | **TR2-D2** | **9.74 ± 0.01** |

---

> ### Author Response · Authors · 2025-11-18
> **Author Response to Reviewer (3/3)**
>
> **W4: MCTS search, if applied in practice, is highly costly. The current version seems to lack an analysis of the computational overhead.**
>
> Thank you for pointing out the importance of efficiency and scalability. Regarding computational cost, we emphasize that TR2-D2 amortizes the cost of tree search by fine-tuning on high-reward sequences stored in the replay buffer, allowing subsequent searches to sample with an increasingly optimized policy. This amortization effect substantially reduces redundant exploration and makes the overall training procedure more efficient.
>
> Empirically, we demonstrate the significance of this amortized search effect in **Tables 1 and 2**, where both single- and multi-objective fine-tuning performance improve markedly over baselines without MCTS, even with as few as **5 and 20 MCTS iterations per buffer update**, respectively. To directly address efficiency, **Table B** below summarizes the **GPU hours required for 1000 epochs** of peptide fine-tuning under varying numbers of MCTS iterations on a single NVIDIA A6000 GPU.
>
> **Table B: Compute time on a single NVIDIA A6000 GPU for peptide fine-tuning with increasing number of MCTS iterations.**
>
> | Number of MCTS iterations | Compute Time (1 NVIDIA A6000 GPU) |
> | --- | --- |
> | **0 (No MCTS)** | 5h 52m |
> | **5** | 6h 42m |
> | **20** | 10h 1m |
> | **50** | 16h 0m |
>
> The results indicate that even **with a modest number of MCTS iterations (5-20)**, TR2-D2 achieves substantial performance gains with only a minor increase in wall-clock cost.
>
> For the regulatory DNA experiment, we maintain the same buffer size of 128 and number of diffusion steps of 128 as the baselines to ensure fair comparison. We assume that the baselines are fine-tuned until convergence to obtain the values reported in the paper. We further note that each DNA experiment takes **at most 16 GPU hours** on an NVIDIA H100 GPU. Since inference costs remain constant as a single diffusion pass with 128 steps, our evaluation ensures a fair comparison.
>
> For the multi-objective peptide experiment, we compare against the multi-objective inference-time guidance method PepTune, since there are no baselines (to our knowledge) for multi-objective fine-tuning. Since inference using the fine-tuned model requires only a **single diffusion pass with no additional classifier evaluations**, it **significantly reduces computational cost for generation** compared to inference-time guidance, which requires iterative reward optimization and classifier evaluation on sequences.
>
> ---
>
> ### **Addressing Questions**
>
> **Q1: Table 2 shows this method is better than baselines, but it misses a key ablation to show whether this good performance indeed comes from MCTS. Please update Table 2 to include the "TR2-D2 w/o MCTS" baseline.**
>
> We appreciate your suggestion and agree that including this baseline in the main tables is critical for clarity. We have added this to Tables 1 and 2 in the revised manuscript to compare against TR2-D2 without MCTS.
>
> ---
>
> **Q2: If possible, it's crucial to ablate MCTS iterations in this experiment as well to show the empirical validity of introducing it.**
>
> This ablation study has already been provided in **Table 8 and Figure 4 in Appendix F of the submitted paper**, where we compare MCTS iterations of $N_{\text{iter}}\in \\{5, 20,50\\}$ and show that all iterations improve multi-objective performance against the baseline finetuning with scalarized reward, which we show below for clarity. **We also show visually in Figure 4 that increasing the number of MCTS iterations increases multi-objective scores.** We show the results in **Table C** below for your reference:
>
> **Table C: Ablation study for multi-objective fine-tuning for therapeutic peptide design for targeting Transferrin receptor (TfR). Metrics are computed for 100 i.i.d. generated sequences from a single forward pass through the fine-tuned model.**
>
> | **Number of MCTS Iterations** | **Binding Affinity** | **Solubility** | **Non-hemolysis** | **Non-fouling** | **Permeability** |
> | --- | --- | --- | --- | --- | --- |
> | **0 (No MCTS)** | 9.336 ± 0.325 | 0.548 ± 0.173 | 0.908 ± 0.034 | 0.122 ± 0.044 | -7.323 ± 0.076 |
> | **5** | 8.980 ± 0.811 | 0.733 ± 0.154 | 0.930 ± 0.024 | 0.140 ± 0.052 | -7.262 ± 0.070 |
> | **20** | 9.324 ± 0.374 | 0.669 ± 0.166 | 0.901 ± 0.039 | 0.133 ± 0.052 | -7.281 ± 0.067 |
> | **50** | 9.722 ± 0.347 | 0.696 ± 0.120 | 0.909 ± 0.030 | 0.095 ± 0.034 | -7.227 ± 0.067 |
>
> ---
>
> **Citations:**
>
> [1] Schulman, John, et al. "Proximal policy optimization algorithms."  (2017).
>
> [2] Shao, Zhihong, et al. "Deepseekmath: Pushing the limits of mathematical reasoning in open language models." (2024).
>
> [3] Zheng, Chujie, et al. "Group sequence policy optimization." (2025).
>
> ---
>
> If we have sufficiently addressed all of your concerns, we kindly ask that you support the acceptance of our paper for poster presentation, and if concerns remain, we are happy to continue the discussion!
>
> Thank you,
>
> The Authors

---

### Official Review · Reviewer_hz8y · 2025-11-01

**Soundness:** 2
**Presentation:** 3
**Contribution:** 1
**Rating:** 4
**Confidence:** 4

**Summary:**

The authors propose TR2-D2, a tree-search guided trajectory-aware fine-tuning method for discrete diffusion models. TR2-D2 utilizes Monte Carlo Tree Search to generate replay buffers that are subsequently used for off-policy fine-tuning. The authors showcase the effectiveness of TR2-D2 on single and multi-objective optimization tasks of biological sequence diffusion models.

**Strengths:**

- The idea to adapt MCTS to construct a replay buffer for the off-policy RL method of discrete diffusion models is interesting.
- Experiments across both single-objective and multi-objective settings demonstrate the empirical effectiveness of TR2-D2.

**Weaknesses:**

- TR2-D2 seems to be a simple combination of two existing methods: the widely applied MCTS algorithm and the MDNS algorithm [1] for off-policy RL of discrete diffusion models, which limits its novelty. Also, combining searching algorithms like MCTS with RL algorithms is not novel and has been applied for, e.g., offline model-based RL [2].
- The computational cost of the MCTS step is intensive. Therefore, a fairer comparison with the baselines should also take the computational cost into account. For example, it would be helpful to compare model performance under the same computational budget and the performance scaling curve of TR2-D2 as the computation increases.
- The comparison to the MDNS baseline is only provided in the appendix and not in the main table. Also, an ablation comparing the combination of different searching algorithms and different off-policy RL algorithms is missing.
- TR2-D2 utilizes an off-policy RL algorithm, which is known to have off-policyness issue and leads to inferior performance than the on-policy RL algorithms. Does TR2-D2 suffer from this issue? How does the size of the replay buffer B affect the performance of TR2-D2?



[1] MDNS: Masked Diffusion Neural Sampler via Stochastic Optimal Control. NeurIPS 2025.

[2] Bayes Adaptive Monte Carlo Tree Search for Offline Model-based Reinforcement Learning. arxiv 2024.

**Questions:**

Please refer to the **Weaknesses** section.

---

> ### Author Response · Authors · 2025-11-18
> **Author Response to Reviewer (1/3)**
>
> **Dear Reviewer hz8y,**
>
> Thank you for taking the time to carefully review our paper. Please see the **general response** comment above and our revised manuscript to view the **key changes** that we have made to improve the manuscript and our responses to the common concerns raised. **Here, we highlight the changes that address each of your concerns one by one:**
>
> ---
>
> ### **Addressing Weaknesses**
>
> **W1: TR2-D2 seems to be a simple combination of two existing methods: the widely applied MCTS algorithm and the MDNS algorithm [1] for off-policy RL of discrete diffusion models, which limits its novelty. Also, combining searching algorithms like MCTS with RL algorithms is not novel and has been applied for, e.g., offline model-based RL [2].**
>
> While TR2-D2 builds on MCTS and MDNS algorithms, we emphasize that the **key novelty lies in the following points:**
>
> 1. While the MDNS algorithm leverages the WDCE loss for training neural samplers with discrete diffusion, we show the **first implementation of the WDCE loss for off-policy discrete diffusion fine-tuning.** Notably, we show that our novel fine-tuning approach exceeds performance against baseline fine-tuning methods even **without MCTS, as shown in Table A below.**
>
>     **Table A: Comparison of reward values for 640 sequences across 3 seeds from model fine-tuned with TR2-D2 for regulatory DNA generation.**
>
>     | Method | Pred-Activity (median; ↑) |
>     | --- | --- |
>     | **Pre-trained** | 0.17 ± 0.04 |
>     | **CG** | 3.30 ± 0.00 |
>     | **SMC** | 4.15 ± 0.33 |
>     | **TDS** | 4.64 ± 0.21 |
>     | **CFG** | 5.04 ± 0.06 |
>     | **DRAKES** | 5.61 ± 0.07 |
>     | **SEPO** | 7.55 ± 0.01 |
>     | **GLID2E** | 7.35 ± 0.07 |
>     | **TR2-D2 w/o MCTS** | 9.13 ± 0.02 |
>     | **TR2-D2** | **9.74 ± 0.01** |
> 2. Our key contribution lies in the synergy of off-policy learning WDCE objective (Eq. 7) with structured search, since WDCE only requires the **log RND from the frozen policy weights to compute the importance weights** and remasks the optimized sequences from the search-optimized buffer to compute the log-likelihood under the **current model**. Unlike other fine-tuning algorithms like GRPO, which uses the exact likelihood of full trajectories under the current model and requires recomputing the log RND values of each sample from the tree, **TR2-D2 can re-use the samples generated from MCTS over an arbitrary number of iterations without having to recompute the log RND weights.**
> 3. To the best of our knowledge, we show the **first** successful framework for **multi-objective fine-tuning of discrete diffusion models** that balances tradeoffs between an arbitrary number of objectives. Our results demonstrate that this performs significantly better than the state-of-the-art inference-time multi-objective guidance approach PepTune while significantly reducing inference cost with optimal sequences generated via only a single diffusion pass. This has significant implications, particularly for biological applications where we aim to iteratively improve our model’s performance on many, often conflicting properties.

---

> > ### Author Response · Authors · 2025-11-18
> > **Author Response to Reviewer (2/3)**
> >
> > **W2: The computational cost of the MCTS step is intensive. Therefore, a fairer comparison with the baselines should also take the computational cost into account. For example, it would be helpful to compare model performance under the same computational budget and the performance scaling curve of TR2-D2 as the computation increases.**
> >
> > Thank you for pointing out the importance of efficiency and scalability. Regarding computational cost, we emphasize that TR2-D2 amortizes the cost of tree search by fine-tuning on high-reward sequences stored in the replay buffer, allowing subsequent searches to sample with an increasingly optimized policy. This amortization effect substantially reduces redundant exploration and makes the overall training procedure more efficient.
> >
> > Empirically, we demonstrate the significance of this amortized search effect in **Tables 1 and 2**, where both single- and multi-objective fine-tuning performance improve markedly over baselines without MCTS, even with as few as **5 and 20 MCTS iterations per buffer update**, respectively. To directly address efficiency, **Table B** below summarizes the **GPU hours required for 1000 epochs** of peptide fine-tuning under varying numbers of MCTS iterations on a single NVIDIA A6000 GPU.
> >
> > **Table B: Compute time on a single NVIDIA A6000 GPU for peptide fine-tuning with increasing number of MCTS iterations.**
> >
> > | Number of MCTS iterations | Compute Time (1 NVIDIA A6000 GPU) |
> > | --- | --- |
> > | **0 (No MCTS)** | 5h 52m |
> > | **5** | 6h 42m |
> > | **20** | 10h 1m |
> > | **50** | 16h 0m |
> >
> > The results indicate that even **with a modest number of MCTS iterations (5-20)**, TR2-D2 achieves substantial performance gains with only a minor increase in wall-clock cost.
> >
> > For the regulatory DNA experiment, we maintain the same buffer size of 128 and number of diffusion steps of 128 as the baselines to ensure fair comparison. We assume that the baselines are fine-tuned until convergence to obtain the values reported in the paper. We further note that each DNA experiment takes **at most 16 GPU hours** on an NVIDIA H100 GPU. Since inference costs remain constant as a single diffusion pass with 128 steps, our evaluation ensures a fair comparison.
> >
> > For the multi-objective peptide experiment, we compare against the multi-objective inference-time guidance method PepTune, since there are no baselines (to our knowledge) for multi-objective fine-tuning. Since inference using the fine-tuned model requires only a **single diffusion pass with no additional classifier evaluations**, it **significantly reduces computational cost for generation** compared to inference-time guidance, which requires iterative reward optimization and classifier evaluation on sequences.
> >
> > ---
> >
> > **W3: The comparison to the MDNS baseline is only provided in the appendix and not in the main table.**
> >
> > Thank you for the suggestion, and we have added the ablation with **no MCTS to Table 1 and 2 of the revised manuscript.**
> >
> > ---
> >
> > **W3: Also, an ablation comparing the combination of different searching algorithms and different off-policy RL algorithms is missing.**
> >
> > We appreciate your suggestion. However, TR2-D2 is designed to leverage the synergy of the WDCE objective with structured search, since WDCE only requires the **log RND from the frozen policy weights to compute the importance weights** and remasks the optimized sequences from the search-optimized buffer to compute the log-likelihood under the **current model**.
> >
> > Unlike other fine-tuning algorithms like GRPO, which uses the exact likelihood of full trajectories under the current model and requires recomputing the log RND values of each sample from the tree, **TR2-D2 can re-use the samples generated from MCTS over an arbitrary number of iterations without having to recompute the log RND weights**.
> >
> > While we have considered several other tree search algorithms (e.g. best-first search, beam search, value-guided rollouts), we believe MCTS is best suited for tasks where there is a reward function that can only reliably be evaluated on the terminal fully unmasked sequence. Best-first search and beam search require reliable intermediate-node scores before rollout and are deterministic, making it unsuitable for biological design tasks where rewards are noisy and exploration is needed to converge to the optimal distribution. Value-guided rollouts require learning a value function that accurately scores intermediate nodes, further increasing computational cost.

---

> ### Author Response · Authors · 2025-11-18
> **Author Response to Reviewer (3/3)**
>
> **W4: TR2-D2 utilizes an off-policy RL algorithm, which is known to have off-policy issues and leads to inferior performance than the on-policy RL algorithms. Does TR2-D2 suffer from this issue?**
>
> Thank you for raising this important point of clarification. While there are known challenges with off-policy RL, we emphasize that **TR2-D2 overcomes all of these limitations**, which we provide a detailed justification for below and include in **App C.2** of the updated manuscript:
>
> **Distribution mismatch:** Our framework is rigorously derived using stochastic optimal control theory, which is theoretically guaranteed to converge to the optimal **reward-tilted distribution of the initial model $\mathbb{P}^0$ defined as $\mathbb{P}^\star\propto \frac{1}{Z}\mathbb{P}^0e^r$ with proof in App. C.1.**
>
> **Comparison between on-policy and off-policy RL:** The difference in performance of on-policy and off-policy RL depends on the objective we use. In general, if we use an objective which is computed **under the expectation of the current policy $\mathbb{E}\_{\mathbb{P}^{u_\theta}}[\cdot]$**, then on-policy will outperform off-policy due to the variance in the importance weights. However, our objective takes the expectation under **optimal policy $\mathbb{E}\_{\mathbb{P}^{\star}}[\mathcal{L}(\theta; \boldsymbol{X}_T)]$ as defined in Eq. (7)**, which is **not dependent on the current policy**. For this objective, on-policy is not necessarily better than off-policy, and off-policy is much more efficient, as it amortizes the search cost by reusing high-reward samples from the buffer without recalculating the log RND weights of the current policy.
>
> **High variance of importance sampling:** Unlike PPO/GRPO-type policy gradient methods [1, 2], which use importance sampling to debias to the current policy distribution at a token level (which suffers from high importance weight variance), **TR2-D2 uses importance sampling to debias to the optimal policy distribution at a sequence level** (where the importance weight variance are more benign). The advantage of sequence-level importance weighting is also discussed in GSPO [3], which uses sequence-level importance weight for AR-LLM finetuning, and we believe that TR2-D2 enjoys a similar benefit.
>
> **Periodic model update frequency:** Since we are not debiasing with respect to the current policy distribution, the debiased target remains fixed throughout training. Therefore, we avoid the typical concern that importance-weight variance may explode as the policy evolves because the target distribution itself is not moving. Instead, we perform **debiasing relative to an optimal, stationary target**, which in theory removes the need to frequently update the sampling policy to maintain numerical stability in the training objective. In practice, we still update the sampling policy periodically to improve sample efficiency and convergence speed, although such updates are not strictly required for stability.
>
> ---
>
> **How does the size of the replay buffer B affect the performance of TR2-D2?**
>
> To answer your question, we conduct an ablation study on multi-objective fine-tuning for peptide design on different buffer sizes $B\in \\{5, 10, 20\\}$ and provide results on multi-objective scores in **Table C below**.
>
> Notably, we observe that all buffer sizes for TR2-D2 **with MCTS outperform TR2-D2 without MCTS with buffer size 20 in multi-objective optimization, indicating the robustness of MCTS in across all settings where buffer size is limited.** These results are included in Table 8 of the revised manuscript.
>
> **Table C: Ablation study on buffer size for multi-objective fine-tuning for therapeutic peptide design for targeting Transferrin receptor (TfR).** Scores are evaluated on 100 sequences generated with the fine-tuned model. Best values are **bolded**.
>
> | **Buffer size B** | **Binding Affinity** | **Solubility** | **Non-hemolysis** | **Non-fouling** | **Permeability** |
> | --- | --- | --- | --- | --- | --- |
> | **TR2-D2 w/o MCTS (B = 20)** | 9.336 ± 0.325 | 0.548 ± 0.173 | 0.908 ± 0.034 | 0.122 ± 0.044 | -7.323 ± 0.076 |
> | **B = 5** | **9.680 ± 0.452** | 0.695 ± 0.139 | **0.909 ± 0.027** | 0.075 ± 0.027 | **-7.202 ± 0.058** |
> | **B = 10** | 9.588 ± 0.359 | **0.704 ± 0.136** | 0.903 ± 0.036 | **0.168 ± 0.073** | -7.223 ± 0.075 |
> | **B = 20** | 9.496 ± 0.357 | 0.640 ± 0.158 | 0.893 ± 0.043 | 0.143 ± 0.074 | -7.249 ± 0.071 |
>
> ---
>
> **Citations:**
>
> [1] Schulman, John, et al. "Proximal policy optimization algorithms." (2017).
>
> [2] Shao, Zhihong, et al. "Deepseekmath: Pushing the limits of mathematical reasoning in open language models." (2024).
>
> [3] Zheng, Chujie, et al. "Group sequence policy optimization." (2025).
>
> ---
>
> If we have sufficiently addressed all of your concerns, we kindly ask that you support the acceptance of our paper for poster presentation, and if concerns remain, we are happy to continue the discussion!
>
> Thank you,
>
> The Authors

---

### Official Review · Reviewer_Kyq1 · 2025-11-01

**Soundness:** 3
**Presentation:** 2
**Contribution:** 2
**Rating:** 2
**Confidence:** 4

**Summary:**

This paper focuses on reward-oriented fine-tuning for discrete diffusion models. Addressing the limitations of traditional reinforcement learning fine-tuning methods—which rely on the current model to generate trajectories and are prone to reinforcing suboptimal trajectories—it proposes the TR²-D² framework. This approach employs Monte Carlo Tree Search (MCTS) to generate high-reward trajectories, constructs a replay buffer, and then combines a stochastic optimal control objective to perform trajectory-aware fine-tuning on pre-trained discrete diffusion models.

**Strengths:**

Precise targeting of the core issue in traditional RL fine-tuning—discrete diffusion (leading to suboptimal reinforcement due to reliance on current model trajectories)—enables MCTS to proactively explore high-reward trajectories. This fundamentally enhances training data quality, preventing model “learning bias,” forming a logically closed-loop system with high innovation.
Using continuous-time Markov chains (CTMC) as the theoretical framework, we ensure the method's mathematical rigor, rather than merely engineering improvements.

**Weaknesses:**

My major concern lies in the novelty of this paper, given (1) CTMC widely used in RL domain and (2) existing works have already demonstrated that the diffusion model can serve as the policy in the context of RL. Adopting RL-related search strategies to diffusion model domain seems ad-hoc.
Regarding the methods itself, I doubt the stability and the local optima problem the method may face. Please refer to questions to see how to get through these problems.

**Questions:**

I understand that Monte Carlo tree search may become stuck in local optima. If this occurs, how can we handle it to prevent the generated sequence from falling into such situations?

Does it depend on the initial model? If the initial model performs poorly, is this fine-tuning method still effective?

---

> ### Author Response · Authors · 2025-11-18
> **Author Response to Reviewer (1/2)**
>
> **Dear Reviewer Kyq1,**
>
> Thank you for taking the time to carefully review our paper. Please see the **general response** comment above and our revised manuscript to view the **key changes** that we have made to improve the manuscript and our responses to the common concerns raised. **Here, we highlight the changes that address each of your concerns one by one:**
>
> ---
>
> ### **Addressing Weaknesses**
>
> **W1: My major concern lies in the novelty of this paper, given (1) CTMC widely used in RL domain and (2) existing works have already demonstrated that the diffusion model can serve as the policy in the context of RL. Adopting RL-related search strategies to diffusion model domain seems ad-hoc.**
>
> We emphasize that the **key novelty lies in the following points:**
> 1. While the MDNS algorithm leverages the WDCE loss for training neural samplers with discrete diffusion, we show the **first implementation of the WDCE loss for off-policy discrete diffusion fine-tuning.** Notably, we show that our novel fine-tuning approach exceeds performance against baseline fine-tuning methods even **without MCTS, as shown in Table A below.**
>
>     **Table A: Comparison of reward values for 640 sequences across 3 seeds from model fine-tuned with TR2-D2 for regulatory DNA generation.**
>
>     | Method | Pred-Activity (median; ↑) |
>     | --- | --- |
>     | **Pre-trained** | 0.17 ± 0.04 |
>     | **CG** | 3.30 ± 0.00 |
>     | **SMC** | 4.15 ± 0.33 |
>     | **TDS** | 4.64 ± 0.21 |
>     | **CFG** | 5.04 ± 0.06 |
>     | **DRAKES** | 5.61 ± 0.07 |
>     | **SEPO** | 7.55 ± 0.01 |
>     | **GLID2E** | 7.35 ± 0.07 |
>     | **TR2-D2 w/o MCTS** | 9.13 ± 0.02 |
>     | **TR2-D2** | **9.74 ± 0.01** |
> 2. Our key contribution lies in the synergy of off-policy learning WDCE objective (Eq. 7) with structured search, since WDCE only requires the **log RND from the frozen policy weights to compute the importance weights** and remasks the optimized sequences from the search-optimized buffer to compute the log-likelihood under the **current model**. Unlike other fine-tuning algorithms like GRPO, which uses the exact likelihood of full trajectories under the current model and requires recomputing the log RND values of each sample from the tree, **TR2-D2 can re-use the samples generated from MCTS over an arbitrary number of iterations without having to recompute the log RND weights.**
> 3. To the best of our knowledge, we show the **first** successful framework for **multi-objective fine-tuning of discrete diffusion models** that balances tradeoffs between an arbitrary number of objectives. Our results demonstrate that this performs significantly better than the state-of-the-art inference-time multi-objective guidance approach PepTune while significantly reducing inference cost with optimal sequences generated via only a single diffusion pass. This has significant implications, particularly for biological applications where we aim to iteratively improve our model’s performance on many, often conflicting properties.
>
> ---
>
> **W2: Regarding the methods itself, I doubt the stability and the local optima problem the method may face. Please refer to questions to see how to get through these problems.**
>
> **See our answer to Q1.**
>
> ---
>
> ### **Addressing Questions**
>
> **Q1: I understand that Monte Carlo tree search may become stuck in local optima. If this occurs, how can we handle it to prevent the generated sequence from falling into such situations?**
>
> In principle, fine-tuning without MCTS requires $B$ independent runs of the model, which does not enforce exploration of high-reward regions of the generation space and can produce non-optimal sequences depending on the quality of the model at the given iteration. By contrast, **MCTS itself mitigates local optima** by balancing exploration and exploitation.
>
> Specifically, the **Upper Confidence Bound (UCB)** selection criterion in **Eq. 9** ensures that nodes with uncertain or underexplored rewards continue to be visited alongside those with high reward potential. This mechanism prevents the search from collapsing prematurely around a single high-reward branch.
>
> In addition, during the **expansion and rollout** steps, MCTS samples stochastically from the model’s policy, which injects Gumbel noise (Eq. 10) that enables the search to escape shallow local optima and discover alternative high-reward trajectories. Over successive simulations, backpropagation of cumulative rewards across the tree refines value estimates globally, allowing the algorithm to **asymptotically approach the optimal policy** rather than a local optimum. Increasing the scaling parameter $c$ of the selection score can also avoid becoming stuck in local optima.

---

> > ### Author Response · Authors · 2025-11-18
> > **Author Response to Reviewer (2/2)**
> >
> > **Q2: Does it depend on the initial model? If the initial model performs poorly, is this fine-tuning method still effective?**
> >
> > Our fine-tuning approach is **theoretically guaranteed** to converge to the optimal **reward-tilted distribution of the initial model $\mathbb{P}^0$ defined as $\mathbb{P}^\star\propto \frac{1}{Z}\mathbb{P}^0e^r$** (see proof **App. C.1**)**.** Therefore, if the distribution generated by the initial model $\mathbb{P}^0_T$ is a poor representation of the data distribution, the fine-tuning approach is likely to perform poorly as well.
> >
> > However, compared to **standard buffer generation approaches**, our tree search buffer generation step is much more likely to discover high reward sequences despite having low likelihood under the base model, compared to running independent samples from the policy.
> >
> > ---
> >
> > If we have sufficiently addressed all of your concerns, we kindly ask that you support the acceptance of our paper for poster presentation, and if concerns remain, we are happy to continue the discussion!
> >
> > Thank you,
> >
> > The Authors

---

### Author Response · Authors · 2025-11-18
**Summary of Key Updates and Responses to Common Reviewer Concerns**

# **General Response**

### **Summary of Key Updates**

1. We added the ablation of **TR2-D2 with no MCTS** to the main experiment tables (Table 1 and 2). Notably, we show that **TR2-D2 with no MCTS still outperforms all regulatory DNA benchmarks in the reward being optimized**, demonstrating the efficacy of our off-policy fine-tuning method on performance.
2. In response to Reviewer hz8y, we added an ablation for buffer size $B\in \\{5, 10, 20\\}$ in **Table 8 in App. F.** We demonstrate that TR2-D2 enhances performance compared to no MCTS for **smaller buffer sizes**, making it robust in memory-constrained settings.
3. We added multi-objective fine-tuning results for three additional proteins (GLAST, AMHR2, NCAM1) in **Table 5** to show that our method consistently outperforms inference-time guidance baselines across a diverse range of therapeutically relevant targets with **and without** existing peptide binders.
4. We justify how **TR2-D2 overcomes several known challenges** associated with off-policy reinforcement learning in **App C.2** of the updated manuscript.
5. To address reviewer concerns, we **clarify the trade-off between the number of MCTS steps used in each rollout generation and the total RL fine-tuning steps completed** during the use of a fixed compute budget in **Figure 7**, which results in the observed decrease in fine-tuning reward with higher search iteration on regulatory DNA design. We conclude that for single-objective tasks, fewer iterations of MCTS are sufficient, and for multi-objective tasks, larger iterations of MCTS will significantly enhance performance.
6. We clarify the computational cost of our experiments in **App. F.1** and provide compute time for increasing MCTS iterations in **Table 4**.
7. We provide the full codebase for our models and experiments in the anonymous repository: https://anonymous.4open.science/r/TR2-D2anon

### **Responses to Common Concerns**

**Novelty of TR2-D2**

While reviewers raised concerns that TR2-D2 is a simple combination of existing algorithms, we emphasize that the **key novelty lies in the following points:**

1. We show the **first implementation of the WDCE loss for off-policy discrete diffusion fine-tuning.** Notably, we show that our novel fine-tuning approach exceeds performance against baseline fine-tuning methods even **without MCTS, as shown in Table A below.**
2. Unlike other fine-tuning algorithms like GRPO which uses the exact likelihood of full trajectories under the current model and requires recomputing the log RND values of each sample from the tree, **TR2-D2 can re-use the samples generated from MCTS over an arbitrary number of iterations without having to recompute the log RND weights**.
3. To the best of our knowledge, we show the **first** successful framework for **multi-objective fine-tuning of discrete diffusion models** that balances tradeoffs between an arbitrary number of objectives.

**Table A: Comparison of reward values for 640 sequences across 3 seeds from model fine-tuned with TR2-D2 for regulatory DNA generation.**

| Method | Pred-Activity (median; ↑) |
| --- | --- |
| **Pre-trained** | 0.17 ± 0.04 |
| **CG** | 3.30 ± 0.00 |
| **SMC** | 4.15 ± 0.33 |
| **TDS** | 4.64 ± 0.21 |
| **CFG** | 5.04 ± 0.06 |
| **DRAKES** | 5.61 ± 0.07 |
| **SEPO** | 7.55 ± 0.01 |
| **GLID2E** | 7.35 ± 0.07 |
| **TR2-D2 w/o MCTS** | 9.13 ± 0.02 |
| **TR2-D2** | **9.74 ± 0.01** |

**Computational Cost**

We thank the reviewers for pointing out the importance of efficiency and scalability. Regarding computational cost, we emphasize that TR2-D2 amortizes the cost of tree search by fine-tuning on high-reward sequences stored in the replay buffer, which substantially reduces redundant exploration and makes the overall training procedure more efficient.

Empirically, we demonstrate the significance of this amortized search effect in **Tables 1 and 2**, where both single- and multi-objective fine-tuning performance improve markedly over baselines without MCTS, even with as few as **5 and 20 MCTS iterations per buffer update**, respectively. To directly address efficiency, **Table B** below summarizes the **GPU hours required for 1000 epochs** of peptide fine-tuning under varying numbers of MCTS iterations on a single NVIDIA A6000 GPU.

**Table B: Compute time on a single NVIDIA A6000 GPU for peptide fine-tuning with increasing number of MCTS iterations.**

| Number of MCTS iterations | Compute Time (1 NVIDIA A6000 GPU) |
| --- | --- |
| **0 (No MCTS)** | 5h 52m |
| **5** | 6h 42m |
| **20** | 10h 1m |
| **50** | 16h 0m |

**Challenges of Off-Policy RL**

The reviewers raised several important challenges associated with off-policy reinforcement learning relative to other RL methods. **We address each concern below to rigorously justify our choice of an off-policy formulation** and add this to **App C.2** of the updated manuscript.

---

### Author Response · Authors · 2025-12-02
**Summary of Reviewer Discussion and Revisions for Reassigned AC (1/3)**

**Dear Area Chair,**

First, we want to sincerely thank the reviewers for their careful evaluation of our submission. We are happy to see that the reviewers found that our paper **addresses a core issue in traditional RL fine-tuning (Reviewer Kyq1)**, that our **methodology is novel (Reviewer fhok)** with **high innovation (Kyq1)** and **theoretically-sound (Reviewer Kyq1, NznU)**, that our **multi-objective fine-tuning is relevant for balancing conflicting objectives in scientific applications (Reviewer fhok, NznU)**, and that our **improvements against SOTA baselines were “striking” (Reviewer NznU)**. We carefully interpreted the **key concerns** raised by each reviewer and substantially strengthened the paper with **new experiments, computational cost analysis, and clearer justifications of our methodological contribution.**

Given the AC reassignment, we would like to provide our new AC with a **concise summary of the outcomes of the reviewer discussion for our paper and the improvements we have made to our manuscript since the original submission**.

---

### **Summary of Responses to Common Concerns**

**Concern 1: Is TR2-D2 more than a simple combination of existing methods (MCTS + off-policy WDCE loss)? (Reviewers Kyq1, hz8y, NznU)**

While the reviewers **highlight that TR2-D2 provides significant empirical contributions** (Reviewer NznU), several reviewers questioned whether TR2-D2 provides a methodological advance beyond existing methods. **Our revisions clarify and demonstrate that TR2-D2 introduces three core methodological contributions:**

1. **First implementation of WDCE for off-policy fine-tuning of discrete diffusion:** We show that WDCE, originally developed for training neural samplers in Ising/Potts models (a completely different application), naturally extends to reward-tilted fine-tuning of masked diffusion models.
    - We also show that **TR2-D2 without MCTS already outperforms all SOTA baselines (Table 1)**.
2. **Synergistic integration of off-policy WDCE loss with structured search:** The WDCE loss uses only log-likelihoods from the frozen policy for computing importance weights, enabling re-use of highly optimal MCTS samples across arbitrarily many fine-tuning iterations without recomputing log-likelihoods (unlike GRPO/on-policy methods), as the target distribution $\mathbb{P}^\star$ is **fixed with respect to the frozen policy** and our unique remasking procedure enables efficient computation of the cross-entropy loss under the current policy. **This yields significant compute savings and amortizes the cost of MCTS search.**
3. **First successful framework for multi-objective fine-tuning of discrete diffusion models:** TR2-D2 learns to generate Pareto-optimal sequences with a single diffusion pass by fine-tuning on sequences that optimized for multiple objectives while minimizing tradeoffs, outperforming the state-of-the-art multi-objective inference-time guidance method PepTune on all new targets **(Tables 2 and 5)**.

---

**Concern 2: Is TR2-D2 efficient and scalable given the search cost from MCTS? (Reviewers hz8y, fhok, NznU)**

- We added GPU-hour measurements for different numbers of MCTS iterations $N_{\text{iter}}\in \\{0, 5, 20, 50\\}$ between each fine-tuning step **(Table 4).**
    - Notably, we find that **TR2-D2 achieves major performance gains with only a minor compute increase with 5-20 MCTS iterations.**
- We added detailed descriptions of the compute cost for both the DNA and peptide experiments in App. F.1.
- We emphasized that TR2-D2 **amortizes search cost by fine-tuning on high-reward samples stored in a buffer**, reducing redundant rollouts.
- We clarified that for peptide multi-objective optimization, inference requires only a single diffusion pass without making any reward evaluations or additional iteration loops, making **inference with TR2-D2 far cheaper than inference-time guidance methods like PepTune**.

---

**Concern 3: Why does increasing MCTS iterations hurt DNA results (Table 5) but help peptide results (Table 6)? (Reviewers fhok, NznU)**

We clarified that these experiments were run under a fixed compute budget, leading to a trade-off between the number of fine-tuning steps and search iterations:

- **DNA experiment (single-objective, easy reward landscape):** Fewer MCTS steps → more RL updates → better final reward. Only a few tree expansions are needed to find high-reward sequences.
- **Peptide experiment (multi-objective, complex reward landscape):** More MCTS steps → higher-quality sequences → better multi-objective learning.

To demonstrate this trade-off, we add a **new figure (Fig. 7)** that compares **fixed compute vs fixed RL steps**, resolving the discrepancy and showing that **TR2-D2 consistently improves with MCTS when compute trade-offs are controlled.**

---

> ### Author Response · Authors · 2025-12-02
> **Summary of Reviewer Discussion and Revisions for Reassigned AC (2/3)**
>
> **Concern 4: How does TR2-D2 overcome the challenges of off-policy RL (distribution mismatch, variance, update frequency)? (Reviewers hz8y, fhok)**
>
> The reviewers raised several important challenges associated with off-policy RL relative to other RL methods. **We address each concern below to rigorously justify our choice of an off-policy formulation** and add this to **App C.2 of the updated manuscript:**
>
> - **Distribution mismatch:** Our framework is rigorously derived using stochastic optimal control theory, which is theoretically guaranteed to converge to the optimal **reward-tilted distribution of the initial model $\mathbb{P}^0$ defined as $\mathbb{P}^\star\propto \frac{1}{Z}\mathbb{P}^0e^r$ with proof in App. C.1.**
> - **Comparison between on-policy and off-policy RL:** The difference in performance of on-policy and off-policy RL depends on the objective we use. In general, if we use an objective which is computed **under the expectation of the current policy $\mathbb{E}\_{\mathbb{P}^{u_\theta}}[\cdot]$**, then on-policy will outperform off-policy due to the variance in the importance weights. However, our objective takes the expectation under **optimal policy $\mathbb{E}\_{\mathbb{P}^{\star}}[\mathcal{L}(\theta; \boldsymbol{X}_T)]$ as defined in Eq. (7)**, which is **not dependent on the current policy**. For this objective, on-policy is not necessarily better than off-policy, and off-policy is much more efficient, as it amortizes the search cost by reusing high-reward samples from the buffer without recalculating the log RND weights of the current policy.
> - **High variance of importance sampling:** TR2-D2 uses sequence-level importance weights, which has far **lower variance** than token-level importance sampling in GRPO/PPO.
> - **Periodic model update frequency:** The target distribution is **fixed**, so importance-weight variance does not explode as training proceeds. MCTS also improves the quality of sequences in the replay buffer, which further minimizes the variance in importance weights.
> ---
>
> ### **Outcomes of Reviewer Discussion**
>
> Here, we give a concise summary of the individual concerns of each reviewer (that were not addressed in the **common concerns** above), our responses, and the status of our discussion with the reviewers:
>
> - **Reviewer Kyq1’s** primary concerns were regarding the methodological novelty **(which we address in Concern 1)** and that MCTS will get stuck in local optima.
>     - **MCTS itself mitigates local optima** by balancing exploration and exploitation with the selection criteria in Eq. 9, which is avoids getting stuck in local optima compared to generating the buffer with independent runs of the model.
>     - **No further questions were raised before the review period was cut short.**
> - **Reviewer hz8y’s** primary concerns were regarding methodological novelty **(which we address in Concern 1)**, computational cost of MCTS **(which we address in Concern 2)**, and challenges of off-policy RL **(which we address in Concern 4)**.
>     - Upon the reviewers' request, we have also **added the comparison to TR2-D2 without MCTS from the appendix to the main tables (Tables 1 and 2).**
>     - They also asked “**How does the size of the replay buffer B affect the performance of TR2-D2?**” and we **added an ablation for different replay buffer sizes $B\in \\{5, 10, 20\\}$ in Table 8.** We show that that all buffer sizes for TR2-D2 with MCTS outperform TR2-D2 without MCTS with buffer size 20 in multi-objective optimization, indicating the robustness of MCTS in across all settings where buffer size is limited.
>     - **No further questions were raised before the review period was cut short.**

---

> > ### Author Response · Authors · 2025-12-02
> > **Summary of Reviewer Discussion and Revisions for Reassigned AC (3/3)**
> >
> > - **Reviewer fhok’s** primary concerns were regarding challenges of off-policy RL **(which we address in Concern 4)**, the conflicting results for different MCTS iterations **(which we address in Concern 3)**, and computational cost of MCTS **(which we address in Concern 2).** They also asked “why the distribution mismatch can be mitigated by the MCTS buffer” and whether our combination of structured search + off-policy learning can be applied to other trajectory-aware tasks.
> >     - We highlight the rigorous theoretical proofs in **App C.1** that show that fine-tuning with TR2-D2 converges to the optimal reward-tilted distribution. Despite these theoretical guarantees, the performance in practice is limited by the quality of samples sampled and stored in the replay buffer, **which motivates the combination with MCTS to ensure high-reward samples are added to the replay buffer.**
> >     - We emphasize that our **search-augmented buffer strategy directly supplies any trajectory-based learning procedure with trajectories that better approximate the support of a reward-tilted path measure**. This means that our core framework (Algorithm 1) can be applied to continuous-time diffusion, discrete flow matching, autoregressive credit-assignment methods, and broader SOC-based fine-tuning frameworks.
> >     - Similar to Reviewer hz8y, they requested adding the comparison to TR2-D2 without MCTS from the appendix to the main tables (Tables 1 and 2), which we added in the revised version.
> >     - **No further questions were raised before the review period was cut short.**
> > - **Reviewer NznU’s** primary concerns were regarding the computational costs of MCTS **(which we address in Concern 2)**, conflicting results for different MCTS iterations **(which we address in Concern 3)**, and the purpose of the hyperparameters $N\_{\text{resample}}, \alpha$.
> >     - Following our initial response and explanation of hyperparameters (**App. F**), the reviewer responded by acknowledging that we “**provided good supplementary information**” and highlighted that our “**practical work is meaningful.**”
> >     - However, they had remaining questions on whether the **core methodological novelty was in the combination of MCTS with off-policy RL** and whether **sensitivity to different $N\_{\text{resample}}, \alpha$ hurts the generalizability of the method across tasks.**
> >     - We responded by confirming that while our off-policy RL fine-tuning method outperforms baselines even without MCTS, MCTS **further enhances the fine-tuning performance and is especially critical for multi-objective reward fine-tuning**.
> >     - **Regarding $\alpha$,** we clarify that even though performance varies between $\alpha=0.001$ and $\alpha=0.1$, this is the case for any fine-tuning method that aims to sample from the reward-tilted distribution defined in Eq. 4. Furthermore, we conclude **empirically and intuitively** that difficult tasks (multi-objective, large vocabulary) work better with higher $\alpha$ (i.e. $\alpha=0.1$) and simpler tasks work better with lower $\alpha$ (i.e. $\alpha=0.001$)
> >     - **Regarding $N\_{\text{resample}}$,** we provide intuition why a lower $N_{\text{resample}}$ works better for simpler tasks (avoids overfitting) and larger $N\_{\text{resample}}$ works better for more complex tasks.
> >     - **No further questions were raised before the review period was cut short**, but we are grateful to have had an insightful conversation with Reviewer NznU, which helped us justify some key points in our method.
> >
> > ---
> >
> > We would like to thank our original reviewers again for their time and effort in reviewing our paper. We regret that we can no longer engage in discussion with our reviewers, and would like to assure them that each of their reviews has improved our paper in many aspects.
> >
> > Although most of our reviewers have not yet had the opportunity to respond to our detailed rebuttals and new experimental results posted on November 17, **we believe that we have thoroughly addressed all concerns raised by reviewers and that, given more time, our discussions would have resulted in positive scores.**
> >
> > We kindly ask that our new AC acknowledge the contributions of our paper and our efforts to thoroughly address each of our reviewer comments and support our paper for acceptance to ICLR.
> >
> > Thank you,
> >
> > The Authors

---

### Meta-Review · Area_Chair_CgtR · 2025-12-17

**Summary:**

The authors introduce a method, TR2-D2, for fine tuning (conditioning or tilting) a diffusion model using off-policy RL with Monte Carlo Tree Search (MCTS) to optimize the rollout trajectories. The authors also use weight denoising cross-entropy (WDCE) as the expected reward, which naturally lends itself to off-policy optimization through the use of importance weights. A number of small molecule and peptide (one-shot/offline) design tasks are used to empirically demonstrate the method.

Strengths:
 - Some of the implementation specifics of MCTS are interesting and novel, e.g. the curation of a replay buffer.
 - Empirical testing of the method is thorough, though some reviewers question the fairness of the comparisons w.r.t. computation time allowed.
 - The MOO extensions are topical and relevant.
 - Paper is well written

Weaknesses:
 - Two reviewers were questioning the novelty of using MCTS+RL (off-policy) with diffusion policies. Reviewer NznU expressed this concern was not dealt with in the initial rebuttal from the authors.
 - Reviewer hz8y, fhok, and NznU questioned whether the added computational complexity of the approach was valid to compare to baselines with far less computational budget.
 - Multiple reviewers express concerns about the robustness and implementation of MCTS. E.g. sufficient details are not given, contradictory results are presented (related to the performance vs iterations of MCTS), sufficient detail not given about function and choice of hyper-parameters etc.

Also the summary to the AC had a possible inconsistency. That is the mention of eqn. 9 being UCB. While it looks related, it is not obvious that is the same quantity being estimated (this would require a derivation from the quantile definition of UCB, see chapter 7.8 of https://bayesoptbook.com/). UCB is also not mentioned in the paper. Furthermore, UCB is not a multi-objective acquisition function, and it's unclear how the second term (scaled probability) in eqn 9. would interact with the MOO reward from section 5. (However I like the idea of using non-dominance as a reward, and it relates to some other recent work on using this indicator as a way to estimate probability of hypervolume improvement).

I think there are some very interesting aspects of this paper, and it brings together many existing ideas in a novel way, with the result potentially being a powerful tool. While the authors have done a good job in attempting to address the reviewer concerns, I believe that this paper would still only achieve a borderline score at best, and would benefit from more time to incorporate this feedback, and to tighten up some of the formulation (proliferation of sensitive hyperparameters, reason for some of the reward terms, etc) and exposition.

**Reviewer Concerns:**

I believe many of the newer concerns were addressed by the rebuttal, however I believe a few major concerns could possibly remain:

- Multiple reviewers still expressed concerns relating to the novelty of the work, in particular NznU was unconvinced by the original rebuttal, and I'm concerned this was not adequately addressed (perhaps the contribution can be more precisely communicated in the paper).
- Experimental fairness. The authors did provide timings for their methods an an ablation for MCTS iterations, however no timing comparisons were made to the baseline methods.
- Some hyperparameters, e.g. $\alpha$, can greatly influence results and can be hard to set apriori. Results were reported for $\alpha=0.1$ and $\alpha=0.001$ for some experiments, but it is still unclear how a practitioner can robustly set this (and some other) hyperparamters. This is not unique to this method, but some more guidance or insight (e.g. a more thorough ablation) may be required.

**Reviewer Scores:**

As an upper bound, I think the reviewers could have raised their scores:

- Kyq1 - 2 -> 4
- hz8y - 4 -> 6
- fhok - 4 -> 6
- NznU - 4 -> 6

Which would be an average of 5 (borderline).

---

### Decision · Program_Chairs · 2026-01-26

Reject